# Expression-based subtypes define pathologic response to neoadjuvant immune-checkpoint inhibitors in muscle-invasive bladder cancer

A. Gordon Robertson[1], Khyati Meghani[2], Lauren Folgosa Cooley[2], Kimberly A. McLaughlin[2], Leigh Ann Fall[2], Yanni Yu[2], Mauro A. A. Castro [3], Clarice S. Groeneveld[4,5], Aurélien de Reyniès[6], Vadim I. Nazarov [7], Vasily O. Tsvetkov [7], Bonnie Choy [8], Daniele Raggi[9], Laura Marandino[9], Francesco Montorsi[10,11], Thomas Powles [12], Andrea Necchi[9,11] & Joshua J. Meeks [2,13] ✉

Checkpoint immunotherapy (CPI) has increased survival for some patients with advanced-stage bladder cancer (BCa). However, most patients do not respond. Here, we characterized the tumor and immune microenvironment in pre- and post-treatment tumors from the PURE01 neoadjuvant pembrolizumab immunotherapy trial, using a consolidative approach that combined transcriptional and genetic profiling with digital spatial profiling. We identify five distinctive genetic and transcriptomic programs and validate these in an independent neoadjuvant CPI trial to identify the features of response or resistance to CPI. By modeling the regulatory network, we identify the histone demethylase KDM5B as a repressor of tumor immune signaling pathways in one resistant subtype (S1, Luminal-excluded) and demonstrate that inhibition of KDM5B enhances immunogenicity in FGFR3-mutated BCa cells. Our study identifies signatures associated with response to CPI that can be used to molecularly stratify patients and suggests therapeutic alternatives for subtypes with poor response to neoadjuvant immunotherapy.

Over 80,000 people in the US are diagnosed with bladder cancer (BCa) each year, with almost 18,000 deaths[1]. Despite improvements in smoking cessation, surgery, and systemic therapy, there has been no improvement in survival for BCa for over twenty years. The current standard for locally advanced BCa (Stage II or greater, muscle-invasive bladder cancer, MIBC) is three to four cycles of cisplatin-based chemotherapy before radical cystectomy. In 2016, the FDA approved therapeutic antibodies targeting the program cell death protein-1/

[1]Dxige Research Inc., Courtenay, BC, Canada. [2]Departments of Urology, and Biochemistry and Molecular Genetics, Northwestern University, Feinberg School of Medicine, Chicago, IL, USA. [3]Bioinformatics and Systems Biology Laboratory, Federal University of Paraná, Curitiba, Brazil. [4]Université Paris Cité, Centre de Recherche sur l'Inflammation (CRI), INSERM, U1149, CNRS, ERL 8252, F-75018 Paris, France. [5]Oncologie Moleculaire, Institut Curie, Equipe Labellisée Ligue Contre le Cancer, Paris, France. [6]Université Paris Cité, INSERM U1138 Centre de Recherches des Cordeliers, APHP, SeQOIA-IT, Paris, France. [7]ImmunoMind Inc., Berkeley, CA, USA. [8]Department of Pathology, Northwestern University, Feinberg School of Medicine, Chicago, IL, USA. [9]Department of Medical Oncology, IRCCS San Raffaele Hospital and Scientific Institute, Milan, Italy. [10]Department of Urology, IRCCS San Raffaele Hospital and Scientific Institute, Milan, Italy. [11]Vita-Salute San Raffaele University, Milan, Italy. [12]Barts Experimental Cancer Medicine Centre, Barts Cancer Institute, Queen Mary University of London, London, UK. [13]Jesse Brown VA Medical Center, Chicago, IL 60611, USA. ✉e-mail: joshua.meeks@northwestern.edu

program cell death death-ligand-1 (PD-1/PD-L1) checkpoint (checkpoint inhibitors, CPI)[2]. Despite improved survival in second-line and even first-line metastatic BCas treated with CPI, only 25% of tumors are responsive to CPI; however, responders are likely to experience a durable cure[3]. Currently, the most effective therapy for locally advanced and metastatic BCa is unknown, with some treated with chemotherapy and others with immunotherapy. Therefore, accurate biomarkers that distinguish CPI responders from non-responders are critically needed. A "neoadjuvant paradigm" in which pre-treatment tumors are biopsied before administration of systemic therapy and cystectomy is essential to gain insights into mechanisms of therapeutic response[4–6].

Due to the limited but favorable response of metastatic BCa to CPI, multiple small Phase II trials have evaluated the pathologic response to CPI instead of chemotherapy before radical cystectomy. The largest trial of neoadjuvant immunotherapy for MIBC is PURE01 (NCT02736266), in which 143 patients were treated with three cycles of pembrolizumab before cystectomy[7–9]. PURE01 reported a complete pathologic response (ypT0N0) of 36%. Increased CD8+ immune infiltration, PD-L1 status, immune-infiltrated basal subtype, and tumor mutation burden (TMB) were all associated with favorable pathologic response to pembrolizumab[9]. ABACUS (NCT02662309)[10], a Phase II trial of neoadjuvant atezolizumab in patients with muscle invasive urothelial cancer, reported similar efficacy (31% complete pathologic response), but the response was not associated with TMB or PD-L1. While identifying biological features associated with response or resistance to CPI could help clinicians stratify patients who should receive neoadjuvant chemotherapy instead of immunotherapy or identify potential combination treatments to improve response and, ultimately, survival, the features associated with response to CPI remains an area of investigation. Here, we dissected the heterogenous cancer cell program associated with response and resistance to CPI using multi-omics analysis of pre-treatment and matched post-treatment tumors from PURE01 (82 pre- and 31 post-treatment). We identified five MIBC expression subtypes with distinct genomic, transcriptomic, and pathologic profiles. We developed a single-patient classifier, which we applied to identify the least-responsive subtype (S1, Luminal-Excluded/LumE) in an independent neoadjuvant CPI trial, ABACUS. Further investigation of S1/LumE tumors, which were enriched for FGFR3 mutations, identified KDM5B as a negative regulator of an immune-related gene network. Enhanced KDM5B activity was unique to S1 tumors, in contrast to the other CPI-resistant subtype, S4. Targeted inhibition of either FGFR3 or KDM5B enhanced immunogenicity in S1-like urothelial cells, suggesting a potential combination treatment for tumors classified as S1/LumE. These results suggest that the biologic classification of MIBC tumors may aid clinicians and patients in precision approaches to CPI prior to radical cystectomy.

## Results

### Clinical, pathologic, and molecular information

Patients were enrolled in the PURE01 clinical trial, a Phase II prospective study of neoadjuvant pembrolizumab before radical cystectomy (NCT02736266). We used tumor samples from 82 patients as a biomarker-evaluable subset representative of the overall study cohort of 114 patients. The median follow-up for 68 patients without progression was 22 months, and 14 patients had recurrence after a median of 11 months. A detailed summary of clinical and histopathological information is provided in Supplementary Data 1. Initial reports from the PURE01 trial included limited mutation and transcriptomic analysis[7–9]. We have now performed a comprehensive multi-omics analysis of 82 pre-treatment tumors. In addition, we have expanded our cohort to include 31 post-treatment tumors (27 matched pre- and post-pembrolizumab specimens), which may identify features associated with treatment response.

**Development of CPI-MIBC subtypes.** To identify molecular features associated with response to pembrolizumab in MIBC, we performed bulk RNA-Seq transcriptional profiling of 82 pre-treatment tumors. We first assessed gene sets for complete responders (CR, $n = 30$) vs. non-responders (NR, $n = 33$) for the overall cohort (Supplementary Fig. 1). While this analysis returned statistically enriched and repressed gene sets, we anticipated that the cohort would contain several molecular subtypes, and that characterizing these subtypes might provide insights into response or resistance to immunotherapy. Using unsupervised consensus clustering, we identified five transcriptomic subtypes that reflected the underlying transcriptional heterogeneity observed in MIBC and were associated with response to pembrolizumab (Fig. 1a–c, Supplementary Fig. 2a, b). Then, using a combination of histology and gene expression profiling, we developed names for the new tumor subtypes that attempted to describe their biology. Subtype 1 (S1) had the worst pathologic response rate, with 65% NRs (≥pT2 or N+), 19% PRs (≤pT1N0, including CIS), and only 15% CRs (pT0N0). S4 had a comparably low pathologic response rate, with 50% NR, 25% PR, and 25% CR. S3 and S2 had the highest pathologic CR rates, with 63 and 47% CR, respectively. For four of the five subtypes (S1, S2, S3, and S4), pathologic response correlated to recurrence-free survival (RFS) after radical cystectomy (Fig. 1b, c). Recurrence rates at two years were 27% for S1 and 38% for S4, while the subtypes with the best response rates had low two-year rates of recurrence of 7% for S2 and 5% for S3. Clinical response was discordant for S5, with an overall pathologic response of 50% but few relapses (7%).

Next, we sought to interrogate biomarkers previously associated with CPI response within the subtypes. The pembrolizumab (Dako 22C3) companion biomarker detects PD-L1 expression in tumor and immune cells and has been weakly associated with response to CPI[11]. Subtypes with high response rates (S2 and S3) had higher proportions of PD-L1(+) tumors. In contrast, subtypes with the lowest pathological response rates (S1, S4, and S5) were enriched in PD-L1(−) tumors (Fig. 1d). We observed a significant correlation between PD-L1 expression levels and pathological outcomes ($p = 0.011$, Pearson, Supplementary Fig. 2e).

Next, we characterized pre-treatment tumors using four expression-based MIBC classifiers: Lund[12,13], TCGA[14], consensusMIBC (cMIBC)[15], and MD Anderson (MDA)[16–18] (Fig. 1e). We found statistically significant similarities between our five pre-treatment subtypes and tumor classes annotated using these independent classifiers (Fisher's Exact tests, Bonferroni correction; Supplementary Data 2). Most S1 tumors were classified as luminal or luminal papillary by TCGA subtyping (24/26), as luminal by MDA subtyping (20/26), and as urothelial-like (Uro) by Lund subtyping (23/26). S2 tumors were comprised of luminal papillary by TCGA, luminal by MDA, and either genomically unstable (GU) or Uro by Lund. S4 tumors were classified as small cell/neuroendocrine-like (SC/NE-like), GU, or Uro by Lund and as luminal or neuronal by TCGA. S5 tumors were classified as mesenchymal-like (Mes-like) by Lund, luminal-infiltrated by TCGA, p53-like by MDA, and Stroma-rich by cMIBC. S3 tumors were enriched for basal (Ba/Sq) tumors by cMIBC and MDA. Thus, the PURE01 subtypes we describe reflect features of different subtypes across previously described classifications, suggesting no prior classification was directly comparable. However, compared to alternative classifications, our subtypes were more strongly associated with pathological responses to CPI (Supplementary Data 3).

To characterize the biological features of each subtype, we performed Gene Set Enrichment Analysis (GSEA) using MSigDB Hallmark[19] and Mariathasan[20] gene sets (Fig. 1f, g; Supplementary Data 4). Tumors from subtypes with limited response to pembrolizumab, S1 and S4, were both characterized by repressed inflammatory pathways. S1 tumors demonstrated a strong activation of FGFR3 pathways and repression of immune checkpoint and antigen presentation pathways. In contrast, S4 tumors showed an upregulation of MYC targets, cell

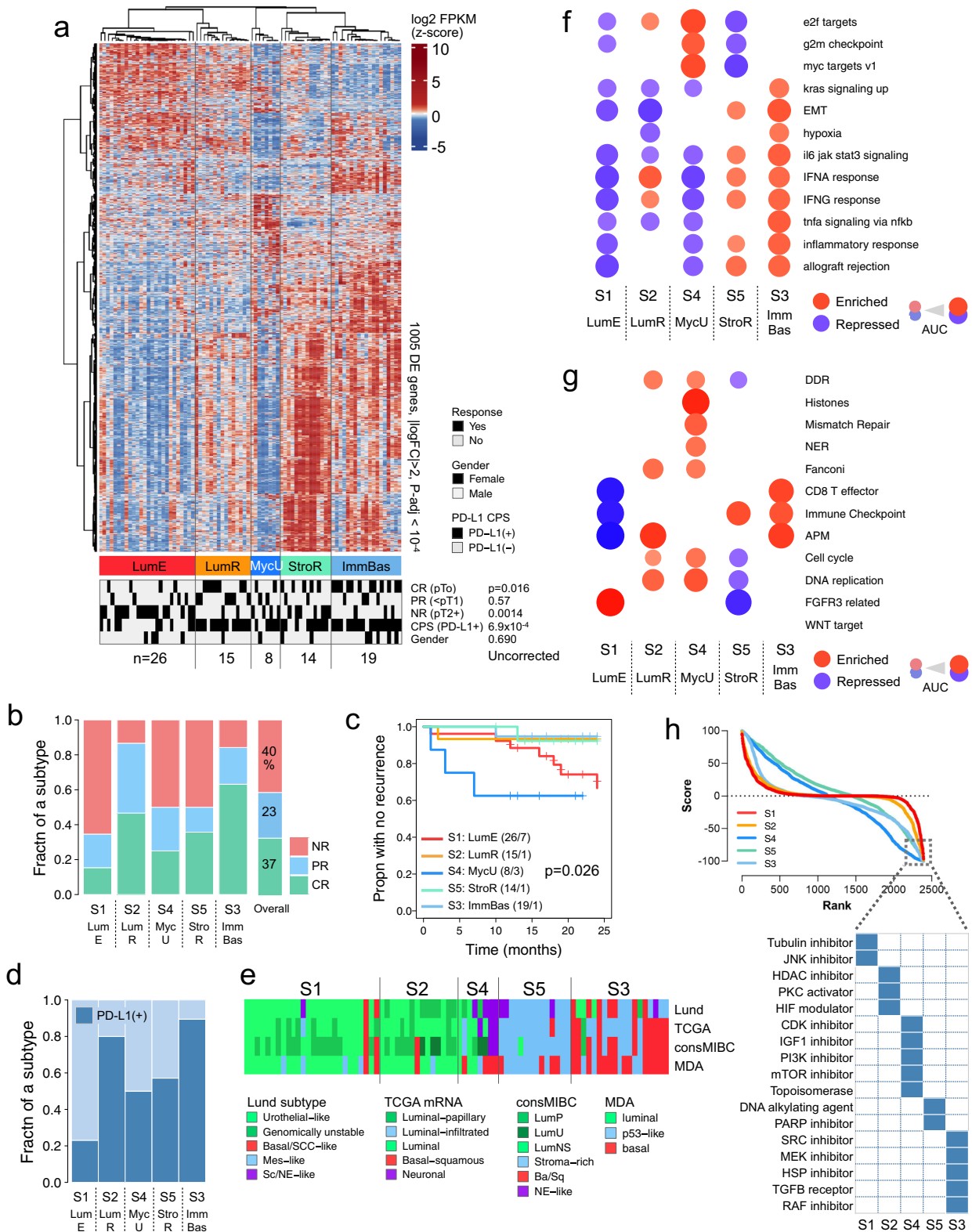

cycle, and signaling modules related to DNA damage (DDR, Fanconi). Thus, despite S1 and S4 having similar poor pathologic and clinical responses to pembrolizumab, they may have different biologic mechanisms of immune evasion. In contrast, responsive subtypes S2 and S3 demonstrated upregulation of genes involved in IFN-α and IFN-γ pathways and antigen presentation machinery (APM). These similarities and differences between subtypes highlight the heterogeneity of

MIBCs and may explain why no single biomarker has been predictive of response to CPI.

To gain further insight into the underlying biology of each subtype, we investigated Connectivity Map (CMap)[21] data, seeking to identify perturbations that resulted in signatures similar or opposed to the expression signature in each subtype (Fig. 1h). Each subtype had a unique enrichment of perturbagens with strong negative connectivity

**Fig. 1 | Overall characteristics of the PURE01 pre-treatment cohort. a** Heatmap showing five unsupervised consensus clusters, and 1005 differentially expressed genes satisfying p(adj) <10$^{-4}$ and |log2(FC)| > 2. Covariate tracks show response (CR complete response, PR partial response, and NR non-response), PD-L1 (+/−) status from a Dako 22C3 combined positive score (CPS) assay (+ corresponds to ≥10%), and gender. *P*-values are from two-sided Fisher exact tests, and are uncorrected for multiple hypothesis testing. **b** Fraction of PURE01 samples in consensus subtypes that had a complete response (CR), partial response (PR), and non-response (NR). The stacked bar at the right shows the overall responses for the cohort. **c** Kaplan−Meier plot of recurrence for the five PURE-01 MIBC expression subtypes, censored at 24 months, with a log-rank *p*-value. **d** PD-L1 +/− status, shown as fractions of samples in each subtype. **e** Predicted Lund, TCGA, consensusMIBC, and MD Anderson subtypes for PURE-01 *n* = 82 expression subtypes. *P*-values for the covariate tracks are from Fisher exact tests and were Bonferroni-corrected for multiple hypothesis testing (x4). **f, g** Dot representation of GSEA AUCs for selected **f** MSigDB Hallmark gene sets, and **g** Mariathasan et al. 2018 gene sets for the five subtypes. Enriched (vs. repressed) gene sets are shown as red (vs. blue) discs, with disc areas proportional to the areas-under-the-curve (AUCs) of the CERNO test results. **h** For the PURE01 subtypes, CMap v1.0 connectivity score-rank distributions, with a binary heatmap showing chemical perturbagens with the most negative scores. The dotted box highlights perturbagens that have large negative connectivity scores.

scores that reflected that subtype's underlying transcriptional changes. For example, for the subtype S4, in which cell-cycle genes were upregulated by GSEA (Fig. 1f, g), CMap identified CDK inhibitors among the top negative correlators, likely reflecting the underlying CDK activity of S4 tumors, which have a proliferative phenotype. Conversely, in S3 tumors, with upregulation of immune checkpoints and EMT signatures, TGF-β receptor inhibitors were predicted to be a potential treatment.

Finally, we evaluated the histomorphology of tumors from the five subtypes, comparing the epithelial, immune, and stromal components (Supplementary Fig. 2f). In general, S1 and S2 tumors had prominent papillary arrangement; however, S1 tumors were depleted of immune cells (immune deserts), while S2 tumors were enriched with immune infiltrate. S4, S5, and S3 tumors had comparatively less papillary architecture than S1 and S2 but had rather more diffuse tumor involvement of the lamina propria and muscularis propria. While both S5 and S3 had increased immune infiltrates, in S5, the immune cells were excluded/separated from the tumor by stromal cells (immune excluded). S4 tumors had few to no immune cells. Areas of necrosis were seen in 2 of the S1 tumors, 4 of the S3 tumors, and 4 of the S4 tumors. Based on the histologic characteristics and expression signatures, we gave each subtype a name that described its unique pathologic findings (Fig. 1).

**Assessing CPI-MIBC subtypes in an independent cohort.** ABACUS is the only other trial of single-agent neoadjuvant CPI in MIBC. We used GLMnet[22] to develop a single-patient RNA-Seq subtype classifier to validate our findings in an independent cohort. We applied it to the ABACUS pre-treatment cohort of 84 tumors (Fig. 2a; Supplementary Fig. 3a–d; Supplementary Data 5). To evaluate the accuracy of the classifier, we compared the PURE01 subtypes to the predicted ABACUS subtypes, assessing Hallmark gene sets, CMap perturbagens, PD-L1 status, and responses to CPI treatment (Fig. 2b–f). GSEA results for Hallmark gene sets were broadly consistent between PURE01 and ABACUS (Figs. 1f, 2b, Supplementary Data 6). CMap connectivity scores for perturbagens showed significant (*p*≪10$^{-30}$) Spearman correlations between PURE01 and corresponding ABACUS subtypes (Supplementary Fig. 4), and perturbagens with significant negative connectivity scores were in many cases similar in related subtypes (Fig. 1h, Supplementary Fig. 2g). PD-L1 expression was consistent between PURE01 and ABACUS cohorts (Fig. 2d), with higher PD-L1 in S2 and S3. We then compared the response rates in each ABACUS and PURE01 subtype (Figs. 1b, 2e). ABACUS tumors classified as S1 had an 18% CR rate, which was below the 24% for the overall cohort. Interestingly, in ABACUS, we observed decreased overall response (OR: CR + PR) in predicted subtypes S2 and S3 (Fig. 2f). This response translated into clinical outcomes. Tumors in ABACUS classified as S4 had the worst recurrence, and 25% of S1 tumors recured by 24 months, similar to PURE01. Yet, S2, S3, and S5 had fewer recurrences in PURE01 but had similar recurrence rates to S1 in ABACUS (Fig. 2c). The higher recurrence rates of S2, S3, and S5 in ABACUS correlate to the limited pathologic response to atezolizumab in these subtypes (Fig. 2e). Given

the similarity of the per-subtype molecular features and gene-enrichment pathways between the cohorts, we hypothesize that this OR difference may be secondary to the differential response of subtypes S2, S3, and S5 to different CPI agents (pembrolizumab vs. atezolizumab), differences in trial design, and inclusion characteristics. To further validate our findings, we evaluated a third cohort, the adjuvant IMvigor010 trial of 670 patients with MIBC[23,24] (NCT02450331). In this Phase III MIBC trial, participants were randomized to atezolizumab or observation after radical cystectomy. When we applied the classifier to cystectomy specimens of the cohort, we found that the "responsive" subtypes (S2, S3, and S5), treated with 12 months of atezolizumab, had an improvement in outcomes compared to "unresponsive" subtypes (S1 and S4) (*p* = 0.046) (Fig. 2g). In the observation cohort we identified no difference in recurrence between subtypes (Supplementary Fig. 2h). Collectively, validation in two additional trials suggest that MIBCs may be evaluated by transcriptional subtype, which may help to identify CPI-responsive tumors.

**Somatic mutation and copy number variation in CPI-MIBC subtypes.** To identify the genetic features of each subtype, we ompared somatic mutations and copy number variations for tumors in the PURE01 and ABACUS cohorts compared to mutations from ABACUS. We started by evaluating the association of Hallmark gene sets to mutations found in at least 10% of tumors. We found mutation patterns associated with the repression or activation of the cell cycle, DNA repair, and inflammatory pathways (Fig. 3a). Tumors in the S1 subtype had frequent mutations in KRAS, FGFR3, and KMT2C and had repressed Hallmarks of immune suppression (IFN-α and IFN-γ response, allograft rejection, IL6-JAK-STAT3 signaling), whereas S2 tumors had frequent mutations in the replication stress response kinase ATR, and had upregulated inflammatory gene sets. Interestingly, mutations in FGFR3, KRAS, and KMT2C were found at a higher frequency in S1 tumors, the 'immune desert' subtype with the highest frequency of non-responders; in contrast, tumors with ATR mutations were more frequent in subtypes with higher fractions of responders in S2. Overall, responders had higher frequencies of mutations in DNA damage repair genes (ATR, KMT2A, FANCD2, CDK12, and PALB2) than non-responders (Fig. 3b).

We then characterized the distribution of somatic mutations and copy number variations (CNVs) in each expression subtype (Fig. 3c). Overall mutation frequencies in the PURE01 and ABACUS cohorts were comparable to those previously reported for the TCGA MIBC cohort (Supplementary Fig. 5a–c). S1, the subtype with the worst pathologic response, had frequent, mutually exclusive mutations in FGFR3 (35%) and KRAS (23%), and amplifications in PPARG (23%) (Supplementary Fig. 5b). CCND1 amplifications, which have recently been associated with poor response to CPI in a pan-cancer analysis[25], were more frequent in S1 tumors (31%) (Supplementary Fig. 5a). Enrichment of S1-specific FGFR3 mutations and CCND1 amplifications was confirmed in ABACUS (Supplementary Fig. 6a). Subtypes with high response rates (S2 and S3) were frequent

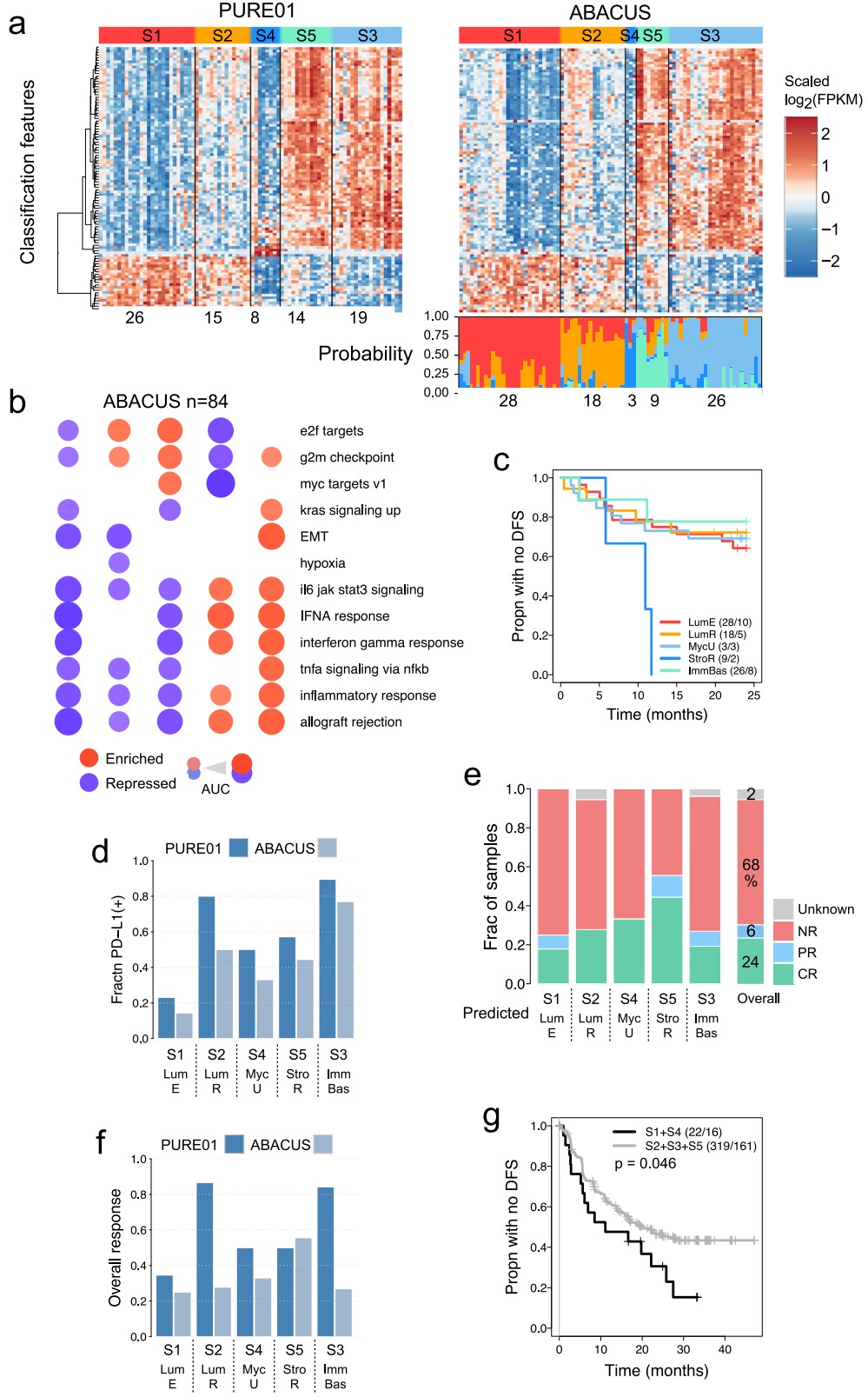

mutations in ATR (S2: 33%, S3: 16%); and TP53, the most frequently reported mutation in TCGA BLCA[14], was also frequently mutated in these subtypes (S2: 73% and S3: 68%). While the distribution of tumor mutational burden (TMB) was not statistically associated with subtype ($p = 0.49$, Kruskal–Wallis test) (Supplementary Fig. 5c), TMB trended towards being positively correlated with response ($p = 0.10$, Kruskal–Wallis test) (Supplementary Fig. 5d).

**Subtype differences in the tumor microenvironment by bulk and spatial analysis.** The compositions of immune and stromal cell populations of the tumor microenvironment (TME) have been associated with the immune response to CPI. For example, an 'inflamed' immune phenotype, with increased immune cells and low numbers of fibroblasts, is associated with improved response to CPI compared to immune-desert and immune-excluded TME[10,20,26]. We sought to

**Fig. 2 | Identifying and characterizing predicted PURE01 subtypes in the ABACUS cohort. a** The GLMnet classifier. Left: Heatmap of the 100 features used by the classifier, shown for the PURE01 pre-treatment cohort (*n* = 82) and its consensus subtypes. Right: Heatmap of the 100 classifier features in the ABACUS pre-treatment cohort (*n* = 84), with semi-supervised clustering within each of the predicted subtypes. The covariate track above the heatmap shows the predicted PURE01 subtype calls, while the covariate track below the heatmap shows the prediction probabilities for each subtype in each ABACUS sample. **b** GSEA results for selected MSigDB Hallmark gene sets for the five classifier-predicted subtypes in the ABACUS cohort. Enriched (vs. repressed) gene sets are shown as red (vs. blue) discs; disc areas are proportional to the areas-under-the-curve (AUCs) of the CERNO test results. **c** Kaplan–Meier plot for DFS for predicted subtypes in the ABACUS *n* = 84 pre-treatment cohort. **d** PD-L1(+) status in each PURE01 subtype and each classifier-predicted subtype in ABACUS. **e** Response (CR, PR, NR) for the classifier-predicted PURE01 subtypes in the ABACUS pre-treatment cohort. **f** Overall response (OR = CR + PR) for PURE01 subtypes and predicted subtypes in the ABACUS pre-treatment cohort. **g** Kaplan–Meier plot for DFS, in the atezolizumab arm of the IMvigor010 *n* = 670 MIBC cohort, for predicted subtypes S1 + S4 vs. predicted subtypes S2 + S3 + S5. The *p*-value is from a log-rank test, and is not corrected for multiple comparisons.

evaluate the unique stromal and immune features of the TME in each subtype by both bulk RNA-Seq and spatial profiling (NanoString GeoMX Digital Spatial Profiling, DSP). Using ESTIMATE[27] and MCP-counter[28] immune deconvolution algorithms, we observed differences between subtypes in immune cell populations (Fig. 4a, Supplementary Data 7, 8). We performed DSP to complement the bulk deconvolution results (Supplementary Figs. 7a–c, 8). Overall, the two subtypes with the lowest pathologic response rates, S1 and S4, had the lowest ImmuneScores by ESTIMATE analysis, and the fewest T cells, CD8+ T cells (cytotoxic T lymphocytes, CTLs), B cells, and myeloid dendritic cell counts by MCP-counter, suggestive of an 'immune desert' TME. The subtypes with the highest pathological response rates, S2 and S3, had higher ImmuneScores and levels of immune cell populations (T cells/CTLs, B cells, and myeloid dendritic cells), consistent with immune-inflamed tumors. By histology (Supplementary Fig. 2f), S5 tumors appeared to be 'immune excluded,' as they had increased immune cells and dense stromal fibroblasts that restricted the immune cells from infiltrating the tumor. By ESTIMATE and MCP-counter, S5 tumors had higher ImmuneScores and StromalScores, and higher levels of fibroblasts and other immune cell types, confirming the immune-excluded phenotype observed by histology (Supplementary Figs. 7a, 9a). In addition, S5 tumors had the highest expression of cancer-associated fibroblast (CAF) markers (Supplementary Fig. 9b).

Tumor-infiltrating B-cells have long been associated with improved prognosis and clinical outcomes[29]. Applying the Immuno-Mind platform's R package immunarch to TRUST4-processed bulk RNA-Seq data, we tested PURE01 pre-treatment B-cell repertoires for specific immunoglobulin signatures associated with response to neoadjuvant pembrolizumab. Gene usage analysis of the IGH, IGK, and IGL segments showed that some V(D)J genes were used more frequently. Fitting generalized linear models for overall response rate (OR: CR + PR) vs. NR, we found that the level of IGHJ1[30] varied across subtypes and was lower in OR samples than in NR samples ($p = 3.4 \times 10^{-3}$, Chi-square test) (Supplementary Fig. 9d). As well, in four of the five subtypes, the mean IGHJ1 level was inversely related to the frequency of complete responders (CR vs. CR + NR), with higher IGHJ1 usage in S1 and S4 subtypes (Supplementary Fig. 9e).

While deconvolution algorithms can quantify immune cell populations within a tumor from bulk RNA-Seq data, the spatial organization of immune cells in the TME is an important determinant of anti-tumor immunity[31]. To assess features of TMEs associated with the response at the spatial level, we performed proteomic DSP, measuring 71 proteins in pre-treatment tumors from 5 complete responders (CR) and three non-responders (NR) (Fig. 4b). This method allows non-destructive, image-based microdissection of distinct compartments of the tumor and TME as multiple areas of interest (AOIs) in each sample. To distinguish the unique proteomic profiles in the TME that were associated with response to CPI, we used principal component analysis (PCA) to capture the differences between the CR and NR samples from the luminal subtypes S1 and S2 and from S4 (Fig. 4c). While TME AOIs from complete responders were relatively concordant, TMEs from non-responders were separated by principal components, suggesting significant inter-TME heterogeneity between the non-responders of different subtypes. For example, for NR AOIs, the first principal

component (PC1) captured global differences in immune infiltration, with immune-infiltrated subtype S2 separating into a distinct cluster with higher expression of lymphocyte markers (CD3, CD4), immune activation markers (ICOS, β2-microglobulin or B2M, TIM-3) and the immune-suppressive marker IDO1. In NR AOIs, the two immune-desert subtypes, S1 and S4, separated strongly across PC2, with a stronger correlation observed for regulatory proteins such as CTLA4 and OX40L.

To further investigate features associated with resistance to pembrolizumab within each subtype, we then compared the expression of immune cell markers and immune regulatory proteins in TME AOIs from CR and NR tumors (Fig. 4d). TMEs from the immune-desert subtypes S1 and S4 had lower levels of immune marker and regulatory proteins than TMEs from S2. Conversely, non-responders of inflamed S2 tumors had increased immune markers (CD3, CD4, CD8), co-stimulatory proteins (B7-H3, ICOS), and the inhibitory immune checkpoint protein IDO1, suggesting that an exhausted immune phenotype may contribute to S2's decreased pathologic response in non-responders. In summary, we profiled pre-treatment immune cell types within each subtype by immune deconvolution of bulk RNA-Seq data with further evaluation by spatial protein data for TMEs. We found greater divergence of immune cell markers and regulatory proteins of the TME of S1, S2, and S4 from non-responders (NR) compared to responders (CR) from the same subtypes.

**Comparison of pre- and post-treatment tumors.** To further dissect tumor and TME features specifically associated with resistance to pembrolizumab, we evaluated 31 post-treatment samples, of which 27 were paired samples taken pre- and post-treatment from the same anatomic site (Supplementary Data 9). We first re-analyzed the cohort of 113 tumors (82 pre- and 31 post-treatment tumors) by unsupervised consensus clustering (Supplementary Fig. 2c, d, Supplementary Data 10). Samples from the original five subtypes clustered together, particularly S1, S2, and S4, while 24 (77%) of the 31 post-treatment tumors clustered within two new subtypes (hereafter named S6 and S7) (Fig. 5a). Approximately half of the resistant tumors (14, 52% of 27 matched pairs) originated from S1, and 11 of these (79% of 14) were classified as S6 or S7 after pembrolizumab (Supplementary Data 11). By MIBC expression subtyping, S6 had more Mes-like tumors, while S7 was comprised of luminal-infiltrated tumors (Supplementary Data 12). By histology, S6 tumors had more 'scar-like' stromal features. In contrast, S7 tumors were characterized by increased immune cell infiltrate (Fig. 5b, Supplementary Fig. 10). S6 tumors had a higher rate of nodal metastasis than S7 (5/11 for S6 vs. 1/10 for S7, *p* = 0.15, Fisher's Exact test) (Fig. 5c), with no statistical difference in tumor volumes by morphometric analysis (*p* = 0.65, t-test) (Fig. 5d, Supplementary Data 13).

GSEA with MSigDB Hallmarks showed that both post-treatment subtypes S6 and S7 had upregulated expression of EMT genes, as well as repressed genes associated with cellular proliferation (G2M checkpoint and E2F targets) (Fig. 5e). Hallmarks for S7 reflected an immune-reactive phenotype, with upregulated inflammatory gene sets (allograft rejection, inflammatory response, IFN-α, and TNF-α response). For Mariathasan gene sets, both S6 and S7 had repressed cell-cycle and DNA damage response and repair pathways (DDR, Mismatch repair,

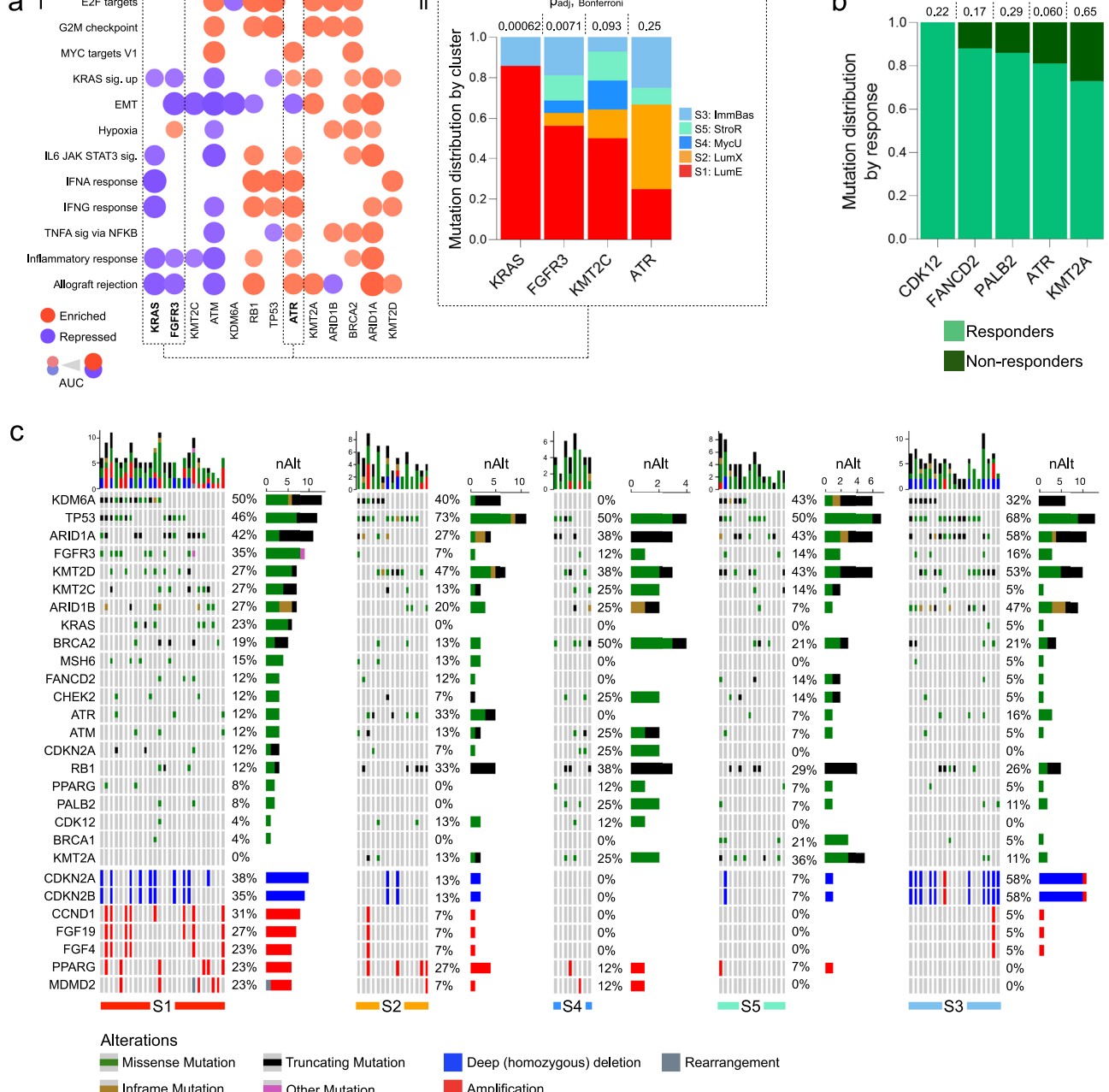

**Fig. 3 | Somatic mutations and copy number alterations in CPI-MIBC PURE-01 expression subtypes. a** Dotplot (left, i): Hallmark gene-set analysis for PURE01 tumors with mutations in 13 selected genes (relative to WT). Barplot (right, ii): fraction of samples in a subtype with somatic mutations in KRAS, FGFR3, KMT2C, or ATR. Exact $p_{adj}$, given for each gene, were calculated as follows. *P*-values were calculated using two-sided Pearson's chi-square tests, and were Bonferroni-corrected for multiple comparisons. **b** Relative mutation frequency of five selected genes, stratified by response (CR or PR; non-response is NR). *P*-values were calculated using two-sided Pearson's chi-square tests, and were Bonferroni-corrected for multiple hypothesis testing. **c** Oncoprints for somatic mutations and somatic copy number alterations in the five $n = 82$ PURE-01 expression subtypes. The top-to-bottom gene order is set by decreasing alteration frequency in the Luminal-Excluded (S1) subtype. For each subtype, horizontal bars to the right indicate the number of samples with an alteration in that gene, and the types of alterations. Barplots at the top of each subtype's oncoprint show the total number of genetic alterations in the oncoprint genes, colored by the alteration type.

Fanconi, DNA replication) (Fig. 5f), contrasting with pre-treatment results (Fig. 1g). By comparing immune deconvolution for the 27 matched-pair tumor samples, we identified that post-treatment tumors had increased immune and stromal populations and decreased tumor purity, with all but two tumors demonstrating an increase in immune score, and all but five with an increase in the stromal score (Fig. 5g, Supplementary Data 9, 14). Specifically, post-treatment subtypes S6 and S7, particularly S6, showed an increase in

fibroblast populations ($p = 4.1 \times 10^{-9}$, Kruskal test) (Supplementary Fig. 9a–c, Supplementary Data 15).

Tumors resistant to neoadjuvant therapy are associated with recurrence and progression, and analysis of post-treatment tumors from PURE01 may identify new therapeutic options for patients with CPI-resistant bladder cancers. We applied the CMap perturbagen gene signatures to identify potential therapeutic targets for S6 and S7 (Fig. 5h, i). This identified several compounds

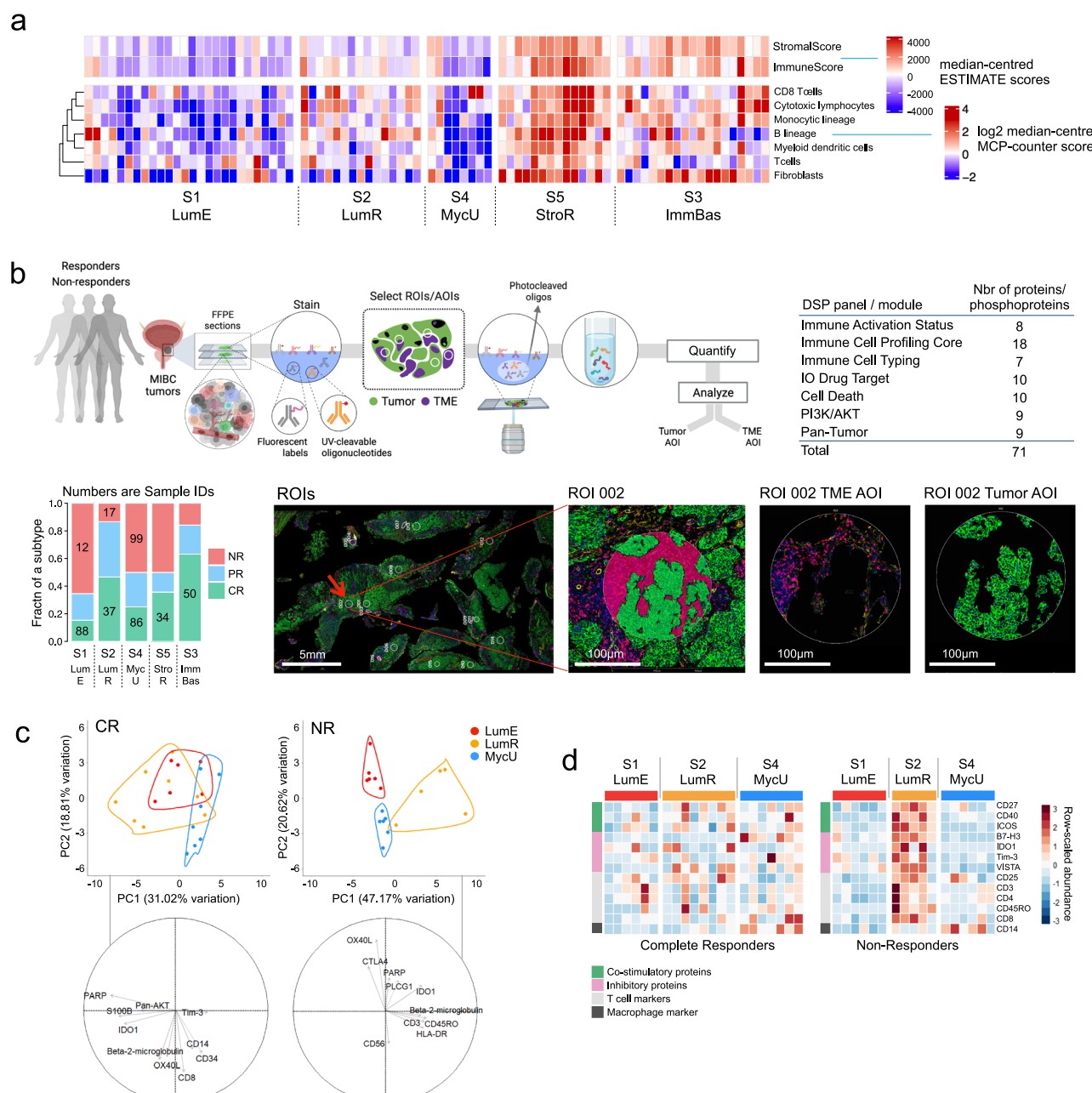

**Fig. 4 | Pre-treatment subtypes have distinct immune infiltration patterns.**
**a** Heatmaps showing ESTIMATE Immune and Stromal scores, and immune cell scores from MCP-counter for the five pre-treatment expression subtypes. See also Supplemental Fig. 4 and the 'Comparing ESTIMATE, MCP-counter and DSP' section in Results. **b** Nanostring GeoMx digital spatial profiling (DSP). Above left: schematic of the DSP workflow. Above right: protein panels used, with the number of proteins or phosphoproteins in each panel. Below left: a response-by-subtype barplot (as in Fig. 1b) showing sample numbers/IDs for the eight samples for which we generated DSP data (three CR-NR pairs for S1, S2, and S4, and one additional CR for both S5 and S3). Below right: a representative digital micrograph image of a stained slide, on which white circles show Regions-of-Interest (ROIs), and a red arrow indicates ROI 2. ('Representative' implies manual selection, with no biological or technical replicates.) Enlarged, this ROI is shown to the right, with green PanCk+ stain

indicating the tumor cells and red-purple CD3 + stain indicating the tumor micro-environment (TME). The two micrographs further to the right are ROI 2's color-filtered Areas-of-interest (AOIs) for TME and Tumor, from which the DSP protein signals were generated. **c** Principal component analysis (PCA) for TME AOI proteins for PURE01 subtypes LumE (S1), LumR (S2), and MycU (S4), comparing complete responders (CR, left) and non-responders (NR, right). Above: PCA similarity plots. Outlined areas indicate sets of AOIs that correspond to a subtype, and X- and Y-axis labels indicate the percentage of total variation explained by the first two principal components. Below: In loading plots, arrow lengths and directions indicate the relative contribution of an important protein to principal components PC1 and PC2. **d** Heatmaps of DSP protein abundance for TME AOIs (columns) for a complete responder and a non-responder from PURE01 subtypes S1, S2, and S4, for 13 immune regulatory proteins and immune markers.

with large negative connectivity scores for each subtype, potentially identifying alternative treatments. For S6, FGFR inhibitors and DNA alkylating agents were among the top negative correlations, suggesting increased FGFR pathway activity in this subtype. In contrast, S7 expression profiles were similar to more

immunologically active signatures generated by treatment with a protein kinase C (PKC) activator, consistent with the elevated expression of immune gene sets in this subtype[32]. Interestingly, for S7, CMap analysis also identified MAPK and MEK inhibitors among the top negative correlations, suggesting upregulated MAPK/ERK

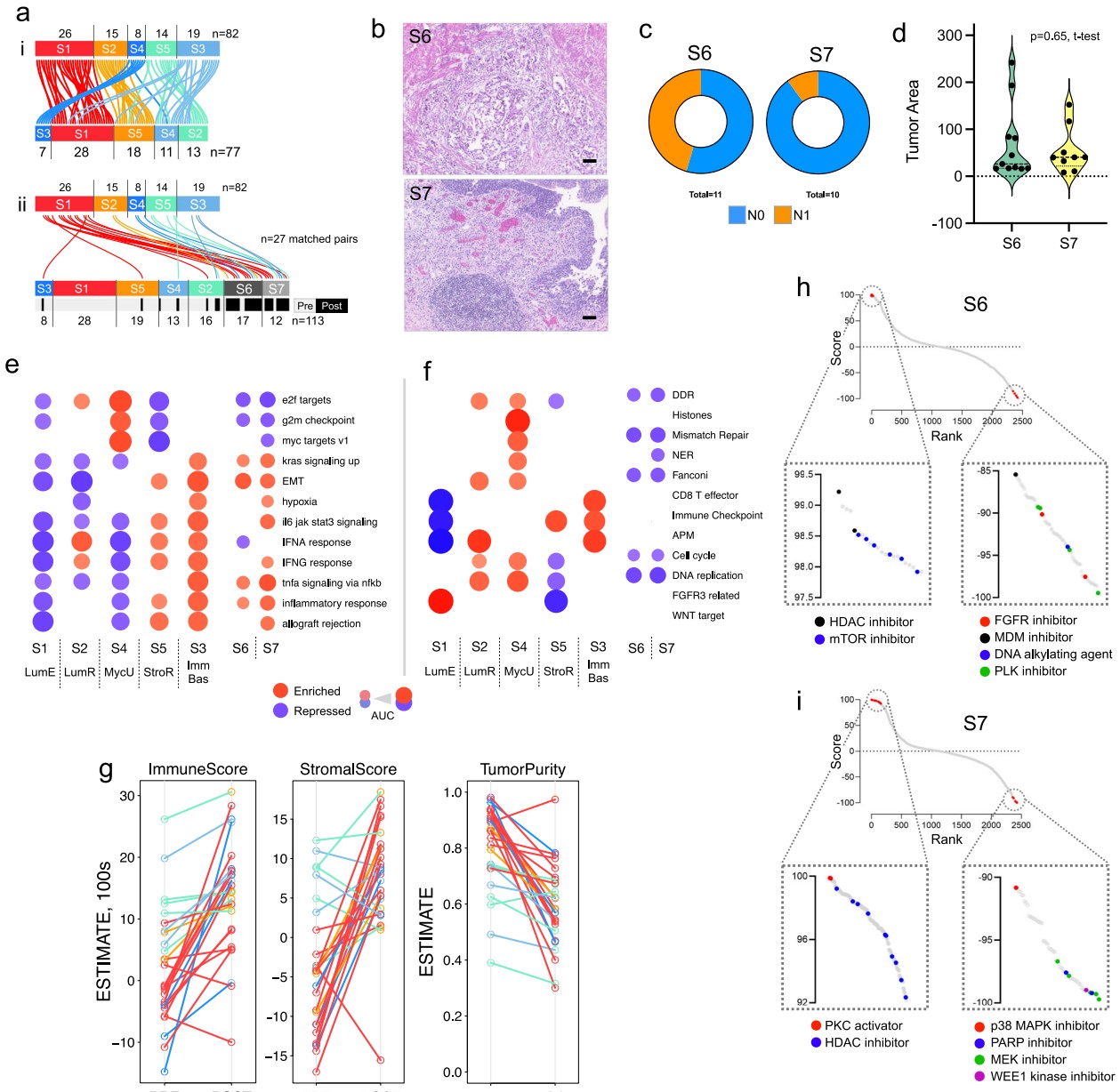

**Fig. 5 | Comparison of pre- and post-treatment PURE01 samples. a** Sankey-like diagram showing: (above) five unsupervised clusters for pre-treatment samples (*n* = 82) and (below) seven clusters for pre+post samples (*n* = 113). Each bezier curve shows the positions of one sample in two consensus clustering solutions. Below, bezier curves are drawn only for the *n* = 27 matched pre-post sample pairs. The *n* = 113 clusters are colored to indicate subtypes that correspond between the *n* = 82 and *n* = 113 clustering results. The gray-black covariate track indicates pre- and post-treatment samples in the *n* = 113 clustering solution. A two-sided Fisher exact test that compared pre/post-treatment status to seven consensus subtypes returned *p* = 8.7 × 10⁻¹³, uncorrected for multiple comparisons. **b** Representative H&E-stained micrographs for post-treatment subtypes S6 and S7. ('Representative' implies manual selection, with no biological or technical replicates.) Scale bars are 0.1 mm. **c** Fraction of samples in S6 (*n* = 11) and S7 (*n* = 10) that were lymph node N0 vs. N1 at radical cystectomy. **d** Distributions of tumor areas in subtypes S6 and S7. Results

were generated from RNA-Seq data for the *n* = 113 PURE01 pre- and post-treatment cohort, with no biological or technical replicates. Dots represent individual samples. The *p*-value is from a two-sided Student's *t*-test. **e**, **f** GSEA results for S6 and S7 using **e** MSigDB Hallmark gene sets and **f** Mariathasan gene sets. See the legend for Fig. 1f, g. **g** Changes in ESTIMATE ImmuneScore, StromalScore, and tumor purity, in pre and post-samples, with lines colored as in (**a**), for the *n* = 27 matched sample pairs. Results were generated from RNA-Seq data for the *n* = 27 PURE01 pre/post-treatment matched sample pairs, with no biological or technical replicates. Dots show matched-pair pre-or-post-treatment samples; lines show pre-to-post changes for an individual sample. **h**, **i** Connectivity score-rank distributions for CMap v1.0 perturbagens identified for subtypes **h** S6 and **i** S7. For each subtype, insets show details of the chemical perturbagens with the largest positive and negative connectivity scores and highlight a subset of these perturbagens.

pathway activity in this subtype. In addition to identifying possible adjuvant therapies, the CMap analysis confirmed that S6 and S7 subtypes have different biologies and suggested that subtype-specific combination therapies may have a role after cystectomy in CPI-resistant tumors.

**FGFR3 and KDM5B are potential therapeutic targets in subtype LumE/S1.** Because 11 (52%) of the 21 CPI-resistant tumors in S6 and S7 were originally from S1 (Fig. 5a), we next sought to identify regulators of the repressed inflammatory gene network observed in S1 tumors. First, we applied regulon analysis to uncover transcription factors that

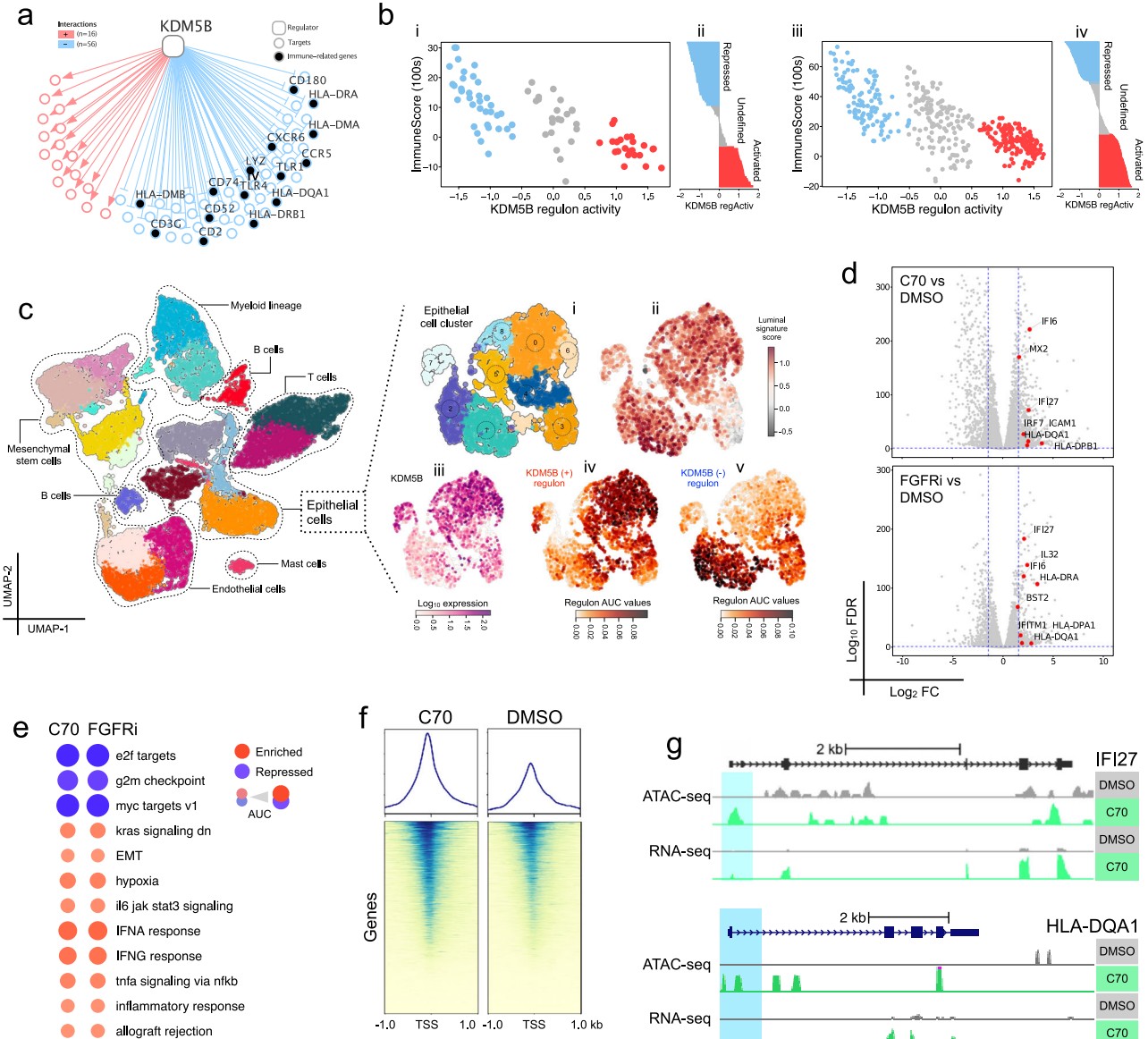

**Fig. 6 | KDM5B and FGFR3 are potential therapeutic targets to activate an immune response in S1 CPI-MIBCs. a** KDM5B regulon target genes (red = positive targets, blue = negative targets), with immune-related genes marked by black discs. **b** Validating the negative association between KDM5B regulon activity and ESTIMATE ImmuneScore. (i) The relationship of KDM5B regulon activity to ESTIMATE ImmuneScore in the PURE01 pre-treatment cohort, with samples (dots) colored by KDM5B regulon activity status. (ii) Rank-sorted profile of KDM5B regulon activity across the PURE01 pre-treatment cohort, showing activated (n = 23, 28%), undefined (n = 23, 28%), and repressed (n = 36, 43%) cohort subsets. (iii) Relationship of KDM5B regulon activity to ESTIMATE ImmuneScore in the TCGA-BLCA cohort (n = 404). (iv) Rank-sorted profile of KDM5B regulon activity across the TCGA-BLCA cohort, showing activated (n = 111, 27%), undefined (n = 146, 36%), and repressed (n = 147, 36%) cohort subsets. **c** Left: Unsupervised clustering of single-cell RNA sequencing data from three human MIBC tumors identified 19 clusters consisting of

cells from the tumor, immune, and stromal compartments. Right: (i) Unsupervised reclustering of scRNA-Seq data for epithelial cells identified nine sub-clusters. (ii) Distribution of a luminal signature score in the epithelial cell sub-clusters. (iii) KDM5B expression in the epithelial cell sub-clusters. (iv, v) AUCell scores reflect the activity of KDM5B(+) and KDM5B(−) regulons in a given cell. **d** Volcano plots for differentially expressed genes from bulk RNA-Seq data for RT4 cells treated with the KDM5Bi C70 or an FGFRi. **e** Dot representation of GSEA AUCs for enriched vs. repressed Hallmark gene sets in RT4 cells treated with C70 and FGFRi See the legend of Fig. 1f, g. **f** Heatmaps of the ATAC-seq signal profiles in RT4 cells treated with the KDM5i C70 or with DMSO as a control, centered on transcriptional start sites. **g** ATAC-seq and RNA-Seq peak profiles at the interferon-inducible IFI27 and HLA-DQA1 gene loci in RT4 cells treated either with DMSO as a control or the KDM5 inhibitor C70. Pale blue rectangles highlight regions around transcriptional start sites.

regulate the expression of inflammatory genes in S1 tumors. This identified the histone H3K4 demethylase KDM5B as having multiple negatively regulated immune target genes (Fig. 6a). KDM5B regulon activity was highest in 18 (69%) of 26 S1 samples and was most strongly and consistently repressed in subtype S5, which had the highest median ImmuneScore (Supplementary Fig. 11a, b; Supplementary Data 7). In comparison, the KDM5B regulon was repressed in subtypes with higher ImmuneScores (S5 and S3) (Supplementary Data 7, 16), and

we confirmed this negative relationship in the independent TCGA-BLCA cohort (Fig. 6b, Supplementary Data 18). We then used STRING[33] network analysis and Reactome pathways to infer and functionally interpret the interactome of KDM5B's negative target genes and found that the network was enriched in genes involved in immune modulation (TNFR1-induced NFkB signaling and the TLR4 cascade) (Supplementary Fig. 11c, Supplementary Data 19). Finally, we used single-cell RNA-Seq from human MIBC tumors to establish that epithelial cells

expressed luminal markers, that KDM5B was relatively highly expressed in subsets of the epithelial cells, and that activated and repressed KDM5B regulon activities were consistent with higher and lower KDM5B expression levels (Fig. 6c, Supplementary Fig. 11d).

To determine whether KDM5B can repress immune target genes in urothelial cancer, we used the KDM5-inhibitor (KDM5i) C70[34,35] in S1-like RT4 bladder cancer cells, which harbor an oncogenic FGFR3-TACC3 fusion[36]. The enzymatic inhibition of KDM5 with C70 in these cells caused upregulation of genes that were negative KDM5B regulon targets (Fig. 6d). Since 35% of S1 tumors had mutations in FGFR3 (Fig. 3c), and the FGFR3 gene set was upregulated in S1 (Fig. 1g), we compared KDM5B inhibition to an FDA-approved mutant-FGFR3-directed therapy, erdafitinib[37]. Of the 1519 genes significantly differentially expressed by KDM5Bi treatment (|log2FC| >1.5, FDR < 0.05), 454 were also significantly differentially expressed after treatment with this FGFR inhibitor (FGFRi) ($p = 2.4 \times 10^{-286}$, Fisher's Exact test) (Supplementary Fig. 11e). Importantly, GSEA indicated that treatment of RT4 cells with either the KDM5i C70, or with erdafitinib, upregulated an inflammatory phenotype associated with expression of genes from IFN-$\alpha$, IFN-$\gamma$, and IL6 JAK-STAT3 gene sets (Fig. 6e, Supplementary Fig. 11f, g). Given KDM5B's role as a histone demethylase of H3K4me3 and H3K4me2[35,38], we further characterized the mechanism of KDM5i-C70 activity in RT4 cells by comparing areas of open or closed chromatin using bulk ATAC-Seq data. Our results suggest a global increase in open chromatin signals close to transcription start sites (Fig. 6f) and enrichment of open chromatin regions at transcription start sites for immune regulatory genes, such as the IFN-$\alpha$ regulated gene IFI27, and the major histocompatibility complex gene HLA-DQA1 (Fig. 6g). Taken together, our results suggest that subtype-specific targeting of KDM5B or FGFR3 in S1 tumors can alter chromatin accessibility profiles, and can enhance immunogenicity to make S1 tumors potentially become more responsive to immunotherapy. KDM5B regulon activity is high in the S1/Lum-E tumor subtype, in contrast to the other resistant subtype, S4/MycU, suggesting that these two subtypes may have disparate mechanisms of resistance. Active regulons in S4 included YY1, DNMT1, and the MYC-regulated gene MAZ1 (Supplementary Fig. 11a), which have been associated with PD-L1 resistance in melanoma[39].

## Discussion

The neoadjuvant paradigm, in which a tumor is sampled before systemic therapy, and then again post-treatment at radical cystectomy, may offer opportunities to identify biological features that can guide higher-efficacy treatment strategies for BCa patients. Currently, no reliable biomarkers are available to guide the selection of individualized neoadjuvant or adjuvant chemo- or immunotherapy in MIBC patients. Unfortunately, the companion PD-L1 biomarker for CPI therapy has not shown a strong association with the response across trials with anti-PD1/PDL1 antibodies, challenging investigators to find other biologic predictors. Our study describes comprehensive multi-omic profiling of MIBC tumors from PURE01, the largest study available of neoadjuvant CPI before radical cystectomy. The study's overall goal was to dissect the molecular heterogeneity of MIBCs to further define features associated with resistance.

By unsupervised consensus clustering, we identified five transcriptomic MIBC subtypes associated with clinical and pathological responses to pembrolizumab (Fig. 7). We assessed each subtype for genetic and transcriptomic features that may be associated with the observed pathologic immune phenotype. Signatures with enhanced IFNA and IFN, such as S2 and S3, had a better response than tumors with a low IFN expression (S1 and S4). In addition, we sought to determine the unique drivers of immune resistance of each tumor subtype. S1 luminal tumors had a high frequency of mutations in FGFR3 and KRAS, along with repressed inflammatory gene signatures. S4 tumors had immune desert phenotype characterized by cooperative transcriptional activity of MYC- and KRAS-driven immune evasion

programs[40]. S5 tumors had a stroma-rich, immune-exclusion phenotype with a high frequency of T cells in the TME with increased IFN signatures; however, S5 had few to no immune cells infiltrating the tumors and had elevated expression of immune checkpoint signatures, potentially explaining this subtype's relatively poor pathologic response rate. Yet, the mechanistic differences between clinical and pathologic responses require further investigation. Collectively, our data identified expression signatures that were associated with intrinsic resistance to CPI, providing further granularity into features associated with response to checkpoint immunotherapy.

We integrated bulk transcriptome and spatial protein data profiles to identify features of non-responsive tumors (Fig. 7). Spatial analysis of the tumor microenvironment from a non-responding S2 sample identified elevated protein expression of the immune suppressive checkpoint marker IDO1. The multi-faceted role of IDO1 in the suppression of T-cell responses is well-studied in several different cancers[41]. IDO1 inhibition in combination with anti-PD-1 therapy (nivolumab) and chemotherapy is being tested in a Phase III clinical trial for MIBC (NCT03661320)[42,43]. While further investigation of the spatial heterogeneity is necessary for a larger cohort, our results may help explain the lack of CPI activity in the small fraction of non-responders within the S2 subtype.

Prior work has suggested activating mutations in FGFR3 may be associated with an immune-evasive phenotype and poor response to CPI[44]. Upper tract urothelial carcinoma (UTUC), which harbors mutations in FGFR3 in 74% of patients[44], exhibited a particularly poor response to adjuvant nivolumab in the CheckMate274 trial[45] (HR = 1.16, 95% CI = 0.62–2.13 for renal pelvis, and 1.55 (0.7–3.45) for ureteral tumors). Herein, we identified a higher frequency of FGFR3 mutations in S1 tumors and KDM5B as a regulator of the immune suppression program in this subtype. While only 35% of tumors had mutations in FGFR3, FGFR3 expression was enhanced in tumors from S1. In vitro, we demonstrated an increase in immune pathway gene sets with KDM5B or FGFR3 inhibition in a luminal cell-line model representative of the S1 subtype. While erdafitinib has been associated with immune activation in lung cancer[46], little is known about mutant FGFR3-dependent immune regulation in BCa. We demonstrated that KDM5B inhibition alters the chromatin accessibility of FGFR3-mutated RT4 cells, resulting in increased expression of proinflammatory genes. These results suggest that genetic and epigenetic drivers of tumor subtypes could be targeted to transform a resistant tumor into a more CPI-responsive state. Transcriptomic analysis identified an increase in immune infiltration and stromal expansion in post-treatment, pembrolizumab-resistant tumors. Furthermore, in S6-subtype post-treatment tumors, CMap analysis identified FGFR inhibitors as a potential therapeutic target, likely reflecting a continued dependence of these tumors on S1 gene expression patterns post-therapy, e.g., FGFR3-related pathways. If an S1 tumor were identified before treatment, it might be possible to change its susceptibility to CPI by using an inhibitor targeting FGFR3 or KDM5B. In contrast, S7 tumors were associated with greater immune activation, and CMap analysis identified the MAPK pathway and WEE1 as potential targets in these tumors. A combination of WEE1 inhibitors with CPI is currently being evaluated in a Phase Ib study in patients with muscle invasive bladder cancer (NCT02546661[47]). These examples further support using an RNA-based platform to identify potential new therapeutics.

We acknowledge the limitations of this study. We developed tumor subtypes from a unique cohort with limited sample numbers and attempted to validate the subtypes in two cohorts of tumors from different clinical trials. ABACUS and PURE01 had different rates of PD-L1 positive tumors, clinical stage, and duration/doses of therapy. In addition, the two cohorts were treated with different immune checkpoint blockade agents (the PD-1 inhibitor pembrolizumab and the PD-L1 inhibitor atezolizumab, respectively) and assessed PD-L1 with different IHC assays. At a technical level, RNA and DNA from the

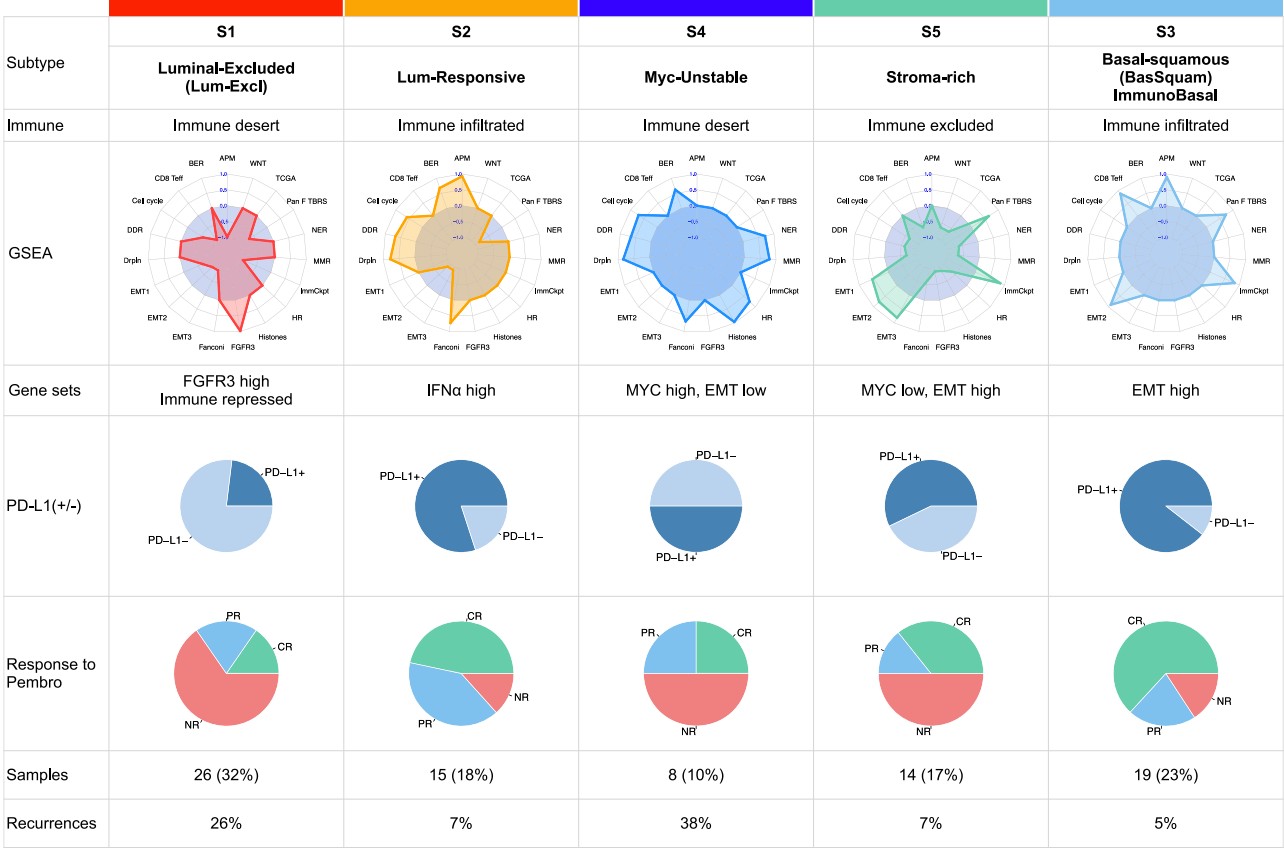

**Fig. 7 | Summary characteristics of PURE-01 consensus expression subtypes.**
Top to bottom: cluster number; subtype name; immune class (desert, infiltrated or excluded); GSEA signed areas under the curve (AUCs) for gene sets from Mariathasan et al. 2018[20], positive vs. negative AUCs indicate enriched vs. repressed sets and AUC = 0 indicates either that no result was returned from a CERNO[52] test with qval <0.05, or $p_{adj}$ > 0.1; GSEA summary, largely from MSigDB Hallmark gene sets; PD-L1 status (CPS > 10% for PD-L1+); response to pembrolizumab treatment (CR, PR, NR); number of samples and percent of the n = 82 cohort; and finally the percent of samples in a subtype that had recurrences within 24 mo.

tumors were processed by different extraction and sequencing methods, and we did not attempt to correct for potential batch effects. We hypothesize that differences in outcomes within subtypes between trials may be secondary to differences in immunotherapy drugs; however, we cannot validate this hypothesis with mechanistic studies. In addition, our spatial proteomic evaluation involved areas of interest from a limited number of tumors; further investigation of spatial heterogeneity should be performed in MIBC. Finally, as almost all post-treatment tumors from ABACUS were from non-responsive tumors, we could not compare the effect of immunoediting or how TCR/BCR diversity changes in responsive tumors. Similarly, tumors from IMvigor010 were biased towards more aggressive tumors.

Our work identifies the heterogeneity associated with responses to CPI in muscle invasive bladder cancer. We describe five subtypes with unique genomic and transcriptomic cancer cell programs and distinct tumor microenvironments with differential responses to neoadjuvant pembrolizumab in bladder cancer. In addition, we have uncovered FGFR3 mutations and KDM5B overexpression as resistance biomarkers to CPI and describe therapeutic alternatives that can help guide patient stratification for combinatorial therapies.

# Methods

## Clinical data and metadata
For ABACUS, clinical/metadata were previously reported, and we accessed FASTQ files from the European Genome-Phenome Archive (EGAS00001004445). For ABACUS, we describe how pathological response was assigned as CR, PR, NR, and Unknown in the 'Pathological

response' section below. We defined PD-L1(+) as IC2+. For the IMvigor010 cohort, we downloaded metadata from EGAD00001007575.

## H&E micrographs
Unstained formalin-fixed and paraffin-embedded (FFPE) bladder tissue/tumor sections were received from the PURE-01 study. A representative slide from each patient was stained with hematoxylin and eosin. Images were taken with a Zeiss Axioskop/Nuance microscope using a 10x objective.

## Tumor morphometry
To quantify the amount of tumor in each bladder specimen, the H&E-stained slides were scanned at low magnification and digitized using the TissueFAXS system (TissueGnostics, Los Angeles, CA, USA). The acquired images were analyzed using TissueGnostics' HistoQuest cell analysis software. Prior to morphometry, the tumor was marked by a GU pathologist (BC) with a black marker. We then used the automated region of interest (auto-ROI) function to identify the borders of the specimen and manually excluded any non-urinary bladder components that auto-ROI had included. Finally, in the HistoQuest software, we traced the area indicated as tumor by our pathologist and recorded the total tumor area (mm², including necrotic tumor) for each slide.

## RNA sequencing
For PURE01, the paraffin block at diagnosis was used as the primary tissue for evaluation. Paraffin blocks were stored at room temperature. A GU pathologist evaluated every section and identified the high grade

and invasive parts of the tumor, which were then macro-dissected for dual RNA and DNA extraction.

RNA was extracted with the Roche HighPure miRNA Kit (Roche #05080576001), following the manufacturer's instructions. Stranded total RNA-Seq was conducted in the Northwestern University NUSeq Core Facility. Briefly, RNA quantity was determined with a Qubit fluorometer. Total RNA examples were also checked for fragment sizing using Agilent Bioanalyzer 2100. The Illumina TruSeq Stranded Total RNA Library Preparation Kit was used to prepare sequencing libraries, following the manufacturer's recommendations, including rRNA depletion with RiboZero Gold, cDNA synthesis, 3′ end adenylation, adapter ligation, library PCR amplification, and validation. Libraries were pooled and sequenced on an Illumina HiSeq 4000, generating 50 bp single-end (SE) reads at a sequencing depth of 20–25 M reads/sample. For ABACUS, RNA sequence data from FFPE samples, and clinical/metadata, were made available by T. Powles and Genentech from the European Genome-Phenome Archive (EGAS00001004445).

For RNA sequence data for both PURE01 and ABACUS, we trimmed Illumina adapters with TrimGalore v0.6.5. We aligned the trimmed reads to the GRCh38.p12 reference human genome with STAR v2.7.5a[48] and converted SAM files to sorted BAMs with Samtools v1.6. We generated read counts with 'htseq-count' from 'htslib' v0.11.2, using Ensembl v95 (GRCh38) gene annotations. To generate gene-level FPKM values, we used 'cuffquant' and 'cuffnorm' from Cufflinks v2.2.1[49], with these gene annotations. For the IMvigor010 cohort, we downloaded RNA-Seq FASTQ files and metadata from EGAD00001007575. We removed adapters from reads with Trimmomatic v0.39, then aligned trimmed reads to the GRCh38 reference genome with STAR v2.5.2, and generated read counts for Ensembl v95 gene annotations with HTseq count v0.11.1.

## Consensus expression subtypes

For PURE01 $n = 82$, we input FPKM profiles for the 1950 coding genes with the largest variance into ConsensusClusterPlus v1.52.0[50], and evaluated results for hierarchical, PAM, and k-means clustering, and for Pearson, Spearman, and Euclidean distances. Specifically, for runs using Pearson and Spearman distances, we input a distance matrix via as.dist(1 - cor(T, method = "…")), where T was the log10-transformed, median-centered FPKMs. We chose to work with a five-cluster solution with Spearman distances, PAM clustering, random subsets of samples set by pItem = 0.85, and 10,000 iterations. For PURE01 $n = 113$, we chose to report on a seven-cluster solution using Spearman distances, PAM clustering, pItem = 0.875, and 50,000 iterations.

We generated a heatmap for the consensus clusters (Fig. 1a) as follows. Using edgeR[51] v3.28.1's 'filterByExpr' function, we retained the 15,918 of the 19,951 coding genes that had sufficiently large RNA-Seq counts in the PURE01 $n = 82$ cohort (minimal count = 10, minimal total count = 15, minimal proportional count in the smallest group = 0.7). We then used DESeq2[52] v1.26.0 to identify genes that were differentially expressed (DEGs) between subtypes, using default parameters for pairwise contrasts. The heatmap shows FPKM gene expression levels for genes with $|log2FC| > 2$ and adjusted $P < 10^{-4}$.

To predict Lund, TCGA, and consensusMIBC subtypes across the consensus clusters, we used the R-based BLCAsubtyping and consensusMIBC subtype classifiers[12] with FPKM gene expression profiles. To transform ten Lund subtypes into five simpler Lund subtypes, we combined 'Ba/Sq' and 'Ba/Sq-Inf' into 'Ba/Sq'; 'GU' and 'GU-Inf' into 'GU'; and 'UroA-Prog,' 'UroB,' and 'UroC' into 'Uro.' We assessed the association of each set of (predicted) subtypes with CR/PR/NR response, using a Chi-square tests, then Bonferroni-correcting the $p$-values for multiple hypothesis testing.

## Pathological response

For PURE01 $n = 82$ pre-treatment samples, clinical data reported CR, PR, and NR pathological responses.

For ABACUS $n = 84$ pre-treatment samples, clinical data reported 'pathological complete response' (PCR) and 'major pathological response' (MPR) for each sample. We assigned response = CR to 20 samples that had PCR == "Yes" and MPR == "No", response = PR to 5 samples with PCR == "No" and MPR == "Yes", response = NR to 57 samples with PCR == "No" and MPR == "No", and response = Unknown to 2 samples with PCR == "Yes" & MPR == "Yes". For the IMvigor010 cohort, for MIBC samples in the atezolizumab trial arm, $n = 127$ samples with relapseID "relapse" were assigned as non-responders (NR), while $n = 144$ samples with relapseID = "nonrelapse" were assigned as complete responders (CR).

## Gene set enrichment analysis (GSEA)

For GSEA with $n = 30$ CR (complete responder) vs. $n = 33$ NR (non-responder) tumor samples from the PURE01 pre-treatment cohort ($n = 82$), we used FPKMs for 15,488 expressed coding genes. We sorted the gene symbols by the signal-to-noise ratio (S2N). Given two groups of samples (here, CR vs. NR, see 'Pathological response', above), we calculated S2N for each gene as the difference in mean FPKM in the two groups, divided by the sum of FPKM standard deviations[53]. We then used tmod v0.46.2, in R v4.1.2, to run CERNO tests on the ranked gene-symbol vector and on its reverse[53,54]. We tested 50 MSigDB v7.2 Hallmark[19] gene sets and, separately, 20 gene sets which were available as Imvigor210CoreBiologies_1.0.0.tar.gz from research-pub.gene.com/Imvigor210CoreBiologies. We reported tmod evidencePlots and tabular results, typically showing only gene sets with AUC > 0.60 and FDR < 0.01.

For each PURE01 expression subtype, we ranked ~15.5 thousand expressed protein-coding genes by the signal-to-noise ratio (S2N). For a similar analysis with predicted subtypes in ABACUS, we used ~15.8 thousand coding genes. Then, working with 50 MSigDB Hallmark gene sets and the twenty-gene set from Mariathasan et al.[20], we used CERNO tests[54], typically with qval = 0.05, to identify enriched and repressed gene sets for each subtype or predicted subtype. We summarized gene set results with 'dot' diagrams that we generated with ComplexHeatmap v2.6.2[55], setting dot radii to be proportional to the CERNO areas under the curve (AUCs).

To summarize these results, we thresholded CERNO results at pAdj (i.e., qval, FDR) = 0.1, then, for Fig. 7, we generated radar charts using the R package fmsb v0.7.0 and a custom R script. We displayed signed AUCs, i.e., using the CERNO test AUC value for an enriched gene set, but taking the negative of the AUC value for a repressed gene set. We set the y-axis limits in each radar chart. An AUC of zero indicates a gene set for which either (a) no result was returned from a CERNO test using a qval = 0.05 significance threshold, or (b) the result returned did not satisfy the Padj = 0.1 threshold.

## DNA extraction, sequencing, and analysis

For PURE01, DNA was extracted from FFPE tissue from pretherapy transurethral resection of the bladder (TURB) samples, and a hybridization-capture panel of approximately 400 cancer-related genes was sequenced to a median coverage of 743x, then analyzed for mutations and copy number alterations by Foundation Medicine (Cambridge, MA, USA), as described previously. For ABACUS, we downloaded mutation and copy number data from EGA as dataset EGAD00001006201. We generated oncoprints of mutations and copy number variations using ComplexHeatmap v2.6.2[55]. For comparisons of mutation frequencies between the different subtypes, we used R's Fisher's Exact test unless otherwise noted. In Fig. 3a, b, we used $p$-values that we Bonferroni-corrected for multiple hypothesis testing; in Fig. 3c, we used R's p.adjust() to correct $p$-values to FDRs.

To understand biological pathways impacted by mutations in a set of selected genes, we used DESeq2 v1.30.0[52] to rank genes by differential expression in samples in which a selected gene was mutated vs. samples with the wild-type gene. We processed each list of DEGs with

tmod v0.46.2, using MSigDB Hallmark v7.5.1 gene sets as an input, to identify gene sets that were enriched or repressed in the mutated samples relative to the wild type. We used ComplexHeatmap v2.10.0 to generate dot graphics representing the area-under-the-curve (AUC) values from a tmod GSEA analysis for each gene mutation.

### Kaplan–Meier plots
For time-to-event analysis, we used the R survival package v3.2–7 to generate Kaplan–Meier (KM) plots and calculate log-rank $p$-values. For PURE01, we calculated times-to-events as follows: if RELAPSE == 1, time = Time to recurrence; otherwise, time = Time to last follow-up. Times were from the date of the first pembrolizumab to the date of relapse (i.e., recurrence). Prior to KM calculations, we censored data for status and times-to-event to 24 months. For the ABACUS pretreatment cohort ($n = 84$), and the Atezolizumab arm of the MIBC samples in the IMvigor010 cohort, we used clinical DFS data.

### Immune cell type deconvolution
We deconvolved bulk RNA-Seq data with MCP-counter v1.2.0[25] and with ESTIMATE v1.0.13[27] (bioinformatics.mdanderson.org/estimate/rpackage.html). To generate heatmaps for MCP-counter, we log2-transformed the FPKM inputs to MCP-counter result for a cell type (adding one); median-centered each record; removed results for NK cells, Endothelial cells, and Neutrophils because they had very low scores; then generated a heatmap with ComplexHeatmap v2.6.2, using the original CCP heatmap column order, and splitting columns by consensus clusters. We re-ordered subtypes in the PDF file by hand, using Affinity Publisher v1.9.3. For ESTIMATE results, we median-centered the output scores, then generated a heatmap with ComplexHeatmap v2.6.2. We compared ESTIMATE and MCP-counter results to distributions of digital spatial profiling (DSP) data for 71 proteins, SNR-normalized, for areas of interest (AOIs) corresponding to each slide's tumor microenvironment (TME) (see DSP, below). To test differences in ESTIMATE ImmuneScore and StromalScore for $n = 113$ subtypes S6 vs. S7 (Fig. 5h), we used Kruskal–Wallis tests, then applied a Bonferroni (x2) correction for multiple hypothesis testing. We used Kruskal–Wallis tests to assess differences in MCP-counter fibroblast results for $n = 113$ S6 and S7 vs. S1-to-S5, and to assess differences in ESTIMATE ImmuneScore and StromalScore for $n = 82$ S5 and S3.

### Expression of cancer-associated fibroblast (CAF) genes across $n = 82$ subtypes
For five marker genes (CFD, COL1A1, DCN, LUM, PTGDS) for inflammatory CAFs (iCAFs) and five (ACTA2, CALD1, MYL9, RGS5, TAGLN) for mCAFs[56], we used a custom R script to generate box-whisker plots of FPKMs of each gene across the PURE01 $n = 82$ subtypes. For S5 and S3, we compared FPKMs for each gene with a Kruskal–Wallis test, then transformed the 10 $p$-values into adjusted $p$-values with a Bonferroni correction for multiple hypothesis testing.

### Gene usage for IGH, IGK, and IGL subrepertoires
We processed RNA-Seq data for $n = 82$ PURE01 pre-treatment samples with TRUST4[57] v0.2.054, then processed the TRUST4 output with the immunarch v0.6.5 R package (ImmunoMind, Berkeley CA). In TRUST4's output, we separated the mixed TCR and BCR receptor sequences and removed out-of-frame, partial, and erroneous CDR3 amino acid sequences. While the data contained too few TCR sequences to allow estimating TCR diversity, it contained a large enough number of BCR sequences to support robustly estimating BCR repertoire diversity. We assessed gene usage for IGH, IGK, and IGL subrepertoires, retaining the $n = 74$, 75, and 66 of 82 samples that had at least 100 clonotypes for each subrepertoire, and fitting generalized linear models against overall response rate (ORR = CR or PR, $n = 49$) vs. NR ($n = 33$).

### PD-L1+/− status
For PURE01, using data from a Dako 22C3 immunohistochemical (IHC) assay, we scored samples as PD-L1(+) if the combined positive score (CPS) was >10%. We set "Neg" CPS values to 0.0 and deleted the "%" and "<" symbols. We calculated a covariate independence $p$-value for PD-L1(+) vs. subtypes with a Fisher exact test. For ABACUS, using PDL1_IC (by immune cells) data from a Ventana SP142 IHC assay (which consisted of values IC0, 1, or 2+), we scored samples as PD-L1(+) when PDL1_IC was 2+.

### Regulon analysis
We calculated a transcriptional regulatory network for 1612 transcription factors (TFs) using RTN v2.13.2, as described elsewhere[58,59]. The TFs are available from RTN's 'tfsData' object as 'Lambert2018'. Briefly, RNA-Seq expression profiles were used to estimate the associations between a TF and all of its potential targets. We used two metrics to identify potential TF-target associations: Mutual Information (MI) and Spearman's correlation. MI-based inference indicates whether a TF's expression is informative of the expression of a potential target gene, while Spearman's correlation indicates whether the 'direction' of an inferred TF-target association is positive or negative. Associations whose MI were below a threshold were eliminated by permutation analysis (BH-adjusted $p$-value <0.026), and unstable interactions were then removed by bootstrapping. RTN regulons were additionally processed by the ARACNe algorithm, which uses the data processing inequality (DPI) theorem to enrich the regulons with direct TF-target interactions[58]. The regulatory network construction and analysis were performed in R v3.x (R-Core-Team, 2020).

### Validation of the KDM5B regulon in the TCGA-BLCA cohort
To validate the KDM5B regulon and the negative association between KDM5B regulon activity and immune scores in the TCGA-BLCA cohort ($n = 404$), we used two approaches. In the first approach, we compared TF-target associations inferred for the PURE01 $n = 82$ KDM5B regulon targets ($n = 72$ targets; 56 negatives and 16 positives) with those for the TCGA-BLCA RNA-Seq data[14]. We downloaded the batch-corrected RNA-Seq data for the TCGA-BLCA cohort ($n = 404$) from gdc.cancer.gov/node/977, and used RTN to compute Mutual Information (MI) between KDM5B and each of its inferred PURE01 regulon target genes, in the TCGA-BLCA RNA-Seq data, using RTN's tni.permutation() function to assign a BH-adjusted $p$-value to each MI value. We then stringently filtered by the TCGA-BLCA adjusted $p$-values ($p_{adj} < 0.001$) and required that each PURE01 gene symbol be present in the TCGA-BLCA RNA-Seq data. This retained 59 (82%) of KDM5B's 72 PURE01 target genes (Supplementary Data 17). For these targets, we compared the positive or negative 'sign' of the target gene's relationship to KDM5B activity in PURE01 and TCGA-BLCA data. In the second approach, we compared the negative relationship between KDM5B regulon activity and ESTIMATE ImmuneScore for the PURE01 and TCGA-BLCA cohorts, as follows. Given the batch-corrected TCGA-BLCA RNA-Seq data, we used RTN's tni.replace.samples() function to replace the PURE01 expression data with the TCGA-BLCA gene expression data in the PURE01 $n = 82$ cohort's transcriptional regulatory network calculated in the previous section. We then used RTN's tni.gsea2() function to calculate regulon activities and activity status.

To calculate ESTIMATE ImmuneScores for the PURE01 $n = 82$ cohort, we log2-transformed the FPKMs for protein-coding genes and ran ESTIMATE v1.0.13 on the transformed data. For RNA-Seq data for the TCGA-BLCA cohort, we transformed the 20531 TCGA gene names (e.g., 'SLC35E2|728661') into 20501 unique gene symbols, then ran ESTIMATE on the log2-transformed RNA-Seq data.

Finally, we generated scatterplots of KDM5B regulon activity vs. ESTIMATE ImmuneScore, for PURE01 and TCGA-BLCA results, coloring scatterplot dots (each dot represented a cohort sample) by the activated/undefined/repressed activity status of the KDM5B regulon in

each respective cohort. To complement each scatterplot, we generated rank-sorted KDM5B activity profiles for each cohort, using red/gray/blue to represent activated/undefined/repressed KDM5B regulon activity status. We note that these rank-sorted activity/status barplots are a specific representation of the KDM5B regulon activity across a cohort, and the ranges of repressed/undefined/activated samples in a barplot are not expected to match the corresponding ranges of ImmuneScore values in the corresponding scatterplot.

### Inferred interactome of KDM5B's negative regulon targets

Given KDM5B's regulon, we input gene symbols for the 56 negative targets into STRING v11.0 (string-db.org)[33], and generated a 'full' PPI network, using the highest confidence (0.90), and allowing up to 10 first and second shell interacting proteins. We exported a table of Reactome[60] pathways that were enriched in this network, then displayed the statistical significance values of the top six Reactome pathways as a horizontal barplot, using a custom R script.

### Digital spatial profiling (DSP) of proteins

We chose representative CR and NR slides for each luminal subtype, and a representative CR slide for each basal/mesenchymal subtype: S1 (CR = sample 88; NR = sample 12), S2 (CR = sample 37; NR = sample 17), S4 (CR = sample 86; NR = sample 99), S3 (CR = sample 50), and S5 (CR = sample 34). ('Representative' implies manual selection, with no biological or technical replicates.)

Using methods previously described[61], slides were deparaffinized and rehydrated by incubating in CitriSolv (3 × 5 min), 100% ethanol (2 × 5 min), 95% ethanol (2 × 5 min), and deionized water (2 × 5 min). For antigen retrieval, slides in 1X Citrate Buffer (pH 6) were placed into a pressure cooker at high temperature and pressure for 15 min. After releasing the pressure, slides were equilibrated to room temperature for 25 min. Slides were washed (1x TBS-T, 5 × 1 min), blocked with Buffer W (NanoString, Seattle WA) in a humidified chamber for 1 h at room temperature, and stained overnight with a panel of 77 oligo-conjugated detection antibodies at a final concentration of ~0.25 μg/ml for each antibody. The next day, slides were washed (1x TBST-T, 3 × 10 min), postfixed in 4% PFA for 30 min at room temperature, washed (1x TBS-T, 2 × 5 min), and nuclei were stained with 500 nM SYTO 83 (Thermo Fisher S11364) for 15 min at room temperature in a humidity chamber, and rinsed with 1X TBS-T.

Prior to slides being loaded into a GeoMx Digital Spatial Profiler (NanoString, Seattle WA), they were stained with immunofluorescent antibodies to facilitate identification of the tumor and tumor microenvironment components: pan-cytokeratin (clone AE1/AE3, Novus Biologicals, NBP2-33200DL594, 1:400) for epithelial/tumor cells; CD3 (UMAB54, OriGene, UM500048, 1:100) for T cells; and smooth muscle actin (1A4, eBio 53-9760-82, 1:400) for muscle. Once scanned, the digital image for each tissue section was assessed by a pathologist, and regions of interest (ROI) were selected that were representative of the tumor morphology and tumor microenvironment for each sample.

Supplementary Data 20–24 give the NanoString DSP protein data. We assessed a total of 71 proteins from the following DSP modules or panels: Cell Death (10 proteins), Immune Activation Status (8 proteins), Immune Cell Profiling Core (18 proteins, plus three positive and three negative controls, see below), Immune Cell Typing (7 proteins), IO Drug Target (10 proteins), Pan-Tumor (9 proteins), and PI3K-AKT (9 proteins/phosphoproteins). The proteins included three negative controls (Ms IgG1, Ms IgG2a, and Rb IgG) and three positive controls (GAPDH, S6, and histone H3).

For each slide (see above), DSP data were generated for regions-of-interest (ROIs), and each ROI could yield one or more areas-of-interest (AOIs) after color filtering (see micrographs in Fig. 4b). For correlation, PCA, and differential abundance calculations, we used a subset of 49 expressed proteins, as follows. We first removed the one

'muscle' record (ROI = AOI) from the dataset. Then, because we have more confidence in proteins whose abundance distributions were above the distributions of the negative controls, we calculated the mean abundance of the three negative controls across all AOIs and retained only signal (i.e., non-control) proteins whose median abundances were above the median of the negative control average. Taking SMA as a muscle marker, we then removed SMA from all AOIs, leaving 49 of the original 71 signal proteins. We then used protein levels normalized by the signal-to-noise ratio (SNR) (Supplementary Data 21) and only TME AOIs, to do two types of calculations. First, with centered and scaled protein abundances, we used PCAtools v2.2.0 to generate principal component similarity and loading plots. Then, for 13 immune regulatory proteins and immune markers, we generated heatmaps of SNR-normalized protein abundance for TME AOIs for complete responders and non-responders, using pheatmap v1.0.12 and scale = "row" within each group.

We extended this analysis for the subset of proteins whose median levels were greater than the median levels of the most abundant negative control, rabbit (Rb) IgG. For the 21 CR AOIs for S1, S2, and S4 (Fig. 4b), and 42 proteins for the 17 NR AOIs, this retained 35 and 42 proteins, respectively (Supp. Fig. 8a, b). For these two sets of more-highly-abundant proteins, we calculated Spearman correlations separately for CR and NR AOIs (Supp. Fig. 8c, d), highlighted certain heatmap regions by adding a rectangle or triangle, then annotated each protein in each heatmap with 'probe groups' (Supp. Fig. 8e, f).

### RNA sequencing of RT4 cells treated with the KDM5i C70 and an FGFRi

RT4 cells were purchased from ATCC and cultured in McCoy's medium (Thermo Fisher Scientific) with 10% FBS. RT4 cells were treated with DMSO, 5 μM KDM5i KDM5-C70 (XcessBio, M60192-2S) for 72 h with or 5 μM FGFRi Erdafitinib (MedChemExpress, HY-18708) for 48 h. All experiments were done with biological triplicates. RNA was isolated using a Qiagen RNeasy Mini kit, and on-column DNAse digestion was performed to avoid DNA contamination. RNA quality was assessed by a High Sensitivity RNA Tapestation (Agilent Technologies Inc., California, USA) and quantified by a Qubit 2.0 RNA HS assay (Thermo-Fisher, Massachusetts, USA). Paramagnetic beads coupled with oligo d(T)$_{25}$ were combined with total RNA to isolate poly(A) + transcripts following the NEBNext Poly(A) mRNA Magnetic Isolation Module manual (New England BioLabs Inc., Massachusetts, USA). All library construction followed the manufacturer's instructions for the NEBNext Ultra II Non-Directional RNA Library Prep Kit for Illumina (New England BioLabs Inc., Massachusetts, USA). Final library quantity was assessed by a Qubit 2.0 (Thermo Fisher, Massachusetts, USA), and quality was assessed by TapeStation D1000 ScreenTape (Agilent Technologies Inc., California, USA). The final average library constructs and insert sizes were, respectively, approximately 380 bp and 260 bp. Illumina 8-nt dual indices were used for demultiplexing. Libraries were equimolar pooled based on QC values. The pools were sequenced on an Illumina NovaSeq S4 (Illumina, California, USA) as 150-bp PE reads, generating 20 M read pairs per sample. BCL2fastq was used to convert BCL files generated by the sequencer into fastq format, and fastq files were processed using the Ceto pipeline created by Elizabeth Bartom (https://github.com/ebartom/NGSbartom). Briefly, RNA sequencing reads were aligned to the GRCh38 reference genome using STAR v2.7.5. Gene expression was quantified using HTSeq v0.11.1, and differential gene expression profiles were generated using edgeR v3.16.5[51]. HTSeq FPKM values were used for downstream analysis in R v3.6.3. Volcano plots and dotplots were generated using ggplot2 v3.3.2, and gene set enrichment analysis was performed using tmod v0.46.2 CERNO tests and Hallmark gene sets from MSigDB v7.2[19]. A Venn diagram showing the overlap of significantly expressed genes between KDM5i C70 and FGFRi treated cells were manually generated using numbers calculated from the HT-seq/edgeR output files.

Coverage profiles were generated using deepTools and were displayed in the UCSC genome browser.

## Single-sample expression subtype classifier

GLMnet elastic nets are generalized linear models with regularization. While they are normally used in regression contexts, they can be easily adapted to classification tasks using binomial family models, and the R package caret includes this possibility by default. Previous publications in our field had used this type of classifier[62,63]. We also tested a random forest (RF) model and a simpler nearest-centroid (NSC) model, all in 3-fold repeated cross-validation runs. Consistently, the best-tuned elastic net models outperformed both alternative models (Supp. Figure 3d). The elastic net classifier's cross-validation accuracy was 92%. Given that we lack the ground-truth labels for ABACUS subtypes, we are restricted to assessing accuracy with cross-validation metrics.

We used log2-transformed FPKM RNA-Seq data from the $n = 82$ pre-treatment PURE01 samples to generate a single-sample classifier, as follows. We used ANOVA tests to find the genes that discriminated between the five unsupervised subtypes, then selected between 20 and 1500 of the most discriminatory of these genes, and applied the R caret package (v6.0-86, R 4.0.4) to fit four different kinds of models: nearest shrunken centroids (NSC, pamr v1.56.1), elastic net (glmnet v4.1-1), multi-class random forest, and the combination of five one-vs-all random forest models (ranger v0.12.1). We evaluated each model using repeated cross-validation (3-fold, five repeats) and, using levels suggested by the R package dials (v0.0.9), tuned parameters to maximize cross-validation accuracy. For NSC, we tuned thresholds (10 levels); for the elastic net, we tuned $\alpha$ (10 levels) and $\lambda$ (20 levels); and for the random forest, using 'gini' as the splitting rule, we tuned the number of trees (6 levels between 2 and 18) and minimum node size (4 levels between 2 and 15). From this assessment, we chose a 100-gene GLMnet model with parameters $\alpha = 0.1$ and $\lambda = 0.05$, which had an estimated cross-validation accuracy of 88%. The R-based classifier, with installation and usage instructions, is available at https://github.com/csgroen/mibcCPIclass.

We applied the 100-gene classifier to log2-transformed FPKM RNA-Seq data from the ABACUS pre-treatment cohort, predicting probabilities that each of the $n = 84$ samples corresponded to one of the five PURE01 expression subtypes.

For comparison, we trained 100-gene and 500-gene GLMnet and random forest classifiers with VST-transformed[52] PURE01 RNA-Seq data in the same manner as described previously and used these classifiers to predict in VST-transformed ABACUS RNA-Seq data. We found no benefit in using VST normalization for building the single-sample classifier [cross-validation accuracy: log2(FPKM) = 92.1% (83.2–100%, 95% CI), VST = 89.7% (78.8–100%, 95% CI; cross-validation kappa: log2(FPKM) = 86.7% (71.8–100%, 95% CI), VST = 89.7% (78.2–100%, 95% CI)]. Given this, we recommend using log$_2$(FPKMs) with the GLMnet classifier, rather than VST-transformed RNA-Seq data.

## Single-cell RNA sequencing of MIBC tumors

Three fresh bladder tumors from patients undergoing TURBT (transurethral resection of a bladder tumor) were collected under IRB STU00204352. Tissue was cut into small pieces and enzymatically digested to achieve a single-cell suspension. Next, cells were resuspended in 0.04% BSA in PBS solution and loaded into a 10X Genomics Chromium platform for Gel Bead-In Emulsion (GEM) generation and barcoding. Samples were processed using the Chromium Single GEM Single Cell 3′ reagent kit v3.1, following the manufacturer's instructions. Libraries from three 10X channels were pooled and sequenced on one lane of an Illumina HiSeq X as 150-bp paired-end reads. Read sequences were demultiplexed to generate FASTQ files, which were further processed using 10X Genomics' Cell Ranger v4.0.0 pipeline. Reads were aligned to the GRCh38 genome, and GENCODE v32/Ensembl 98 gene counts were quantified using the counts

command in Cell Ranger to generate feature-barcode matrices. Samples were further analyzed in Python3.8 using Scanpy v1.8.2[64] with the following modifications. To reduce false positive readouts from cells with low gene counts, we removed from the analysis cells with fewer than 200 unique features and features that were identified in fewer than 200 cells, as well as cells for which greater than 20% of UMIs mapped to mitochondrial genes and greater than 5% of UMIs mapped to ribosomal genes. Post-filtering, the remaining features (i.e., genes) across 15922 unique cells were scaled and centered to define the relative expression of features across cells. Next, we used the SCSA[65] package to annotate cell types within each cluster. After identifying the main cell lineages (tumor, immune and stromal compartments), we used unsupervised clustering to refine the cell states identified for the epithelial cell lineage, identifying nine sub-clusters. We generated a luminal gene expression score using scanpy.tl.score() genes and 16 luminal markers: CYP2J2, ERBB2, ERBB3, FGFR3, FOXA1, GATA3, GPX2, KRT18, KRT19, KRT20, KRT7, KRT8, PPARG, XBP1, UPK1A, and UPK2, which are associated with the luminal subtype in muscle invasive bladder cancer[14]. Next, we used the top ten principal components as input to Louvain graph-based clustering, with a resolution parameter of 0.4. All gene expression and clustering results were visualized using the UMAP function in Scanpy. Regulon analysis was conducted using pyScenic v0.11.2[66]. For the epithelial cell sub-cluster, an AnnData object was created and used as an input to the pyScenic pipeline. The count matrix was used as an input to pyScenic and processed with default parameters. Next, regulon prediction was performed using cisTarget with default parameters and three hg38 .feather ranking databases. pyScenic runs using default settings detect only activating targets and not repressive targets. We modified this behavior by using the –all-modules option in the -ctx command to generate a list of all enriched motifs for activating and repressive targets, then used aucell to quantify the activity of each gene signature across single cells in the epithelial cell sub-cluster. Finally, results from pyScenic were integrated into a Scanpy AnnData object. For marker genes in each epithelial cell sub-cluster, we show the top five genes, ranked by adjusted $p$-value, requiring a minimum absolute value log2-fold change of 1.5.

## ATAC-seq sample and data processing

Replicate samples from RT4 cells treated with DMSO or $5\,\mu$M KDM5i-C70 for 3 days were processed for chromatin profiling by ATAC-sequencing using a previously described protocol[67]. Briefly, 50,000 cells were washed in cold PBS and lysed. Transposition was performed at 37 °C for 30 min using Illumina Tagment DNA Enzyme and Buffer kit (Illumina, 20034197). DNA was isolated using the Qiagen MinElute Reaction Cleanup kit and was amplified for five cycles. The final number of additional PCR cycles was evaluated by real-time PCR. The final library was purified using double-sided AMPure XP beads purification to remove primer dimers and >1000 bp fragments. Library quality was assessed by running 1 $\mu$l of library on a High Sensitivity DNA Bioanalysis chip (Agilent, Santa Clara, CA). The DNA concentration was quantified by QuBit. Samples were sequenced on an Illumina HiSeq as 150-bp paired-end reads, with an average of 92.5 M paired-end reads generated per sample.

For the raw sequencing data, quality metrics were generated using FastQC v0.11.5. Cutadapt v3.3 was used to trim adapter sequences. Paired-end-reads were aligned to the GRCh38 reference human genome using Bowtie2 v2.4.1. Non-uniquely mapped reads were removed. Using a previously described method that accounts for Tn5 transposase binding during analysis[68], all positive-strand reads were shifted 4 bp downstream, and all negative-strand reads were shifted 5 bp upstream using the alignmentSieve tool in deepTools v3.1.1. Shifted reads were pooled by condition, and peaks were identified using MACS2 v2.1.0. Profile plots and heatmaps for the ATAC-seq signal profiles around transcription start sites (TSS) were generated using plotHeatmap and plotProfile tools in deepTools v3.1.1 by pooling

DESeq2 v1.30.0 normalized read counts for each condition. ATAC-seq coverage tracks were generated using deepTools and were visualized using the UCSC genome browser.

## CMap/LINCS analysis

To identify potential drug targets for S6 and S7 subtypes, and to compare PURE01 subtypes S1-5 and ABACUS predicted subtypes S1-5, we utilized the Connectivity Map (CMap) LINCS gene expression resource[21,69] to identify perturbed gene expression signatures that were similar (vs dissimilar) to the gene expression signatures of each of these subtypes. For each subtype, we ranked coding genes by the signal-to-noise ratio (SNR) and submitted gene symbols for the top 150 high-SNR and 150 low-SNR genes to the expression Query tool (https://clue.io) using the Touchstone v1 L1000 dataset[70]. Results from each query assigned a connectivity score between −100 to 100 to each CMap perturbagen. A positive score indicated that the query signature was similar to a perturbagen signature (i.e., genes that were upregulated in the query were upregulated in reference data for treatment with a perturbagen), while a negative score indicated the opposite (i.e., upregulated query genes were downregulated in reference perturbagen data). We ranked perturbagens based on the connectivity scores, and generated score-rank plots with Graphpad Prism, then manually generated per-subtype heatmaps of selected perturbagens.

## Statistics and reproducibility

We did calculations in R, typically with v4.x. Statistical tests were typically done with R's two-sided fisher.test(), kruskal.test(), and chisq.test() functions. *P*-values reported were corrected for multiple hypothesis testing using Benjamini–Hochberg (FDR) or Bonferroni approaches, where noted. Boxplots were generated in R using default settings. Each box spans the 25th to 75th percentile range in the data (i.e., the interquartile range, IQR) and shows a horizontal line at the median value. Whiskers extend 1.5 times the IQR from the box, and dots, where shown, show all data values, including minima (minimum values) and maxima (maximum values). Violin plots (Fig. 5d) were generated with Graphpad Prism; as in the boxplots, dots show all data values. Results for each cohort (PURE01, ABACUS, and IMvigor010) were generated from RNA-Seq data for the cohort, with no biological or technical replicates. Similarly, 'representative' micrographs in Figs. 4b, 5b, and in Supplementary Fig. 2f, were chosen manually; they do not include technical or biological replicates.

## Reporting summary

Further information on research design is available in the Nature Portfolio Reporting Summary linked to this article.

## Data availability

The raw sequencing data have been deposited in the EGA database (https://ega-archive.org/studies) under the accession code EGAS00001005549: (a) for the PURE01 cohort, bulk RNA-Seq data derived from human samples; (b) scRNA-Seq data for human muscle-invasive tumor samples; (c) bulk RNA-Seq data for the human RT4 cell line treated with either the KDM5Bi C70, an FGFRi, or a DMSO control; and (d) bulk ATAC-Seq data for RT4 cells treated with the KDM5B inhibitor C70, or with a DMSO control. Access can be granted by contacting the corresponding author (joshua.meeks@northwestern.edu) with responses addressed within 14 working days. Data will be available for at least 24 months from publication. Human subject data are available under restricted access to protect patient information. For the ABACUS cohort, clinical and bulk RNA-Seq data can be requested from the EGAS00001004445 Data Access Committee (DAC, bci-cecmqa@qmul.ac.uk); data for the IMvigor010 cohort (EGAS00001004997) can be requested at devsci-dac-d@gene.com. Source data to generate most figure panels are publicly available at Zenodo (https://doi.org/10.5281/zenodo.7750550).

## Code availability

Original R and Python code are publicly available on Zenodo (https://doi.org/10.5281/zenodo.7750550).

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

## Acknowledgements

JJM is supported by grants from VHA BX003692-01, DoD (W81XWH-18-0257 and W81XWH-19-1-0477), AACR-Bayer Innovation and Discovery Grant, the Polsky Urologic Cancer Institute Research Award. AN is supported by grants/research support by Merck, Ipsen, Immunomedics, and Astra Zeneca. The authors are grateful to Sanjeev Mariathasan and Genentech for facilitating access to the IMvigor010 cohort.

## Author contributions

Conceptualization: J.J.M., A.N., and A.G.R. Methodology: A.G.R., K.M., L.F.C., L.A.F., Y.Y., M.A.A.C., C.S.G., V.I.N., V.O.T., B.C., and T.P. Software: A.G.R., K.M., K.A.M., and M.A.A.C. Validation: T.P., K.M., A.G.R., and M.A.A.C. Formal analysis: A.G.R., K.M., K.A.M., and M.A.A.C. Investigation: A.G.R., K.M., L.F.D., K.A.M., L.A.F., Y.Y., M.A.A.C., C.S.G., A.D.R., V.I.N., V.O.T., B.C., D.R., L.M., F.M., T.P., A.N., and J.J.M. Resources: D.R., L.M., F.M., and T.P. Data curation: A.G.R., K.M., L.F.C., K.A.M., L.A.F., M.A.A.C., C.S.G., A.D.R., V.I.N., V.O.T., L.M., D.R., T.P., A.N., and J.J.M. Writing—original draft: A.G.R., K.M., L.F.C., M.A.A.C., C.S.F., and J.J.M. Writing—review & editing: A.G.R., K.M., L.F.C., K.A.M., L.A.F., Y.Y., M.A.A.C., C.S.G., A.D.R., V.I.N., V.O.T., B.C., D.R., L.M., F.M., T.P., A.N., and J.J.M. Visualization, A.G.R., K.M., L.F.C., K.A.M., L.A.F., Y.Y., M.A.A.C., C.S.G., A.D.R., V.I.N., V.O.T., B.C., D.R., and J.J.M. Supervision: A.G.R., A.D.R., F.M., A.N., T.P., and J.J.M. Project administration: L.M., D.R. Funding acquisition: A.N., T.P., and J.J.M.

## Competing interests

J.J.M. participated in advisory boards for AstraZeneca, Astellas/Seagen, BMS, Janssen, Prokarium, Pfizer, Merck, and UroGen. A.N. is a consultant for Merck, Astra Zeneca, Janssen, Incyte, Roche, Rainier Therapeutics, Clovis Oncology, Bayer, Astellas/Seattle Genetics, Ferring, and Immunomedics. Travel expenses/Honoraria: Roche, Merck, Astra Zeneca, and Janssen. L.M.: Speaker compensation: Merck; Travel expenses and accommodation: Janssen; Research funding: AstraZeneca. AdR is a member of the scientific advisory board of Qlucore and cofounder of Minos Biosciences. A.G.R., K.M., L.C., K.M., L.A., Y.Y., M.C., CG., V.N., V.T., B.C., F.M., and T.P. have no disclosures.
