## [Peer Review File · Nature Communications]

Expression-Based Subtypes Define Pathologic Response to Neoadjuvant Immune-Checkpoint Inhibitors in Muscle-Invasive Bladder CancerREVIEWER COMMENTS

Reviewer #1 (Remarks to the Author): expert in bladder cancer

Authors identified five subtypes that have unique genomic and transcriptomic cancer cell programs, and distinct tumor microenvironments, which are associated with response to neoadjuvant pembrolizumab in bladder cancer. Authors also discovered that FGFR3 mutation and KDM5B regulon activity were associated with the resistance to CPI in one (S1) of subtypes. These results are important as there is an unmet clinical need for patient selection tool for CPI. However, there are some concerns that need to be addressed before publication.

1. Why did the authors choose $K=5$ in Fig 1? Provide silhouette score analysis and CDF plot to define $k=5$ solution.
2. Provide p-values for association between responses and subtypes in Fig 1b.
3. The survival outcome of S4 may lead to the significance of the p-value in Fig 1c. If the authors claim that S1 has the poor survival outcome as S4, then a direct comparison of S1 with S2 and S3 should be made. Also explain why there is a strong discrepancy between pathological response and overall survival (OS) in the S5 subtype.
4. Provides survival data from the Abacus trial in Fig 2.
5. In bladder cancer, there are already several molecular subtype classifiers (TCGA, Consensus, Lund, etc.). The authors demonstrated the similarity of this new five subtype classifier to the existing subtype classifier (Fig. 1e using Fisher's exact test, Boferroni modification). So, what is the significance of this new five subtype classifier since the clinical significance of the new classifier cannot be validated in an independent dataset (Abacus trial)?
6. On page 8, the authors used 10% of mutations as a cutoff to find associations of mutations with hallmark gene sets. What is the rationale to use a 10% cutoff?
7. Mutations in KRAS and FGFR3 appear to be found at a higher frequency in S1. Is there a 'direct' association between the responses and mutations of KRAS and FGFR3 in S1?
8. Perform the same mutation analysis using ABACUS data to show this new classifier could find (at least) the same biology.
9. Fig 4a and Fig S4 showed that S5 and S3 were enriched with both stroma and immune signatures. In S5 (StromR), not only stroma but also immune scores were highest in both ESTIMATE and MCP-counter. Since, both S5 and S3 showed good survival outcomes, what is the role and relationship between immune cells and stroma cells here?
10. Although the authors claim that S4 is a luminal subtype, S4 appears to be a mixture of neuronal, basal and luminal tumors. If the authors would like to identify the unique proteomic profiles in the TME that could contribute to CPI outcomes in luminal tumors, S4 should be removed in PCA analysis in Figure 4c or change the description of analysis.
11. On p11, the authors should provide the exact number of sample distribution in subtypes such as 'most of the resistant tumors originated from S1 and were classified as S6 after pembrolizumab (pre: $n=x$ in S1; post $n=y$ in S6, $n=z$ in S7, etc)
12. Provide silhouette score analysis and CDF plot to define $k=7$ solution in Fig 5a. Also, subtype membership of some tumors was switched before ($k=5$) and after combining post tumors ($k=7$). How many tumors were switched?
13. In addition to H&E staining (Fig 5b), to demonstrate 'scar-like stromal' features' and 'immune infiltrate', please provide molecular marker gene expression data and IHC data in S6 and S7. Fig

5b is a representative image of each subtype. Are there any statistics showing how many tumors in S6 have scar-like features and how many tumors have immune-invasive features?

14. If S6 and S7 are distinct from S1-5 molecularly to form additional cluster, the authors should provide the molecular difference between S1-5 and S6-7 including in fig 5e and 5f.

15. In addition to Fig 5g, additional figures showing immune scores/stromal scores/tumor purity stratified by S6 and S7 should be generated as the authors claim that there are differences in stromal characteristics and immune invasion between S6 and S7.

16. On p13, authors described that "the KDM5B regulon was repressed in subtypes with high immunoscores (S3, S5)" - This is incorrect. The regulon activity is relatively very low only in S5, not S3. The KDM5B regulon activity of S3 is similar to that of S2 or S4 in Fig S8b. Also, the negative association between KDM5B regulon activity and immune scores should be validated in independent datasets including TCGA data.

Reviewer #3 (Remarks to the Author): expertise in digital spatial profiling and transcriptomics

The paper implements omics-technologies, including digital spatial profiling and DNA-seq, to profile and characterize molecular signatures of the tumor microenvironment for pre- and post-treatment (pembrolizumab immunotherapy). The paper characterized the five subtypes of muscle-invasive bladder cancer (MIBC), developed a classifier for distinguishing a tumor sample subtype, and found that histone demethylase KDM5B as a repressor of tumor immune signaling pathway plays an important role in the treatment resistance mechanisms. This finding can be used for stratifying patients and giving an alternative treatment for the poor therapy responsiveness. However, the paper has some limitations on computational methods. First, the model of classifying bladder to tumor subtype needs additional data in evaluation. Second, the paper should unify some computational tools (e.g., different versions of the same tool) to enhance reproducibility. Third, some results need to be clearly presented to demonstrate biological insights. Overall, the paper is a good resource paper (using multiple omics data) for the community and provides insights into the clinic side, but it still has room to be enhanced in terms of in-depth analysis.

Major comments:

1. The reviewer cannot access data and code scripts to reproduce and validate the findings.

Therefore, the following comments are mainly based on the manuscript, figures, and tables. The reviewer cannot evaluate the scientific rigor and reproducibility without data and analysis code.

2. In the introduction part, "However, biomarker analysis in ABACUS..... TMB were associated with response in ABACUS." When read the first sentence, the reviewer expects the author to explain why the two clinical trials show a molecular difference rather than "somewhat discordant." Secondly, what is the definition of "activated CD8+ gene expression?" How to define "activated"? It is higher CD8+ gene expression or other a group of signature genes associated with cytotoxic T cells activity.

3. Regarding the histology representative tumor (Figure S2c), the author should give a systematic evaluation for whether all samples displayed a similar histological feature in five categories, respectively.

4. In the Method section (single-sample expression subtype classifier), first, please introduce the rationale why choose GLMnet as a classifier basis. Second, could the author provide more results regarding the model training regarding parameter tuning because there are lots of parameters (or hyperparameters) to determine the final model? A table showing parameters' combination and results (i.e., show precision, recall, or accuracy) will be helpful. Third, the authors used two normalization methods as input to train the classifier based on GLMnet. What is the rationale here to compare the two normalization methods? These two normalization methods push me to think about the previous GSEA analysis, whether it has two results?

5. In the results part (assessing CPI-MIBC subtypes in an independent cohort), the reviewer likes validating the model using the secondary dataset and digging out more biological insights. However, before going to this section, the author should show the power of the classifier, as suggested in comment #3. Either a figure or a summary table will be helpful.

6. In the result section (somatic mutations of CPI-MIBC subtype), why using fgsea to calculate

enrichment score. In the previous analysis in the first section, the author used CERNO method to show the enrichment results, but the method was changed in this section without any rationale. The author should elucidate why they changed the method. Because two different computational methods mean two different algorithms and assumptions, resulting in poorly reproducible results (ref: PMID: 32695141). In addition, it may be a minor comment, but in terms of professionalism, the software version is not unified, which is not usually seen in one paper. For example, R shows two versions in one study, including 3.6 and 4.1. DESeq2 also shows two versions, including 1.30 and 1.26.

7. Regarding the result section (Subtype differences in the tumor microenvironment by bulk and spatial analysis), using deconvolution for bulk-RNAseq data could distinguish the dominant cell types in TME, but the cell types in deconvolution results still show a low granularity. For example, cell types include CD8 T cells, B cells, overall T cells, etc. This may be another reason to perform DSP spatial proteomics experiments (71 proteins) and analyses to show the real cells' colocalization and higher granularity of cell composition. However, the spatial analysis is not eye-catching (the take-home message is not clear to me), and where do the other proteins go (Figure 4d shows 13 proteins)?

8. For the last result section (FGFR3 and KDM5B are potential LumE/S1 subtype targets), it used bulk-seq to identify the potential regulator and then uses scRNA-seq & ATAC-seq to validate KDM5B's role. The author could try to use SCENIC (python package) or IRIS3 (webservice) to further investigate the regulon network at the single-cell level with an advanced computational framework and validate the findings.

Minor comments:

9. Please add more references in the introduction part to strengthen the background section. For example, the author adds references for supporting the "neoadjuvant paradigm" in the first paragraph.

10. In the method section (Consensus expression subtypes), the author described "Spearman and Pearson distance" to do clustering. Did the author used Spearman and Pearson dissimilarity as distance or used Spearman and Pearson coefficient as distance.

11. Please assume the diverse background readers, introduce terminology, and keep reading smoothly. The paper has some terminologies that need more clarifications. For example, what is CPS in figure 1a, and the reviewer cannot find the explanation for this term?

12. In the Method section (Gene set enrichment analysis), paragraph "For PURE01 n=82, for each expression subtype, vs all other samples, we ..." need to be reorganized for grammar issues and unrelated words.

13. In the result section (somatic mutations of CPI-MIBC subtype), how many tumor samples were used for evaluating somatic mutations and copy number variation?

Reviewer #4 (Remarks to the Author): expertise in checkpoint immunotherapy response

The authors submit a comprehensive characterization of patient samples from two neoadjuvant trials of checkpoint inhibitors in bladder cancer. The overall goal of the study (p14) is to dissect the molecular heterogeneity of MIBCs to further define mechanisms of resistance. The manuscript is clear, structured logically, and well written. The figures are all relevant and well designed. There are several key strengths to this work that are important to highlight.

1. Direct clinical relevance. Immunotherapy is an effective therapeutic for metastatic urothelial cancer. It is being tested in multiple phase III trials in the neoadjuvant setting. Biomarkers predictive of response to IO are lacking to date and could be immediately useful clinically.
2. Access to prospectively collected tissue samples. The authors analyzed pretreatment tissue from two therapeutic clinical trials of checkpoint inhibitors in localized bladder cancer: PURE-01 and ABACUS.
3. Access to high quality clinical data. By leveraging clinical trial tissue samples, each was well annotated with respect to therapeutic dosing and clinical and pathologic outcomes.
4. Extensive experiments resulting in a comprehensive "multi-omics"
5. The availability of a discovery and validation set: With access to tissue samples and annotation from two prospective trials, the authors are able to use PURE01 for discovery and ABACUS for validation.

Comments:

General:

This manuscript is extensive and far reaching, using samples described above, defining and then looking deeply into 5 new subtypes, extending them to a pre/post treatment analysis and including experiments done on a cell line linked to one of the subtypes. However, the extent of the analyses included in the manuscript falls prey to a trend best described by Bill Kaelin in an editorial for this journal: Nature volume 545, page 387 (2017). The work described here is careful and sound, but suffers from being too broad with many tangential inquiries that individually lack depth. The core set of investigations, once the analyses are revised to describe a direct link between biology and therapeutic response, will be of great value to the bladder cancer community. Furthermore, the tangential analyses could be papers in their own right, allowing distribution of credit to those doing the bulk of the work in an academic system that values first and last authorship. The comments below suggest ways to overcome this and produce a strong manuscript that can be accepted for publication without delay. It is not in our collective best interest to delay reporting these findings, nor public deposit of the tissue based data which will be a critical resource for further discovery in bladder cancer.

Major:

The authors collected RNA-seq data from 30 complete responders and 33 non-responders (omitting the middle clinical category of stable/partial responders). Unsupervised consensus clustering was then used to define five transcriptomic subtypes. These subtypes are applied to the ABACUS cohort as validation. The authors then carry these subtypes throughout the manuscript, correlating all other results to the subtypes (GSEA, PDL1, cMAP, histology). They are also compared to existing subtypes (Lund, TCGA, consMIBC, MDA).

1. It is not clear that the subtypes validate in ABACUS. A statistical comparison to validate these new subtypes should be provided. This is particularly important since such small numbers of patient/samples were used to derive the subtypes in the first place.
2. It is not clear that the subtypes align with the 4 subtype classifications previously described. In particular, S4 S5 and S3 appear heterogeneous.
3. It is not clear why the subtypes in and of themselves are of value. 4 others have already been defined, why not align these samples to those?
4. It is not clear that the subtypes are reproducible within the same patient. Testing more than one TURBT chip from the same patient's procedure would help clarify this question.
5. The PURE01 S2 and S3 subtypes are enriched for response, but this enrichment is not validated in ABACUS, nor is the correlation described statistically.

The overall goal of the paper is to provide insight into biology that can predict response and/or resistance. However, the -omics analyses are correlated to the new subtypes, not to response or resistance directly. For instance, to correlate PDL1 the authors state that the subtypes with highest responses have a higher proportion of PDL1 – notably this is true for the PURE01, not ABACUS. Therefore, rather than answer the 4 questions above which would require more samples and analyses for this manuscript, this reviewer recommends describing the new subtypes, acknowledging that they don't replicate in the ABACUS validation set, and then using the data already generated from the experiments done to correlate the following investigations directly with response and resistance using all samples from PURE01 (not just the extremes of the response spectrum).

1. PDL1 (ideally using CPS for PURE01 and sp142 IC score for ABACUS)
2. GSEA
3. CMap
4. Somatic mutations and copy number profiles
5. Bulk and spatial analysis using ImmuneScores by ESTIMATE analysis
6. The expression of immune cell markers and regulatory proteins

Any significant findings can then be validated using the ABACUS patient/samples and outcome data.

The analysis investigating B cell repertoires and immunoglobulin signatures and correlating them to response is well done, but would be strengthened by also including those with SD among the non responders for greater power. It was nice to see statistics applied here and a significant difference noted.

The paper in general suffers from over reliance on vague descriptive language to describe

correlations and underuse of statistical comparisons. For instance, statements such as "S4 tumors had more mitotic features" "overall mutation frequencies were comparable to", "results ...were largely consistent" "were relatively concordant" and "were enriched for" – are not specific enough to draw conclusions around relationships. Statistical support should be provided where correlations are claimed.

DSP analysis was done on just 5 samples. Here descriptive results are all that can be used in the absence of a more robust sample set. The authors try to draw conclusions around the largely concordant findings in the responders, and the discordant findings in the non-responders, and then further describe responders and non-responders within subtypes S1, S2 and S4. However, with only 5 samples these aren't meaningful. This analysis should be done with more depth (more samples), omitted from this paper and published separately so as not to delay publication of the key findings.

Pretreatment tumors were compared with post treatment tumors by the subsets defined in this paper. Most of the post treatment samples didn't fit one of the 5 predefined subtypes, but landed in one of two new subtypes S6 and S7. Samples from these two post treatment (resistant) subsets are then re-analyzed by GSEA and CMap. While interesting, more robust conclusions could be drawn from these analyses were they applied to pre/post comparison from samples in ABACUS and as a control pre/post from chemo treated and non-treated samples to determine if the subtype plasticity is artifact or real. Because doing this would require more samples and experiments, this reviewer would recommend removing the pre/post analysis from this manuscript, expanding the analyses to more samples including controls, and submitting those results separately for publication as they tell an important but not directly related story.

The discussion of FGFR2 and KDM5B as targets for S1 tumors is interesting but again puts a close focus on these new subtypes defined using few samples and not statistically validated. I am not sure focus on the S1 subset is of high enough relevance to include in this manuscript. If included, please clarify how many of the S1 tumors analyzed were from PURE01 and how many from ABACUS.

The cell line work seems tangential to the central hypothesis of the paper. It is not clear how many cell lines were used or how they are "S1-like." Erdafitinib is already in clinical use in a biomarker defined subset of patients. It's not clear how KDM5 inhibition would be therapeutic and prior experience with similar agents in bladder cancer clinical trials is not discussed. Rather than go back to flesh out and strengthen this analysis which does not relate to the central hypothesis, nor leverage the PURE01 or ABACUS samples, this reviewer would recommend omitting it here and publishing it separately.

Minor:

Page 5, last 3 lines. Two year RFS numbers seem incorrect – the authors probably meant the. The proportions seem to describe rate of recurrence at 2 years not proportion recurrence free at 2 years.

The acronym GSEA is not defined in the manuscript.

RESPONSE TO REVIEWERS

Reviewer #1 (Remarks to the Author): expert in bladder cancer

Authors identified five subtypes that have unique genomic and transcriptomic cancer cell programs, and distinct tumor microenvironments, which are associated with response to neoadjuvant pembrolizumab in bladder cancer. Authors also discovered that FGFR3 mutation and KDM5B regulon activity were associated with the resistance to CPI in one (S1) of subtypes. These results are important as there is an unmet clinical need for patient selection tool for CPI. However, there are some concerns that need to be addressed before publication.

We appreciate the reviewer's evaluation of our manuscript and thank them for their comments.

R1.1. Why did the authors choose K=5 in Fig 1? Provide silhouette score analysis and CDF plot to define k=5 solution.

We agree with the reviewer that the number of subtypes (5) is a central factor in our manuscript. For the n=82 PURE01 pre-treatment cohort, we used both quantitative and qualitative metrics to choose a five-cluster solution. We prefer consensus clustering, because it can offer information that is helpful in choosing an informative clustering solution. We used R's ConsensusClusteringPlus (CCP). We assessed the results from a number of CCP runs, comparing Pearson, Spearman, and Euclidean distances, and hierarchical, PAM, and k-means clustering. While we agree that CCP's CDF plots can be informative, we typically consider CDF, delta and tracking plots, over a range of coarser-grained and finer-grained clustering solutions (for the work reported here, from two to eight clusters). We also consider blue-white consensus membership heatmaps, and their dendrograms, *relative to* both the number of consensus repeats ('reps' parameter), and the intensity of gene ('pFeature') and sample ('pltem') subsampling. Consistent with Aine *et al* 2015, we consider this information in the context of the research questions that we are addressing, understanding that the context may indicate that one of several reasonable clustering solutions will be more informative. Using this approach, we chose a five-cluster solution.

Silhouette width profiles for a clustering solution are typically calculated from input gene expression data (Choi *et al* 2014). So that we could compare silhouette widths for our clustering solutions to those in published work, we have also done this expression-based calculation for PURE01 n=82 and n=113 cohorts, as follows.

Method: For the log₁₀-transformed and median-centered FPKMs for 1950 high-variance coding genes, which were the CCP input for consensus clustering, we calculated a distance matrix from 1 minus a Spearman correlation coefficient. We then input the cluster calls and this distance matrix into the silhouette() function in the cluster v2.1.2 package. This generated a standard graphic that showed clusters in sorted numeric order, and sorted samples within each cluster by descending silhouette width, reporting average silhouette widths for each subtype, and for the overall cohort.

```
ds <- as.dist(1 - cor( log10.median_centre, method = "spearman"))
si.1 <- silhouette( pure01.subtypes$cluster, ds )
```

To respond to the reviewer's request, we have added PURE01 CDF plots to **Supplementary Fig. 2**, as panels **b** and **d**. And, in **Fig. R1.1**, below, we show blue-white consensus membership heatmaps with CDF plots, and silhouette widths calculated from the clustering input gene expression data, for the five-cluster n=82 solution and the seven-cluster n=113 solution. A CDF plot reports the empirical distribution functions of consensus membership values for the range of solutions requested (here, two to eight clusters); CCP also reports these values as heatmaps. An alternative summary 'delta' plot shows the changes in areas under successive CDF curves (**Fig. R1.1g-i**).

We found that our silhouette width results were consistent with publications in our field. Further, we found that authors of recent bladder cancer publications that report subtypes appear to place progressively less

emphasis on silhouette results. This change is consistent with the multiparameter, contextual approach that we describe above, which we used to select informative clustering solutions in the work reported here.

Specifically, we compared our average per-subtype silhouette widths to widths reported in two recent, large-cohort bladder cancer publications. The first publication, Hedegaard et al 2016 showed 'delta' plots, rather than CDF plots, and reported per-cluster average subtype widths in its supplemental information. Their average cluster silhouette widths were comparable to average widths for most of our n=82 and n=113 subtypes (**Fig. R1.1**). We also noted that that publication showed silhouette widths in main results only as a simple positive/negative-width covariate track (their Fig. 1). The second publication, Linskrog et al 2021, showed no CDF or delta plots, and again showed main-figure silhouette width results only as a positive/negative width covariate track (their Fig. 1e). That their description of consensus cluster membership values is consistent with our understanding of these values: their Fig. 1 legend states: "...pairwise values range from 0 (samples never cluster together; white) to 1 (samples always cluster together; dark blue)".

We compared our results to a third publication. While per-subtype silhouette widths were reported as box-whisker plots in Seiler et al. 2017 (their Fig. S3E), we find nothing in that publication that describes how the silhouette widths were calculated. Seiler et al. 2017 refers to Choi et al. 2014, whose Figure S1D reports a per-subtype silhouette width profile, which we show as **Fig. R1.1k**. From this graphic we can estimate their per-subtype mean scores to be approximately 0.1 to 0.2, i.e. comparable to what Hedegaard et al. 2016 reports, and comparable to what we show for PURE01 in **Fig. R1.1c,f**. While Choi's methods section includes a 'Silhouette score analyses' subsection, which gives a distance-based equation, it gives no details of the how silhouette widths were calculated.

Aine M, Eriksson P, Liedberg F, Sjö Dahl G, Höglund M. Biological determinants of bladder cancer gene expression subtypes. *Sci Rep.* 2015 Jun 8;5:10957. doi: 10.1038/srep10957. PMID: 26051783.

Choi W, Porten S, Kim S, Willis D, Plimack ER, et al. Identification of distinct basal and luminal subtypes of muscle-invasive bladder cancer with different sensitivities to frontline chemotherapy. *Cancer Cell.* 2014 Feb 10;25(2):152-65. PMID: 24525232.

Hedegaard J, Lamy P, Nordentoft I, Algaba F, Høyer S, et al. Comprehensive Transcriptional Analysis of Early-Stage Urothelial Carcinoma. *Cancer Cell.* 2016 Jul 11;30(1):27-42. PMID: 27321955.

Linskrog SV, Prip F, Lamy P, Taber A, Groeneveld CS, et al. An integrated multi-omics analysis identifies prognostic molecular subtypes of non-muscle-invasive bladder cancer. *Nat Commun.* 2021 Apr 16;12(1):2301. PMID: 33863885.

Seiler R, Ashab HAD, Erho N, van Rhijn BWG, Winters B, et al. Impact of Molecular Subtypes in Muscle-invasive Bladder Cancer on Predicting Response and Survival after Neoadjuvant Chemotherapy. *Eur Urol.* 2017 Oct;72(4):544-554. PMID: 28390739.

Figure R1.1. Consensus clustering solutions for PURE01: a five-cluster solution for $n=82$, and a seven-cluster solution for $n=113$. **a,d**) Consensus membership heatmaps. In **(d)**, a covariate track indicates pre-treatment and post-treatment samples. **b,e**) CDF plots of clustering solutions with from two to eight clusters; triangles indicate the five- and seven-cluster solutions used. A CDF plot reports the empirical distribution functions of consensus membership values; these values are shown as heatmaps in **(a)** and **(d)**. An alternative summary 'delta' plot **(g,h,i)** shows the changes in areas under successive CDF curves. **c,f**) Silhouette width profiles and per-subtype mean silhouette widths for the clustering solutions in **(a)** to **(e)**. **g**) Hedegaard et al. 2016 reported a three-cluster consensus solution for 460 early-stage urothelial carcinomas. **h**) Consensus clustering results for 4000 genes, from Figure S2 of Hedegaard et al 2016. **i**) Consensus clustering results for genes with the highest median-absolute deviation (MAD) ranks, from Figure S3 of Hedegaard et al 2016. **j**) Lindskrog et al 2021 reported a consensus membership heatmap for a four-cluster solution, and a Sankey plot that comparing Hedegaard et al's 2016 three-cluster solution to a 2021 four-cluster solution. **k**) Per-subtype silhouette 'score' profiles for three MDA subtypes, from Figure S1D in Choi et al 2014.

R1.2. Provide p-values for association between responses and subtypes in Fig 1b.

We assessed the association between pathologic response and **Fig. 1b** subtypes in three ways (**Fig. R1.2**).

First, we assessed whether pathologic response was significantly associated with subtypes. Subtypes were significantly associated with CR, PR, and NR, $p = 0.0059$, and with CR and NR, $p = 0.0011$ (**Fig. R1.2a**, uncorrected p values; CR, PR and NR are complete response, partial response, and nonresponse). These associations would remain significant if we applied a Bonferroni x2 correction.

Second, we assessed whether pathologic responses were significantly different for each individual subtype, compared to the overall cohort (**Fig. R1.2b**). We used two-sided Fisher exact test to compare response (CR, PR, and NR) in each subtype, comparing outcomes in each subtype to the overall $n=82$ cohort. Overall, the responses were significantly different ($p = 0.0059$; and with CR and NR, $p = 0.0011$).

Finally, we used Fisher's exact tests and a Bonferroni correction to assess whether CR/PR/NR pathologic responses were statistically different between pairs of subtypes (**Fig. R1.2c,d**). Responses in S1 were statistically different from those in S2 and S3 ($p_{adj} < 0.05$, bold p values), before and after correction, and no other subtype pair had statistically different responses.

Figure R1.2. Association of responses and subtypes in **Fig. 1b**. **a**) P values for Fisher exact tests (alternative hypothesis: two-sided) of CR/PR/NR (above) and CR/NR (below) vs. subtypes S1 to S5. **b**) P values for Fisher exact tests (alternative hypothesis: two-sided) of CR/PR/NR (or CR/NR) sample counts per subtype, against the overall cohort counts ($n=82$, CR=30, PR=19, NR=33). Numbers in curved parentheses are Bonferroni-corrected p values. **c**) P values for Fisher exact tests (alternative hypothesis: two-sided) of CR/PR/NR sample counts per subtype pair. **d**) As **(c)**, but showing Bonferroni-corrected (x10) p values.

R1.3 The survival outcome of S4 may lead to the significance of the p-value in Fig 1c. If the authors claim that S1 has the poor survival outcome as S4, then a direct comparison of S1 with S2 and S3 should be made. Also explain why there is a strong discrepancy between pathological response and overall survival (OS) in the S5 subtype.

The Reviewer is correct: the significant p value ($p = 0.026$) for the Kaplan-Meier (KM) plot in **Fig. 1c** is largely due to the rapid recurrence events in subtype S4. As requested, we used KM analyses to test S1 vs S2, S1 vs S3, and S1 vs S2+S3, censoring time-to-event data to 24 months in each case. Before correcting for multiple hypothesis testing, log-rank p-values were respectively $p=0.14$, $p=0.068$, and $p=0.027$, so only the third test was statistically significant ($p < 0.05$). After Bonferroni correction, only the third comparison (S1 vs S2+S3) remained significant ($p_{adj} < 0.10$).

“Most of the recurrences by two years was attributed to S4 (38%). While 27% of S1 recurred, the best response rates had low two-year rates of recurrence of 7% for S2 and 5% for S3.”

Figure R1.3. Kaplan-Meier curves for subtypes and subsets of subtypes. Log-rank p values are uncorrected for multiple hypothesis testing.

The Reviewer notes that survival and pathologic response differ for S5 (**Figs. 1b,c**). S5 tumors appear morphologically immune-excluded, which may prevent immune infiltration and tumor killing after pembrolizumab treatment. In the manuscript (lines 399-401, “S5 tumors had...”), we have attempted to hypothesize this survival vs pathologic response discrepancy for S5.

R1.4. Provides survival data from the Abacus trial in Fig 2.

We agree that survival data for the ABACUS trial would be quite helpful. However, we have spoken to the lead ABACUS author (TP), who is a co-author for the current manuscript, and he is unable to provide survival data.

R1.5. In bladder cancer, there are already several molecular subtype classifiers (TCGA, Consensus, Lund, etc.). The authors demonstrated the similarity of this new five subtype classifier to the existing subtype classifier (Fig. 1e using Fisher's exact test, Bonferroni modification). So, what is the significance of this new five subtype classifier since the clinical significance of the new classifier cannot be validated in an independent dataset (Abacus trial)?

We agree with the Reviewer that the long-term impact of a subtyping system will be determined by its clinical utility. However, we disagree about the accuracy of the new classifier to be validated in ABACUS. Using multiple orthogonal methods that included GSEA (**Figs. 1f,2b**), PD-L1 IHC (**Fig. 2d**), CMap signatures (**Supplementary Fig. 4**), and genetic alterations (**Supplementary Figs. 5 and 6**), we validated the classifier's prediction of subtypes in ABACUS. We identified agreement between PURE01 and ABACUS subtypes in all of these domains.

Despite the concordant features of each subtype when compared between trials, the pathologic response was different in the two trials, with greater response achieved in S2 and S3 subtypes in PURE01 (**Figs. 1b,2e**). We hypothesize about the differences in response of these two subtypes. Overall, response to pembrolizumab in PURE01 was greater than response to atezolizumab in ABACUS. This clinical response may be separate than the biologic activity of the subtypes. Causes of differences in response of S2 and S3 include: 1) greater number of cycles of therapy (three cycles of pembrolizumab in PURE01 vs

two cycles of atezolizumab in ABACUS), 2) cis-eligibility of PURE01 tumors, which may have influenced response to therapy, and 3) differences in drug activity of pembrolizumab compared to atezolizumab.

To our knowledge, to date, ours is the first study to have performed a cross-therapy comparison. Even more traditional biomarkers (TMB, PD-L1) have not demonstrated a consistent association with response when comparing trials. In the discussion (lines 440-51), we address the limitations of the work reported, including the comparison of the two trials. Validating the clinical significance of subtyping systems will be future work.

R1.6. On page 8, the authors used 10% of mutations as a cutoff to find associations of mutations with hallmark gene sets. What is the rationale to use a 10% cutoff?

We agree that a 10% cutoff is somewhat arbitrary. We set this threshold based on the small size of some of our subtypes, which ranged from 8 to 26 patients (**Fig. 1a**). If a mutation occurred in one patient from the S4 subtype (12% of that subtype) this would likely have limited significance. Thus, we selected a 10% minimum prevalence of alterations in a subtype as a threshold. A recent publication on GU tumors (Motzer et al 2020) reported for a similar-sized cohort and similar number of subtypes, and also used this threshold.

Motzer RJ, Banchereau R, Hamidi H, Powles T, McDermott D, et al. Molecular Subsets in Renal Cancer Determine Outcome to Checkpoint and Angiogenesis Blockade. *Cancer Cell*. 2020 Dec 14;38(6):803-817.e4. doi: 10.1016/j.ccell.2020.10.011. PMID: 33157048.

R1.7. Mutations in KRAS and FGFR3 appear to be found at a higher frequency in S1. Is there a 'direct' association between the responses and mutations of KRAS and FGFR3 in S1?

We compared the frequency at which somatic mutations in FGFR3 and KRAS were found in the S1/LumE subtype relative to the n=82 cohort, and to other subtypes. While FGFR3 mutations were more prevalent in S1, they were not significantly enriched in S1 vs not-S1 after correcting for multiple hypothesis testing ($p = 0.067$, Fisher's Exact test, Bonferroni correction) (**Fig. R1.7.1**). In contrast, mutations in KRAS were significantly enriched in S1 vs not-S1 ($p = 0.0071$, Fisher's Exact test, Bonferroni correction).

Figure R1.7.1. Frequency of KRAS and FGFR3 somatic mutations in S1/LumE vs the PURE01 n=82 pre-treatment cohort (S1 vs n_82), and vs non-S1 subtypes (S1 vs notS1).

We did not find a statistically significant association between somatic mutations in FGFR3 or KRAS and pathological response in S1 tumors ($p = 1.0$ for both genes, Fisher's Exact test, Bonferroni correction, **Fig. R1.7.2**). While we acknowledge that the lack of statistical significance may be due to the small cohort

size, these results further support our overall hypothesis that response is secondary to the overall subtype, and that subtype membership reflects the overall gene expression profile of a tumor, rather than mutation in a single gene.

Figure R1.7.2. Association of somatic mutations in FGFR3 and KRAS with response in the n=82 PURE01 pre-treatment cohort.

R1.8. Perform the same mutation analysis using ABACUS data to show this new classifier could find (at least) the same biology.

Because the PURE01 S1 subtype is important in the work reported here, we assessed all genes from those most frequently altered in S1 by somatic mutations (21 genes) or copy number variation (CNV, 7 genes) (**Fig. 3c**). We compared genetic alterations for S1 vs not-S1 subtypes in the PURE01 and in ABACUS cohorts. We found that both cohorts had statistically higher frequencies in S1 vs not-S1 only for FGFR3 somatic mutations and copy number amplifications of cell cycle-related cyclin D1 (CCND1) (**Supplementary Fig. 6a**). In subtypes other than S1, alterations in certain genes may be statistically enriched, e.g. RB1 mutations in PURE01 S2), but we limited our analysis to the genes noted above. Complementing these results for genetic variations, we found that GSEA results for MSigDB Hallmarks were consistent between PURE01 S1 and ABACUS predicted S1, consistent with similar biology identified by the classifier (**Figs. 1f,2b**).

R1.9. Fig 4a and Fig S4 showed that S5 and S3 were enriched with both stroma and immune signatures. In S5 (StromR), not only stroma but also immune scores were highest in both ESTIMATE and MCP-counter. Since, both S5 and S3 showed good survival outcomes, what is the role and relationship between immune cells and stroma cells here?

The reviewer again raises an astute point about immune/stromal interaction. We agree that this interaction is potentially critical for immune function. Probably the most intriguing differences between S5 and S3 were EMT and antigen presentation machinery signatures enriched by GSEA in S3 (**Table 1**). We were unable to profile subtypes S3 and S5 by DSP, mostly due to cost constraints and the overall favorable responses of these subtypes, but future work could involve a spatial evaluation of stromal and immune cells in prospectively collected samples.

Cancer-associated fibroblasts (CAFs) can contribute to EMT (Li et al. 2017), and we found that that mCAF and iCAF genes were most highly expressed in S5 tumors (**Supplementary Fig. 8b**; Methods, from line 887). This suggests that stromal cells play more active roles in immune regulation in this subtype. We have added this result in lines 253-5. Comparing GSEA results for CD8 effector pathways and antigen presentation pathways (**Table 1**), we observed enrichment in S3 relative to S5. Thus, S3 tumors appear to have less activation of CAF/stromal pathways and more immune activation, potentially explaining the greater response of this subtype to pembrolizumab.

Li H, Courtois ET, Sengupta D, Tan Y, Chen KH, et al. Reference component analysis of single-cell transcriptomes elucidates cellular heterogeneity in human colorectal tumors. Nat Genet. 2017 May;49(5):708-718. PMID: 28319088.

R1.10. Although the authors claim that S4 is a luminal subtype, S4 appears to be a mixture of neuronal, basal and luminal tumors. If the authors would like to identify the unique proteomic profiles in the TME that could contribute to CPI outcomes in luminal tumors, S4 should be removed in PCA analysis in Figure 4c or change the description of analysis.

Figure 4c compares the areas of interest (AOIs) from tumor microenvironment of responders and non-responders from S1, S2 and S4 using DSP. We selected these three subtypes because their responses varied (**Fig. 1b**), and samples in S1/2/4 remained together in n=82 and n=113 consensus clustering (**Fig. 5a**). Alternative MIBC subtype classifiers indicated that S1 and S2 were mostly luminal, while S4 contained *some* luminal tumors (**Fig. 1e**). The reviewer points out that S4 is not strictly luminal. Of the two S4 tumors evaluated in **Fig. 4c**, one was luminal (a non-responder) while the other was neuronal (a responder), based on TCGA subtypes.

The goal of **Fig. 4c** was actually to compare these three subtypes, and not strictly to compare luminal tumors. Thus, we have rephrased the wording from the manuscript describing this comparison. We have changed (lines 275-6) from "... (PCA) to capture the differences between the CR and NR samples from the luminal subtypes S1, S2, and S4..." to:

"...(PCA) to capture the differences between the CR and NR samples from the luminal subtypes S1 and S2, and from S4..."

R1.11. On p11, the authors should provide the exact number of sample distribution in subtypes such as 'most of the resistant tumors originated from S1 and were classified as S6 after pembrolizumab (pre: n=x in S1; post n=y in S6, n=z in S7, etc)

Figure 5a shows the relationship between the sample placement in the pre-treatment (n=82) subtypes and the pre+post-treatment (n=113) subtypes. To respond to the reviewer, we have added a **Supplementary Table 11** (which we also show below), and have adjusted the manuscript text to refer to this new supplementary table (starting at line 300):

We first re-analyzed the cohort of 113 tumors (82 pre- and 31 post- treatment tumors) by unsupervised consensus clustering. Samples from original five subtypes clustered together, particularly S1, S2, and S4, while 24 (77%) of the 31 post-treatment tumors clustered within two new subtypes (hereafter named S6 and S7) (**Fig. 5a**). Approximately half of the resistant tumors (14, 52% of 27 matched pairs) originated from S1, and nearly 80% of these (11 of 14) were classified as S6 or S7 after pembrolizumab (**Supplementary Table 11**).

Supplementary Table 11. Summary of the distribution of pre-treatment (n=82) subtypes into pre+post-treatment (n=113) subtypes, for the n=27 matched pre-post-treatment sample pairs.

Pre-treatment subtype	Pre+post-treatment (n=113) subtypes							Matched pairs	
	S3 (Pre.S4)	S1 (Pre.S1)	S5 (Pre.S2)	S4	S2	S6	S7	n	%
S1	1	0	1	0	1	6	5	14	52
S2	0	0	0	0	0	2	0	2	7
S4	0	0	0	0	0	2	1	3	11
S5	0	0	0	1	1	1	2	5	19
S3	0	0	0	0	1	0	2	3	11
Sum	1	0	1	1	3	11	10	27	100

R1.12. Provide silhouette score analysis and CDF plot to define k=7 solution in Fig 5a. Also, subtype membership of some tumors was switched before (k=5) and after combining post tumors (k=7). How many tumors were switched?

Figures R1.1(1)d-f show consensus heatmap, CDF plots, and silhouette width results for the seven-cluster solution for the PURE01 n=113 pre+post-treatment cohort. The silhouette plots are specifically shown in **Fig. R1.1f**. In our response to **R1.1**, we described how we work with clustering results from ConsensusClusterPlus, and summarize what recent large-cohort studies in our field have reported for silhouette widths in consensus clustering. We note that average per-subtype silhouette widths in three publications in our field are comparable to those for most of our n=82 and n=113 subtypes.

The reviewer also asks how many tumors switched subtype membership before (k=5) and after combining post tumors (k=7). Below we outline how we approached this, and the results.

Methods. From the seven-cluster n=113 clustering solution, we removed samples from S6 and S7, and from the remaining S1 to S5 we removed five post-treatment samples, leaving n=77 of n=82 pre-treatment samples. While most subtype numbers changed between the n=82 and n=77 solutions (e.g. n=82 S4 became n=77 S3), such subtype numbers have little meaning. Given this, to show the relationships between pre-treatment samples in PURE01 n=82 S1 to S5, and pre-treatment samples in the filtered n=77 S1 to S5, we generated a Sankey-like diagram. We summarized the relationships in a sample count table, and assessed the concordance between the n=82 and n=77 clustering solutions with a Chi-square test on this table.

Results. As outlined in Methods, we assessed subtype switching for n=77 (94%) of n=82 pre-treatment samples (**Fig. R1.12a**). Of n=49 filtered samples in S1, S2, and S4, only one (2%) S4 sample switched subtypes. In contrast, relatively few S5 and S3 samples remained together; we estimate that 4 (33%) of 12 S5 samples and 9 (56%) of 16 S3 samples switched subtypes (**Fig. R1.12.2b**).

Figure R1.12. Subtype switching between n=82 subtypes and n=113 subtypes. **A)** Sample relationships for n=82 pre-treatment samples and filtered n=77 samples (see Methods). **B)** Subtype switching in n=77 pre-treatment samples. Red boxes compare sample counts in the n=82 cohort vs the filtered n=77 cohort.

R1.13. In addition to H&E staining (Fig 5b), to demonstrate ‘scar-like stromal’ features’ and ‘immune infiltrate’, please provide molecular marker gene expression data and IHC data in S6 and S7. Fig 5b is a representative image of each subtype. Are there any statistics showing how many tumors in S6 have scar-like features and how many tumors have immune-invasive features?

Unfortunately, we are unable to perform IHC on the tissue, as we have little left for analysis.

To respond to the reviewer using the RNA-Seq data, we calculated ‘scar-like stromal’ features using ESTIMATE stromal scores. We found that all but five tumors had an increased stromal score in post-treatment tumors (**Fig. 5g**), and we have added this to the results (lines 320-1). Using other methodology, Seiler et al (Clin Cancer Res, 2019) describe one of their clusters as ‘CC4-Scar-like’. This cluster had high expression of genes associated with a p53-like signature; lower proliferation, characterized by

decreased expression of cell-cycle genes'; high expression of EMT and ECM marker genes; and low immune infiltration, but a markedly high stromal content.

Seiler R, Gibb EA, Wang NQ, Oo HZ, Lam HM, et al. Divergent Biological Response to Neoadjuvant Chemotherapy in Muscle-invasive Bladder Cancer. *Clin Cancer Res.* 2019 Aug 15;25(16):5082-5093. PMID: 30224344.

We annotated S6 subtype tumors as having "scar-like stromal features" based on characteristics shared with the subtype described by Seiler et al. Fourteen (82%) of 17 S6 tumors had 'scar-like' features and 9 (75%) of 12 S7 tumors were 'scar-like'. Following this, we assessed the following:

1. Using the MDA classification system (Choi et al, *Cancer Cell*, 2014) available in the BLCAsubtyping package v2.1.1, 14 of 17 (82%) S6 tumors were classified as 'p53-like'.
2. By GSEA, S6 and S7 tumors had repressed pathways associated with proliferation i.e., 'G2M checkpoint' and 'E2F targets', and enriched EMT pathways (**Fig. 5e**).
3. Using a list of ECM marker genes from Choi et al, *Cancer Cell* 2014 (LAMA5, LAMA4, COL6AA1, LAMB2, LAMB1, COMP, THBS4, THBS3, ITGA8, VWF), we found that S6 tumors had high ECM marker gene expression (**Fig. R1.13.1**).

Figure R1.13.1. Distributions of ECM signature z-scores across the seven pre+post-treatment sample subtypes (n=113).

4. S6 tumors also had elevated levels of expression of MYH11, CNN1 and DES. Both Seiler et al 2019 and Choi et al 2014 describe these genes as associated with a 'wound-healing' phenotype (**Fig. R1.13.2**).

Figure R1.13.2. Distribution of FPKM gene expression three genes associated with a wound-healing phenotype (MYH11, CNN1, and DES) across the seven n=113 pre+post-treatment sample subtypes.

5. Lastly, we found that S6 tumors trended towards a higher stromal content (p = 0.092, Kruskal-Wallis test, uncorrected for multiple testing) identified by immune deconvolution (**Supplementary**

Fig. 8b, right), similar to the features described for the “scar-like” phenotype tumors described by Seiler et al 2017. Similarly, S7 tumors had a non-significant increase in immune score relative to S6 ($p = 0.13$, Kruskal-Wallis test, uncorrected) (**Supplementary Figs. 8b,c**). See the first paragraph of R1.15, below.

The reviewer also asks how many S6 tumors have immune-invasive features. **Supplementary Fig. 8c** shows distributions of ESTIMATE ImmuneScores for $n=113$ subtypes, and reports nonsignificant adjusted p values ($p_{adj} = 0.20$) for both S6 vs S7. Nine (53%) of 17 S6 samples had ImmuneScores that were at or above the median.

R1.14. If S6 and S7 are distinct from S1-5 molecularly to form additional cluster, the authors should provide the molecular difference between S1-5 and S6-7 including in fig 5e and 5f.

The Pre-Post covariate track in **Fig. 5a** (right) shows that the $n=113$ pre+post-treatment consensus subtypes S6 and S7 largely consisted of post-treatment samples ($p=8.7 \times 10^{-13}$, Fisher’s Exact test that compared pre/post vs the seven subtypes).

As requested, in **Figs. 5e,f** we now show GSEA results for both $n=82$ S1 to S5 and $n=113$ S6 and S7, for 12 MSigDB Hallmark gene sets, and 12 gene sets from Mariathasan et al. 2018. Taken together, and with results in R1.13 above, these results are consistent with $n=113$ subtypes S6 and S7 being molecularly distinct from subtypes S1-S5.

R1.15. In addition to Fig 5g, additional figures showing immune scores/stromal scores/tumor purity stratified by S6 and S7 should be generated as the authors claim that there are differences in stromal characteristics and immune invasion between S6 and S7.

We have added box-whisker plots of ESTIMATE ImmuneScore, StromalScore, and TumorPurity for S6 and S7 (**Supplementary Fig. 8c**). While the median ImmuneScore was higher in S7, and the median StromalScore higher in S6, none of these three distributions differed statistically between S6 and S7 ($p_{adj} = 0.30, 0.30$ and 1.0 , respectively, Kruskal-Wallis tests, Bonferroni correction).

For $n=113$ subtypes S6 and S7, **Fig. R1.15** shows distributions of MCP-counter v1.2.0 results for the ten cell types predicted from bulk RNA-seq data. For eight of 10 cell types, medians were higher in S7 than S6; however, after applying Kruskal-Wallis tests and adjusting p values for multiple hypothesis testing (Bonferroni $\times 10$ correction), only distributions of neutrophils remained significantly higher in S7 ($p_{adj} < 0.1$).

Figure R1.15. MCP-counter cell types in $n=113$ S6 and S7. **A)** Distributions of MCP-counter v1.2.0 cell types in S6 and S7. **B)** For each cell type, p values from a Kruskal-Wallis test, and Bonferroni-corrected ($\times 10$) adjusted p values.

R1.16. On p13, authors described that “the KDM5B regulon was repressed in subtypes with high immunoscores (S3, S5)” – This is incorrect. The regulon activity is relatively very low only in S5, not S3. The KDM5B regulon activity of S3 is similar to that of S2 or S4 in Fig S8b. Also, the negative association between KDM5B regulon activity and immune scores should be validated in independent datasets including TCGA data.

We appreciate the reviewer’s attention to detail. The reviewer asks about two issues: 1) the relationship between per-subtype KDM5B regulon activities and per-subtype ImmuneScores, and 2) validating PURE01’s negative correlation (association) between KDM5B regulon activities and ImmuneScores in the independent TCGA-BLCA cohort.

For the first issue, the reviewer is correct, and we apologize for the error. **Supplementary Fig. 10b** shows distributions of both KDM5B gene expression and KDM5B regulon activity. We have corrected the manuscript text (lines 344-6) to read:

The KDM5B regulon activity was highest in 18 (69%) of 26 S1 samples, and was most strongly and consistently repressed in subtype S5, which had the highest median ImmuneScore (**Supplementary Fig. 10a,b; Supplementary Table 7**).

For the second issue, we have validated the negative correlation between KDM5B regulon activities and ESTIMATE ImmuneScore in the TCGA-BLCA cohort (**Fig. 6b**, and **Supplementary Table 17**). In the manuscript’s Methods, we have added a subsection on ‘Validation of the KDM5B regulon in the TCGA-BLCA cohort’ (starting at line 923).

Reviewer #3 (Remarks to the Author): expertise in digital spatial profiling and transcriptomics

The paper implements omics-technologies, including digital spatial profiling and DNA-seq, to profile characterize molecular signatures of the tumor microenvironment for pre-and post-treatment (pembrolizumab immunotherapy). The paper characterized the five subtypes of muscle-invasive bladder cancer (MIBC), developed a classifier for distinguishing a tumor sample subtype, and found that histone demethylase KDM5B as a repressor of tumor immune signaling pathway plays an important role in the treatment resistance mechanisms. This finding can be used for stratifying patients and giving an alternative treatment for the poor therapy responsiveness. However, the paper has some limitations on computational methods.

Major comments:

R3.1. The reviewer cannot access data and code scripts to reproduce and validate the findings. Therefore, the following comments are mainly based on the manuscript, figures, and tables. The reviewer cannot evaluate the scientific rigor and reproducibility without data and analysis code.

We apologize to the reviewer. Data and code are available at:

<https://drive.google.com/drive/folders/1A0QRCTtCfV-HkmMgxLpNyKkrcAJ6Ocbf?usp=sharing>

R3.2. In the introduction part, “However, biomarker analysis in ABACUS..... TMB were associated with response in ABACUS.” When read the first sentence, the reviewer expects the author to explain why the two clinical trials show a molecular difference rather than “somewhat discordant.” Secondly, what is the definition of “activated CD8+ gene expression?” How to define “activated”? It is higher CD8+ gene expression or other a group of signature genes associated with cytotoxic T cells activity.

The reviewer requests that we “explain why the two clinical trials show a molecular difference rather than ‘somewhat discordant’”, noting a sentence in the introduction. We understand that the issues of concern

are 1) molecular differences between PURE01 and ABACUS cohorts, and 2) discordant clinical results between PURE01 and ABACUS trials.

In the published papers (Necchi et al. 2018, Powles et al. 2019), the biomarker emphasis was placed on TMB and on PDL1 expression. PDL1 and TMB were associated with response in PURE01 (Necchi et al 2018), but not in ABACUS (Powles et al. 2019). In our hands, TMB trended to association with response in PURE01 (**Supplementary Fig. 5d**, $p = 0.10$). Therefore, we suggest that the molecular differences between the cohorts are limited, but the trial responses to pembrolizumab and atezolizumab treatments differed. In the manuscript (lines 201-4), in the Discussion's limitations paragraph (lines 440-51), and in our response to **R1.5** above, we suggest reasons for the trial discrepancies.

The reviewer also asks how "activated CD8+ gene expression" was defined. In the manuscript (line 81), we refer to the biomarker analysis of the original PURE01 paper (Necchi et al. 2018's reference 4), which quantified "activated" CD8+ cells that express the protease granzyme B+. Because this can be difficult for the reader to interpret, we removed the "activated" adjective from line 81.

Necchi A, Anichini A, Raggi D, Briganti A, Massa S, et al. **Pembrolizumab as Neoadjuvant Therapy Before Radical Cystectomy in Patients With Muscle-Invasive Urothelial Bladder Carcinoma (PURE-01): An Open-Label, Single-Arm, Phase II Study.** *J Clin Oncol.* 2018 Dec 1;36(34):3353-3360. PMID: 30343614.

Powles, T., Kockx, M., Rodriguez-Vida, A. et al. **Clinical efficacy and biomarker analysis of neoadjuvant atezolizumab in operable urothelial carcinoma in the ABACUS trial.** *Nat Med* 25, 1706–1714 (2019).

R3.3. Regarding the histology representative tumor (Figure S2c), the author should give a systematic evaluation for whether all samples displayed a similar histological feature in five categories, respectively.

We agree with the reviewer that evaluation of tumor histology is valuable. All slides were reviewed by the project's genitourinary pathologist (Bonnie Choy). We have updated the section related to histology in the manuscript (lines 176-85) to provide additional information relevant to each subtype. In general, the tumors within each subtype demonstrated similar morphologic features. However, it is difficult to succinctly and meaningfully share all histologic features evaluated without taking attention away from the main points of the manuscript. We plan to further explore the morphologic findings in a more focused and in-depth manner in the future.

R3.4. In the Method section (single-sample expression subtype classifier), first, please introduce the rationale why choose GLMnet as a classifier basis. Second, could the author provide more results regarding the model training regarding parameter tuning because there are lots of parameters (or hyperparameters) to determine the final model? A table showing parameters' combination and results (i.e., show precision, recall, or accuracy) will be helpful. Third, the authors used two normalization methods as input to train the classifier based on GLMnet. What is the rationale here to compare the two normalization methods? These two normalization methods push me to think about the previous GSEA analysis, whether it has two results?

GLMnet Elastic nets are generalized linear models with regularization. While they are normally used in regression contexts, they can be easily adapted to classification tasks using binomial family models, and the R package caret includes this possibility by default. Indeed, we included them because previous publications in our field had used this type of classifier (Seiler et al, 2017; de Jong 2019). We also tested a random forest (RF) model and a simpler nearest-centroid (NSC) model, all in 3-fold repeated cross-validation runs. Consistently, the best-tuned elastic net models outperformed the random forest models.

The elastic net classifier's cross-validation accuracy was 92%. Given that we lack the ground-truth labels for ABACUS subtypes, we are restricted to assessing accuracy with cross-validation metrics.

De Jong JJ, Liu Y, Robertson AG, Seiler R, Groeneveld CS, et al. Long non-coding RNAs identify a subset of luminal muscle-invasive bladder cancer patients with favorable prognosis. *Genome Med.* 2019 Oct 17;11(1):60. Doi: 10.1186/s13073-019-0669-z. PMID: 31619281.

Seiler R, Ashab HAD, Erho N, van Rhijn BWG, Winters B, et al. Impact of Molecular Subtypes in Muscle-invasive Bladder Cancer on Predicting Response and Survival after Neoadjuvant Chemotherapy. *Eur Urol.* 2017 Oct;72(4):544-554. Doi: 10.1016/j.eururo.2017.03.030. PMID: 28390739.

Parameter tuning In a new **Supplementary Fig. 3**, panels (a-c) show parameter tuning metrics, and, for the final best models, panel (d) shows accuracy and Cohen's kappa coefficient. From these results we chose to use a tuned elastic net model. The regularizations in the elastic net model have properties that may explain its good performance for this task: it can select features through ridge penalization, and it will tend to group highly correlated predictors.

Two normalization methods Fig. 1a's heatmap was generated using variance-stabilizing (VST) normalization, and we have modified the Methods text to reflect this (lines 809-10). In contrast, for classifier training and testing, we compared the normalization that we used in all of the analyses with variance-stabilizing (VST) normalization, as suggested by Bioconductor workflows. For subtype classification we found no benefit in using VST normalization; instead, we found that VST led to a small drop in accuracy in all models (accuracy, best GLMnet model: 89.7% with VST vs 92% with log-FPKM). Our intent was to find robust classifier models, and we tested alternate normalizations to gauge whether the choice of normalization would greatly change accuracy. Ultimately, for consistency, we used the FPKM normalization for classification.

GSEA and normalization We used VST-normalized gene expression data in only one analysis: differentially expressed genes for the heatmap in Fig. 1a. As described above, for classifier training and testing we evaluated VST-normalized expression data, but did not use it to generate the results that we report. We ran GSEA analysis only for FPKM-normalized, and not for VST-normalized gene expression; consequently, we have only one set of (FPKM-based) GSEA results. As we stated above, ultimately, we did not use the VST-based classifier for the ABACUS validation cohort.

R3.5. In the results part (assessing CPI-MIBC subtypes in an independent cohort), the reviewer likes validating the model using the secondary dataset and digging out more biological insights. However, before going to this section, the author should show the power of the classifier, as suggested in comment #3. Either a figure or a summary table will be helpful.

As we noted above (R3.4), we lack the ground-truth labels for ABACUS subtypes, so are restricted to assessing classifier accuracy with cross-validation metrics. The elastic net classifier's cross-validation (3-fold, 5 repeat) accuracy was 92%, and its sensitivity (True Positive Rate or 'power') was 92.1%.

As a secondary validation (not based on RNA), we have documented above (R1.8) that genetic alterations were similar in PURE01 subtype S1 and predicted ABACUS subtype S1, and the final sections of the manuscript focus on subtype S1.

Classifier 'power' has been assessed with a Vapnik-Chervonenkis (VC) dimension (e.g. Soni D, 2018). In peer-reviewed publications, we find VC-dimension described as 'bounds on sample complexity', or classifier 'generalizability' (Holden and Niranjana, 1995; Shao X et al 2000). While using a VC-dimension to evaluate genomic or transcriptomic classifiers is interesting theoretically, we are unaware of cancer genomics studies in which this has been done. This is a possible focus for future work.

Holden SB, Niranjana M. On the practical applicability of VC dimension bounds. *Neural Comput.* 1995 Nov;7(6):1265-88. doi: 10.1162/neco.1995.7.6.1265. PMID: 7584902.

Shao X, Cherkassky V, Li W. Measuring the VC-dimension using optimized experimental design. *Neural Comput.* 2000 Aug;12(8):1969-86. doi: 10.1162/089976600300015222. PMID: 10953247.

R3.6. In the result section (somatic mutations of CPI-MIBC subtype), why using fgsea to calculate enrichment score. In the previous analysis in the first section, the author used CERNO method to show the enrichment results, but the method was changed in this section without any rationale. The author should elucidate why they changed the method. Because two different computational methods mean two different algorithms and assumptions, resulting in poorly reproducible results (ref: PMID: 32695141). In addition, it may be a minor comment, but in terms of professionalism, the software version is not unified, which is not usually seen in one paper. For example, R shows two versions in one study, including 3.6 and 4.1. DESeq2 also shows two versions, including 1.30 and 1.26.

For the issue of fgsea vs CERNO tests, we now report MSigDB Hallmark gene set enrichment in tumours used CERNO tests, comparing mutated vs wild-type for 13 genes (**Fig. 3a**).

We addressed the issue of software versions as follows. In general, the manuscript uses standard statistical methods (e.g. Fisher's Exact tests, Wilcoxon tests, p.adjust(), ...) that should be sufficiently stable that we anticipate that the results reported should be insensitive to the versions of R and of R packages. For the heatmap in Figure 1a, we had used DESeq2 v1.26.0 in R v3.6. To check that the results were insensitive to versions, we have re-run the analysis with DESeq2 v1.36.0 and R v4.2.0 (which are current at the time of writing). We found the differentially expressed genes were identical between the previous and current results. For single-cell RNA-Seq analysis we used current versions of Python and related tools.

R3.7. Regarding the result section (Subtype differences in the tumor microenvironment by bulk and spatial analysis), using deconvolution for bulk-RNAseq data could distinguish the dominant cell types in TME, but the cell types in deconvolution results still show a low granularity. For example, [MCP-counter] cell types include CD8 T cells, B cells, overall T cells, etc. This may be another reason to perform DSP spatial proteomics experiments (71 proteins) and analyses to show the real cells' colocalization and higher granularity of cell composition. However, the spatial analysis is not eye-catching (the take-home message is not clear to me), and where do the other proteins go (Figure 4d shows 13 proteins)?

The reviewer asks for striking differences in the TMEs of the subtypes. In results from DSP data for subtypes S1, S2 and S4 — PCA in **Fig. 4c**, and expression of immune-regulatory proteins in **Fig. 4d** — we find large differences between TMEs of non-responders. We suggest that these spatial findings further support our hypothesis that spatial differences in immune regulation between subtypes may explain the response to pembrolizumab. At the same time, we agree that spatial analysis in immunotherapy is a relatively new field, and we hope to gain further insight with more samples and further analysis in the future analysis with RNA and protein data. Given the limited number of samples assessed by DSP, we have toned down the statements for this part of the work (lines 292-4), and acknowledge the constrained samples in a limitations section in the Discussion (lines 440-51).

To respond to the reviewer's request to indicate 'where the other proteins go', we did the following analysis.

Method. We read in SNR-normalized DSP data for 71 proteins from tumor microenvironment (TME) areas-of-interest (AOIs, **Fig. 4b**), and for three negative controls (rabbit IgG, and mouse IgG1 and IgG2a). Because proteins whose abundances are comparable to or below the levels of negative controls are less reliable as indicators of biology, we assessed Spearman correlations between all proteins whose median levels were above the median level for the most-abundant negative control, rabbit (Rb) IgG. For the 17 TME AOIs for one NR sample from each of subtypes

S1, S2, and S4, and the 21 TME AOIs for one CR sample from each of these three subtypes, we used Spearman correlations in R's `corrplot()` function to assess relationships between 35 NR and 42 CR SNR-normalized protein levels. We then annotated the correlation heatmaps using functional 'probe group' information (NanoString, Seattle WA).

We removed approximately 35 to 40% of the 71 TME proteins in NR and CR samples that were expressed at levels that were comparable to or below the negative control Rb IgG (**Figs. R3.7a,b**). For NR samples, this filtering removed Treg marker proteins CD25 and FOXP3. For both CR and NR samples, filtering removed cancer/testis (CT) antigen markers MART1 and NY-ESO-1, hormone receptor markers ER-alpha and PR, and the neutrophil marker CD66b.

For TME AOIs from one sample each of subtypes S1, S2, and S4 (**Fig. 4b**), we assessed correlations for the remaining proteins. Samples in these three subtypes had clustered together in both $n=82$ and $n=113$ runs (**Fig. 5a**), suggesting that the biologies of these subtypes were more distinct than those of S5 and S3. We separated CR samples from NR samples, as we did for PCA and the levels of 13 TME proteins in these subtypes (**Figs. 4c,d**). The correlation heatmaps differed between CR and NR samples, so were generally consistent with **Figs. 4c,d**.

Figure R3.7. DSP protein abundance and correlations for tumor microenvironment (TME) AOIs from subtypes S1, S2, and S4. **a,b** Empirical distribution functions of median abundance for 71 proteins, and medians for three negative controls (vertical grey lines): **a**) for 17 TME AOIs from NR samples 12 (S1), 17 (S2), and 99 (S4, see **Figure 4b**). **b**) for 21 TME AOIs for CR samples 88 (S1), 37 (S2), and 86 (S4). For correlations, we used only the 35 proteins that were more abundant in Rb IgG in NR samples, and the 42 proteins in CR samples. **c,d** Spearman correlations for proteins with median expression above the median for the negative control Rb IgG (see **a,b**): **c**) for 35 thresholded proteins in NR samples. **d**) for 42 thresholded proteins in CR samples. Boxes or triangles highlight proteins with strong positive or negative correlations. Diagonal cells would have correlations of 1.0, and are not shown. **e,f** DSP probe (functional) groups for the proteins in the correlation heatmaps.

R3.8. For the last result section (FGFR3 and KDM5B are potential LumE/S1 subtype targets), it used bulk-seq to identify the potential regulator and then uses scRNA-seq & ATAC-seq to validate KDM5B's role. The author could try to use SCENIC (python package) or IRIS3 (webserver) to further investigate the regulon network at the single-cell level with an advanced computational framework and validate the findings.

We appreciate Reviewer #3's recommendation to use SCENIC for regulon analysis of our single-cell RNA-Seq data set. RTN was developed using bulk RNA-Seq data. In such data, it analyzes the expression of target genes that are regulated by transcription factors in a mixture of multiple cell types. Regulon analysis in single-cell RNA-Seq data has the potential to inform on regulatory relationships for specific cell types.

To evaluate the KDM5B regulon activity in scRNA-Seq data, we first used pySCENIC to identify clusters of similar cell types, producing a UMAP clustering (**Fig. 6c**) (Methods, starting at line 1069). We then re-clustered only the epithelial cells (**Fig. 6c**, and found that luminal marker genes were relatively highly expressed in six of the nine epithelial cell clusters (**ii**). Consistent with RTN results from bulk RNA-Seq data (**Fig. R3.8**, **Fig. 1b**, and **Supplementary Fig. 10b**). KDM5B regulon activity was activated (i.e. strongly positive) in the subset of luminal epithelial cells in which KDM5B expression was high (**iii, iv**), and repressed (i.e. strongly negative) in luminal epithelial cells in which KDM5B expression was low (**iii,v**).

Figure R3.8. For KDM5B, relationship of regulon activity to RNA-Seq gene expression in the TCGA BLCA cohort (n=404). We sorted this cohort by ascending regulon activity. The lower FPKM graph shows a loess smoothing curve.

Minor comments:

R3.9. Please add more references in the introduction part to strengthen the background section. For example, the author adds references for supporting the “neoadjuvant paradigm” in the first paragraph.

In the introduction (line 75), we have added three more references to support the neoadjuvant paradigm, which is the standard of care for treating bladder cancer, and is supported by the AUA, NCCN, ASCO and EAU.

Witjes JA, Bruins HM, Cathomas R, Compérat EM, Cowan NC, et al. **European Association of Urology Guidelines on Muscle-invasive and Metastatic Bladder Cancer: Summary of the 2020 Guidelines.** Eur Urol. 2021 Jan;79(1):82-104. doi: 10.1016/j.eururo.2020.03.055. Epub 2020 Apr 29. PMID: 32360052.

Flaig TW, Spiess PE, Agarwal N, Bangs R, Boorjian SA, et al. **Bladder Cancer, Version 3.2020, NCCN Clinical Practice Guidelines in Oncology.** J Natl Compr Canc Netw. 2020 Mar;18(3):329-354. doi: 10.6004/jnccn.2020.0011. PMID: 32135513.

Chang SS, Bochner BH, Chou R, Dreicer R, Kamat AM, et al. **Treatment of Non-Metastatic Muscle-Invasive Bladder Cancer: AUA/ASCO/ASTRO/SUO Guideline.** J Urol. 2017 Sep;198(3):552-559. doi: 10.1016/j.juro.2017.04.086. Epub 2017 Apr 26. Erratum in: J Urol. 2017 Nov;198(5):1175. PMID: 28456635; PMCID: PMC5626446.

R3.10. In the method section (Consensus expression subtypes), the author described “Spearman and Pearson distance” to do clustering. Did the author used Spearman and Pearson dissimilarity as distance or used Spearman and Pearson coefficient as distance.

Above, in our response to **R1.1**, we describe how we choose a consensus clustering solution that is informative for a research question. Below, we repeat what we said there on distances.

When we use ConsensusClusterPlus to calculate consensus expression clusters, we typically calculate distances using one minus a Pearson or Spearman correlation coefficient. For example, for a Pearson correlation, we would typically use the following commands ('T' is a log-transformed, median-centered gene expression matrix) -

```
ntp.FPKMs <- as.dist(1 - cor(T, method="pearson"))
# as.dist: This function computes and returns the distance matrix
computed by using the specified distance measure to compute the distances
between the rows of a data matrix.
ConsensusClusterPlus(ntp.FPKMs, ...
```

We have modified the Methods text to clarify this point (lines 799-801), and we also supply an R script for consensus clustering in our 'data + code' repository (for Fig. 1a).

R3.11. Please assume the diverse background readers, introduce terminology, and keep reading smoothly. The paper has some terminologies that need more clarifications. For example, what is CPS in figure 1a, and the reviewer cannot find the explanation for this term?

We have clarified that CPS is the 'combined positive score' for this specific biomarker (lines 502, 906).

We have checked acronyms throughout the manuscript to ensure that each is explained on first use. Acronyms include: AOI, BCR, BLCA, CMap, CPI, CR/PR/NR, CTL, DSP, GSEA, MIBC, MSigDB, PCA/PC1/PC2, ROI, and TME.

R3.12. In the Method section (Gene set enrichment analysis), paragraph “For PURE01 n=82, for each expression subtype, vs all other samples, we ...” need to be reorganized for grammar issues and unrelated words.

We apologize. We have rewritten the paragraph (lines 831-37).

Old paragraph: For PURE01 n=82, for each expression subtype, vs all other samples, we calculated a ranked set of ~15.5 thousand expressed protein-coding gene symbols (for ABACUS ~15.8k genes), again ranking genes by S2N. We again used 50 Hallmark gene sets and the 20 gene sets from⁵, as an input to tmod v0.46.2, running CERNO tests Click or tap here to enter text., typically with qval = 0.05, but relaxing this when appropriate. We summarized gene set results with 'dot' diagrams that we generated with ComplexHeatmap v2.6.2, setting dot radii to be proportional to the CERNO AUC.

Rewritten: For each PURE01 expression subtype we ranked ~15.5 thousand expressed protein-coding genes by signal-to-noise ratio (S2N). For a similar analysis with predicted subtypes in ABACUS we used ~15.8k coding genes. Then, working with 50 MSigDB Hallmark gene sets and

the 20-gene set from Mariathasan et al. 2018, we used CERNO tests (tmod v0.46.2), typically with $qval = 0.05$, to identify enriched and repressed gene sets for each subtype or predicted subtype. We summarized gene set results with 'dot' diagrams that we generated with ComplexHeatmap v2.6.2, setting dot radii to be proportional to the CERNO areas-under-the-curve (AUC).

R3.13. In the result section (somatic mutations of CPI-MIBC subtype), how many tumor samples were used for evaluating somatic mutations and copy number variation?

For PURE01 pre-treatment samples, we assessed somatic mutations and copy number variation for 82 tumors.

Reviewer #4 (Remarks to the Author): expertise in checkpoint immunotherapy response

The authors submit a comprehensive characterization of patient samples from two neoadjuvant trials of checkpoint inhibitors in bladder cancer. The overall goal of the study (p14) is to dissect the molecular heterogeneity of MIBCs to further define mechanisms of resistance. The manuscript is clear, structured logically, and well written. The figures are all relevant and well designed.

There are several key strengths to this work that are important to highlight.

1. Direct clinical relevance. Immunotherapy is an effective therapeutic for metastatic urothelial cancer. It is being tested in multiple phase III trials in the neoadjuvant setting. Biomarkers predictive of response to IO are lacking to date and could be immediately useful clinically.
2. Access to prospectively collected tissue samples. The authors analyzed pretreatment tissue from two therapeutic clinical trials of checkpoint inhibitors in localized bladder cancer: PURE-01 and ABACUS.
3. Access to high quality clinical data. By leveraging clinical trial tissue samples, each was well annotated with respect to therapeutic dosing and clinical and pathologic outcomes.
4. Extensive experiments resulting in a comprehensive "multi-omics"
5. The availability of a discovery and validation set: With access to tissue samples and annotation from two prospective trials, the authors are able to use PURE01 for discovery and ABACUS for validation.

Comments:

General:

This manuscript is extensive and far reaching, using samples described above, defining and then looking deeply into 5 new subtypes, extending them to a pre/post treatment analysis and including experiments done on a cell line linked to one of the subtypes. However, the extent of the analyses included in the manuscript falls prey to a trend best described by Bill Kaelin in an editorial for this journal: Nature volume 545, page 387 (2017). The work described here is careful and sound, but suffers from being too broad with many tangential inquiries that individually lack depth. The core set of investigations, once the analyses are revised to describe a direct link between biology and therapeutic response, will be of great value to the bladder cancer community. Furthermore, the tangential analyses could be papers in their own right, allowing distribution of credit to those doing the bulk of the work in an academic system that values first and last authorship. The comments below suggest ways to overcome this and produce a strong manuscript that can be accepted for publication without delay. It is not in our collective best interest to delay reporting these findings, nor public deposit of the tissue based data which will be a critical resource for further discovery in bladder cancer.

Major:

The authors collected RNA-seq data from 30 complete responders and 33 non-responders (omitting the middle clinical category of stable/partial responders). Unsupervised consensus clustering was then used to define five transcriptomic subtypes. These subtypes are applied to the ABACUS cohort as validation. The authors then carry these subtypes throughout the manuscript, correlating all other results to the subtypes (GESA, PDL1, cMAP, histology). They are also compared to existing subtypes (Lund, TCGA, consMIBC, MDA).

R4.1. It is not clear that the subtypes validate in ABACUS. A statistical comparison to validate these new subtypes should be provided. This is particularly important since such small numbers of patient/samples were used to derive the subtypes in the first place.

We appreciate the reviewer's recommendation for a statistical comparison of accuracy of the classifier between PURE01 and ABACUS. We compared subtypes in the two cohorts using multiple features that included gene expression by GSEA (**Figs. 1f,2b**), PD-L1 biomarker activity (**Fig. 2d,f**), genetic alterations (**Fig. 3c, Supplementary Fig. 6**), and correlations of CMap perturbagen results (**Supplementary Fig. 4**). These results demonstrated that the PURE01 subtypes and the predicted PURE01 subtypes in the ABACUS cohort had similar biologies. We note that statistical comparisons of similarity were constrained by the small number of tumors in each subtype. We suggest that the concordance of these diverse metrics indicate that the subtypes validate in ABACUS.

R4.2. It is not clear that the subtypes align with the 4 subtype classifications previously described. In particular, S4 S5 and S3 appear heterogeneous.

In **Fig. 1e**, our five n=82 subtypes were statistically aligned with four alternative subtype classifications (Lund/simpler, TCGA, consensusMIBC and MDA): adjusted p-values were in each case $\ll 10^{-10}$ (Fisher's exact tests, with a Bonferroni correction, **Supplementary Table 2**). We describe the transformation from nine Lund subtypes to five 'simpler' Lund subtypes in the legend of **Fig. R4.3**, below, and in Methods (line 813-15).

R4.3. It is not clear why the subtypes in and of themselves are of value. 4 others have already been defined, why not align these samples to those?

Our consensus expression subtypes are overall way to examine molecular heterogeneity. None of the alternative subtyping systems assessed (Baylor, UNC, CIT, Lund, TCGA, cMIBC, MDA) were as strongly associated with response as our five PURE01 subtypes (**Fig. R4.3**), by approximately a factor of seven ($p_{\text{adj}} = 0.076$ vs 0.51 for UNC, **Supplementary Table 2**). Ideally, we could identify combinations or alternative treatments for patients in specific molecular subtypes. We report this in lines 152-3. The work reported here focuses on subtype S1.

Figure R4.3. Associations between CR/PR/NR pathological response and subtype for the PURE01 n=82 subtypes and six-plus alternative subtyping systems. To transform nine Lund subtypes into five simpler Lund subtypes we combined Ba/Sq and Ba/Sq-Inf into 'Ba/Sq', GU and GU-Inf into 'GU', and UroA-Prog, UroB, and UroC into 'Uro'. We assessed the association of each set of subtypes with response with a Chi-square test, then corrected the p values for multiple hypothesis testing with a Bonferroni (x8) correction.

R4.4. It is not clear that the subtypes are reproducible within the same patient. Testing more than one TURBT chip from the same patient's procedure would help clarify this question.

The reviewer makes an interesting point about intratumor heterogeneity. Unfortunately, we are unable to isolate different parts of the TURBT chips for comparison. To our knowledge, such a comparison has not been performed previously on any other subtyping platform.

R4.5. The PURE01 S2 and S3 subtypes are enriched for response, but this enrichment is not validated in ABACUS, nor is the correlation described statistically.

While CR/PR/NR responses were not statistically different between S2 and S3 for either pembrolizumab or atezolizumab (Fig. R4.5, p=0.48 and 0.42 respectively), we agree that qualitative differences in pathologic response between the PURE01 and ABACUS trials suggest that S2 and S3 may have different responses in these two trials. As we noted for R1.5, above, we hypothesize this may be secondary to multiple differences in clinical factors outside of the biology of the tumors; specifically, the number of cycles of CPI, drugs used in the trials, and platinum eligibility.

Figure R4.5. The relationships between CR/PR/NR response and subtypes S1 to S5 and S2/S3 in PURE01 n=82, and predicted subtypes S1 to S5 and S2/S3 in ABACUS n=84.

R4.6 The overall goal of the paper is to provide insight into biology that can predict response and/or resistance. However, the -omics analyses are correlated to the new subtypes, not to response or resistance directly. For instance, to correlate PDL1 the authors state that the subtypes with highest responses have a higher proportion of PDL1 – notably this is true for the PURE01, not ABACUS. Therefore, rather than answer the 4 questions above which would require more samples and analyses for this manuscript, this reviewer recommends describing the new subtypes, acknowledging that they don't replicate in the ABACUS validation set, and then using the data already generated from the experiments done to correlate the following investigations directly with response and resistance using all samples from PURE01 (not just the extremes of the response spectrum).

1. PDL1 (ideally using CPS for PURE01 and sp142 IC score for ABACUS)
2. GSEA
3. CMap
4. Somatic mutations and copy number profiles
5. Bulk and spatial analysis using ImmuneScores by ESTIMATE analysis
6. The expression of immune cell markers and regulatory proteins

Any significant findings can then be validated using the ABACUS patient/samples and outcome data.

We appreciate the reviewer's thoughtful recommendations regarding comparisons for the subtypes and ways to perform the analysis. Both PURE01 and ABACUS had published biomarker analysis describing features associated with response or non-response. Yet, response to immunotherapy is a complex trait, and our hypothesis was that more than one feature contributes to response. In our manuscript, we tried to describe the "why" of response or non-response and our results indicate that the tumor biology clusters into five subtypes, and these vary strongly in being responsive or non-responsive. From our analysis, we suggest there is more than one possible explanation of clinical response. Thus, our hope was to identify the two or three molecular features that were associated with response.

R4.7. The analysis investigating B cell repertoires and immunoglobulin signatures and correlating them to response is well done, but would be strengthened by also including those with SD among the non responders for greater power. It was nice to see statistics applied here and a significant difference noted.

We thank the reviewer for this comment. For neoadjuvant therapy we don't really have a Stable Disease (SD) category, only tumors that have a complete response or not. Stable disease after neoadjuvant checkpoint immunotherapy suggests resistance.

R4.8. The paper in general suffers from over reliance on vague descriptive language to describe correlations and underuse of statistical comparisons. For instance, statements such as “S4 tumors had more mitotic features” “overall mutation frequencies were comparable to”, “results ...were largely consistent” “were relatively concordant” and “were enriched for” – are not specific enough to draw conclusions around relationships. Statistical support should be provided where correlations are claimed.

We have added quantitative or statistical support for many statements, while seeking to keep the text easily readable.

- For initial descriptions of how our five PURE01 n=82 subtypes related to classes predicted by Lund, TCGA, cMIBC, and MDA classifiers (line 143), we have added **Supplementary Table 2**.
- On lines 152-3, and in **Supplementary Table 3**, we report that our five PURE01 subtypes were more strongly associated with response than class calls from alternative subtyping systems.
- We did not find the phrase “S4 tumors had more mitotic features” in the manuscript.
- “Overall mutation frequencies” being “comparable” is in a sentence that starts at line 222: “Overall mutation frequencies in the PURE01 cohort were comparable to those previously reported in the TCGA MIBC dataset (**Supplementary Fig. 5c,d**).”
 - While the legend for this supplemental figure seems clear, we have added “PURE01”, “ABACUS” and “TCGA” titles above the respective oncoprints. Each panel states the overall percentages of somatic mutations (for ABACUS, ‘Short variant (SV)’ for each gene in the respective oncoprint. We suggest that this should be sufficiently quantitative for readers. For example, TP53 is somatically mutated in 57% of samples in PURE01, 68% in ABACUS, and 48% in TCGA. We note that some differences in mutation frequencies are to be expected, given that different teams carried out the PURE01 and ABACUS trials, and did the TCGA BLCA work.
- “Results ...were largely consistent” is in a sentence that starts at line 193: “GSEA results for Hallmark gene sets were largely consistent between PURE01 and ABACUS (Figures 1f and 2b, Supplementary Table 6).”
 - **Figs. 1f and 2b** are ‘dot’ graphical representations of GSEA AUCs for MSigDB Hallmarks, from CERNO tests. Dot area is proportional to the AUC value (see also **Supplementary Figs. 10f,g**). **Fig. R4.8** shows the relationship of Hallmark AUCs to adjusted p values for the PURE01 and ABACUS cohorts. Statistical results for these relationships were as follows. For PURE01 n=82, Hallmarks, CERNO tests, filtering at $AUC \geq 0.65$, and $p.adj \leq 0.01$; Spearman's rank correlation (alternative hypothesis: true rho is not equal to 0): $\rho = 0.74$, $p = 0.0$. For ABACUS n=84, Hallmarks, CERNO tests, filtering as for PURE01, Spearman's $\rho = 0.66$, $p = 1.2 \times 10^{-6}$.

- As well, we have added **Supplementary Table 4**, which gives the AUCs and adjusted p values for this Hallmark GSEA analysis in PURE01, and refer to this new table at the end of the sentence (lines 154-6).

Figure R4.8. Relationships of AUC and padj for CERNO tests on Hallmark gene sets for a) PURE01 and b) ABACUS pre-treatment cohorts. See **Figures 1f** and **2b**.

- For “...[results]...were relatively concordant” is in a sentence that starts at line 276, and that we have modified to: “To distinguish the unique proteomic profiles in the TME that could contribute to CPI outcome, we used principal component analysis (PCA) to capture the differences between the CR and NR samples from the luminal subtypes S1 and S2, and from S4 (**Fig. 4c**). PCA separated subtypes for non-responder samples more effectively than for complete responders, suggesting that non-responders in these subtypes had greater inter-TME heterogeneity.”
 - The upper part of **Fig. 4c** shows PC1-PC2 scatterplots for CR and NR TME AOIs from the DSP samples from S1, S2, and S4. Methods text describes how the calculations were done, and we provide data + code for **Fig. 4c**. The manuscript text states the goal of the calculation, and the three subtypes are clearly more separate for NR samples than for CR samples. In the Discussion and our response document, we state that we are aware that we have limited DSP data, and we have toned down statements related to the DSP work (lines 292-4). However, our results suggest that continued work with spatial data could be informative.

- Statements with “were enriched for” occurs in several locations.
 - Starting at line 149, “S3 tumors were enriched for basal (Ba/Sq) tumors by cMIBC and MDA.” We have added **Supplementary Table 2**, which compares our PURE01 subtypes and classifier calls from four alternative systems. Fisher’s Exact tests, with a Bonferroni correction: Lund (5-class, Methods): $p_{\text{adj}} = 1.2 \times 10^{-17}$. TCGA mRNA: $p_{\text{adj}} = 3.5 \times 10^{-17}$; cMIBC subtypes: $p_{\text{adj}} = 4.4 \times 10^{-17}$; MDA: $p_{\text{adj}} = 5.0 \times 10^{-12}$.
 - Starting at line 223: “S1, the subtype with the worst pathologic response, was enriched for mutually exclusive mutations in FGFR3 (35%) and KRAS (23%), and amplifications in PPARG (23%) (**Supplementary Fig. 6a**).”
 - Starting at line 228: “Subtypes with high response rates (S2 and S3) were enriched for mutations in ATR (S2: 33%, S3: 16%); and TP53, the most frequently reported mutation in TCGA¹¹, was also frequently mutated in these subtypes (S2: 73% and S3: 68%).”
- To respond to the request to provide statistical support where we claim correlations, we have done the following:
 - Lines 251-3: “By ESTIMATE and MCP-counter, S5 tumors had higher ImmuneScores and StromalScores, and higher levels of fibroblasts and other immune cell types, confirming the immune-excluded phenotype observed by histology (**Supplementary Fig. 8b**).”
 - Lines 317-23: “By comparing immune deconvolution for the matched-pair tumor samples, we identified that post-treatment tumors had increased immune and stromal populations, and decreased tumor purity, with all but three tumors demonstrating an increase in stromal score and all but two with an increase in immune score (**Fig. 5g; Supplementary Tables 9,14**). Specifically, post-treatment samples showed an increase in fibroblast populations (**Supplementary Fig. 8a-c; Supplementary Table 15**).”

R4.9. DSP analysis was done on just 5 samples. Here descriptive results are all that can be used in the absence of a more robust sample set. The authors try to draw conclusions around the largely concordant findings in the responders, and the discordant findings in the non-responders, and then further describe responders and non-responders within subtypes S1, S2 and S4. However, with only 5 samples these aren’t meaningful. This analysis should be done with more depth (more samples), omitted from this paper and published separately so as not to delay publication of the key findings.

We appreciate the reviewer’s thoughtful comments. We have toned down the conclusions from the DSP data, and have added a limitation statement to the discussion regarding the number of samples and DSP areas-of-interest.

- Old: In summary, we profiled pre-treatment immune cell types within each subtype by immune deconvolution of bulk RNA-seq data. Evaluation of spatial proteins of the TMEs provided finer-grained results and suggested that each subtype may have a distinct immune TME composition that may influence response to pembrolizumab.
- New (lines 292-4): In summary, we profiled pre-treatment immune cell types within each subtype by immune deconvolution of bulk RNA-Seq data with further evaluation by spatial protein data of TMEs. Subtype comparisons identified differences in the immune microenvironment.
- New (lines 448-50): Our spatial proteomic evaluation involved areas of interest from a limited number of tumors; further investigation of spatial heterogeneity should be performed in MIBC.

R4.10. Pretreatment tumors were compared with post treatment tumors by the subsets defined in this paper. Most of the post treatment samples didn't fit one of the 5 predefined subtypes, but landed in one of two new subtypes S6 and S7. Samples from these two post treatment (resistant) subsets are then re-analyzed by GSEA and CMap. While interesting, more robust conclusions could be drawn from these analyses were they applied to pre/post comparison from samples in ABACUS and as a control pre/post from chemo treated and non-treated samples to determine if the subtype plasticity is artifact or real. Because doing this would require more samples and experiments, this reviewer would recommend removing the pre/post analysis from this manuscript, expanding the analyses to more samples including controls, and submitting those results separately for publication as they tell an important but not directly related story.

We again appreciate the reviewer's thoughtful evaluation of the manuscript. To date, no study has really evaluated how the biology of tumors changes during immunotherapy. We think this has direct translational importance to how we should consider post-CPI-treated tumors. Yet, we agree the number of samples is limited. We have toned down conclusions drawn from these analyses (lines 292-4) and have added limitations to the Discussion (lines 448-50).

R4.11. The discussion of FGFR2 and KDM5B as targets for S1 tumors is interesting but again puts a close focus on these new subtypes defined using few samples and not statistically validated. I am not sure focus on the S1 subset is of high enough relevance to include in this manuscript. If included, please clarify how many of the S1 tumors analyzed were from PURE01 and how many from ABACUS.

Subtype S1 involves 26 PURE01 patients, which is a relatively large subset (32%) of the n=82 pre-treatment cohort (**Fig. 1a**). The work reported here focused on this subtype, because it had a strikingly poor response to pembrolizumab (**Fig. 1b**). We sought to demonstrate the overall utility of identification of molecular features of the cohort, and how those features could potentially be used in the future to approach a biomarker-defined cohort of therapy-resistant cancers.

As we discussed for R1.5 and R4.5, we compared diverse properties of S1 samples in PURE01 to predicted S1 samples in ABACUS. Both subtypes showed similar gene set enrichments (**Figs. 1f, 2b**), PD-L1 expression (**Fig. 2d**), poor response to pembrolizumab and atezolizumab (**Figs. 1b, 2e**), top-ranked CMap perturbations (**Figs. 1h, 2c**), and statistically higher frequencies of FGFR3 somatic mutations and CCND1 amplifications (**Supplementary Fig. 6a**) (lines 227-8). Taken together, these results demonstrate that the PURE01 classifier identified S1-like samples in the ABACUS cohort.

R4.12. The cell line work seems tangential to the central hypothesis of the paper. It is not clear how many cell lines were used or how they are "S1-like." Erdafitinib is already in clinical use in a biomarker defined subset of patients. It's not clear how KDM5 inhibition would be therapeutic and prior experience with similar agents in bladder cancer clinical trials is not discussed. Rather than go back to flesh out and strengthen this analysis which does not relate to the central hypothesis, nor leverage the PURE01 or ABACUS samples, this reviewer would recommend omitting it here and publishing it separately.

We used the RT4 bladder cancer cell line in this report. Our classifier reported that RT4's RNA-Seq data was S1-like (**Fig. R4.12**).

Condition	prediction	class_name	probability_s1	probability_s2	probability_s4	probability_s5	probability_s3
DMSO	S1	LumE	0.50	0.20	0.30	1.0E-05	2.7E-04
DMSO	S1	LumE	0.49	0.20	0.32	1.0E-05	2.7E-04
DMSO	S1	LumE	0.51	0.19	0.31	1.0E-05	2.7E-04
C70	S1	LumE	0.67	0.16	0.17	1.1E-05	8.8E-04
C70	S1	LumE	0.67	0.15	0.19	1.2E-05	1.0E-03
C70	S1	LumE	0.70	0.14	0.15	1.1E-05	8.5E-04

Figure R4.12. Subtype classification of RT4 cell line RNA-Seq data.

We apologize to the reviewer that we did not make the translational importance of KDM5B clearer. We compared the response of KDM5Bi to erdafitinib *because* erdafitinib is already FDA-approved. While the similarities between results for KDM5Bi and erdafitinib (e.g. **Fig. 6e**) suggest that a KDM5Bi could be as effective as erdafitinib, basing treatment on RNA expression for KDM5B rather than on FGFR3 genetic alteration would potentially reach three times more patients. We have added this information to the discussion (lines 429-51).

Minor:

R4.13. Page 5, last 3 lines. Two year RFS numbers seem incorrect – the authors probably meant the. The proportions seem to describe rate of recurrence at 2 years not proportion recurrence free at 2 years.

We thank the reviewer for noting this issue. We have changed the manuscript text on (lines 127-9) to now read:

Recurrence rates at 2 years were 27% for S1 and 38% for S4, while the subtypes with the best response rates had low two-year rates of recurrence of 7% for S2 and 5% for S3.

R4.14. The acronym GSEA is not defined in the manuscript.

We have adjusted the text (line 155) to read: “To characterize the biologic features of each subtype, we performed Gene Set Enrichment Analysis (GSEA) using”

REVIEWER COMMENTS

Reviewer #1 (Remarks to the Author):

The authors have revised the manuscript according to the comments. However, there are several additional concerns that need to be addressed before the publication.

1. As described by authors and others, the immune infiltration is one of the important biomarkers to predict the response to immunotherapy, please include an additional column in Fig 4a to indicate which one is responder or non-responders and arrange the samples based on responses in each subtype.
2. Authors investigated the association between mutations in DDR and the immunotherapy response in Fig 3. Since the mutations in ERCC2 are associated with the response to immunotherapy, please also include mutation status of ERCC2 in this analysis.

Reviewer #3 (Remarks to the Author):

The authors solved most of my concerns, including data accessibility, evaluation, and concept clarification. However, I still have six comments on the cancer type classifier, spatial proteomics, and scRNA-seq regulon analysis. In the attached document, the black sentences are previous comments. The blue sentences are the responses to these comments. The red sentences are the remaining comments.

The authors solved most of my concerns, including data accessibility, evaluation, and concept clarification. However, I still have six comments on the cancer type classifier, spatial proteomics, and scRNA-seq regulon analysis. The black sentences are previous comments. The blue sentences are the responses to these comments. The red sentences are the remaining comments.

R3.4. In the Method section (single-sample expression subtype classifier), first, please introduce the rationale why choose GLMnet as a classifier basis. Second, could the author provide more results regarding the model training regarding parameter tuning because there are lots of parameters (or hyperparameters) to determine the final model? A table showing parameters' combination and results (i.e., show precision, recall, or accuracy) will be helpful. Third, the authors used two normalization methods as input to train the classifier based on GLMnet. What is the rationale here to compare the two normalization methods? These two normalization methods push me to think about the previous GSEA analysis, whether it has two results?

GLMnet Elastic nets are generalized linear models with regularization. While they are normally used in regression contexts, they can be easily adapted to classification tasks using binomial family models, and the R package caret includes this possibility by default. Indeed, we included them because previous publications in our field had used this type of classifier (Seiler et al, 2017; de Jong 2019). We also tested a random forest (RF) model and a simpler nearest-centroid (NSC) model, all in 3-fold repeated crossvalidation runs. Consistently, the best-tuned elastic net models outperformed the random forest models. The elastic net classifier's cross-validation accuracy was 92%. Given that we lack the ground-truth labels for ABACUS subtypes, we are restricted to assessing accuracy with cross-validation metrics.

De Jong JJ, Liu Y, Robertson AG, Seiler R, Groeneveld CS, et al. Long non-coding RNAs identify a subset of luminal muscle-invasive bladder cancer patients with favorable prognosis. *Genome Med.* 2019 Oct 17;11(1):60. Doi: 10.1186/s13073-019-0669-z. PMID: 31619281.

Seiler R, Ashab HAD, Erho N, van Rhijn BWG, Winters B, et al. Impact of Molecular Subtypes in Muscle-invasive Bladder Cancer on Predicting Response and Survival after Neoadjuvant Chemotherapy. *Eur Urol.* 2017 Oct;72(4):544-554. Doi: 10.1016/j.eururo.2017.03.030. PMID: 28390739.

I appropriate the additional efforts regarding model selection. I also suggest the author should disclose these details in the manuscript (e.g., before line 180 or Methods section).

Parameter tuning In a new **Supplementary Fig. 3**, panels (a-c) show parameter tuning metrics, and, for the final best models, panel (d) shows accuracy and Cohen's kappa coefficient. From these results we chose to use a tuned elastic net model. The regularizations in the elastic net model have properties that may explain its good performance for this task: it can select features through ridge penalization, and it will tend to group highly correlated predictors.

The authors should provide p-values in panel d to demonstrate GLMnet is better than the other two models.

Two normalization methods Fig. 1a's heatmap was generated using variance-stabilizing (VST) normalization, and we have modified the Methods text to reflect this (lines 809-10). In contrast, for classifier training and testing, we compared the normalization that we used in all of the analyses with variance-stabilizing (VST) normalization, as suggested by Bioconductor workflows. For subtype classification we found no benefit in using VST normalization; instead, we found that VST led to a small drop in accuracy in all models (accuracy, best GLMnet model: 89.7% with VST vs 92% with log-FPKM). Our intent was to find robust classifier models, and we tested alternate

normalizations to gauge whether the choice of normalization would greatly change accuracy. Ultimately, for consistency, we used the FPKM normalization for classification.

I agree to keep the consistency of using one normalization across all analyses. But it is still confusing why to use VST for heatmap and FPKM for the training model from lines 809-810. For the small drop issue regarding using VST, if the author expected to draw a conclusion that VST led to a small drop compared with FPKM, a statistical significance test is necessary. Otherwise, the entire paper should not mention VST.

GSEA and normalization We used VST-normalized gene expression data in only one analysis: differentially expressed genes for the heatmap in Fig. 1a. As described above, for classifier training and testing we evaluated VST-normalized expression data, but did not use it to generate the results that we report. We ran GSEA analysis only for FPKM-normalized, and not for VST normalized gene expression; consequently, we have only one set of (FPKM-based) GSEA results. As we stated above, ultimately, we did not use the VST-based classifier for the ABACUS validation cohort.

Using two preprocessing steps for the same data in one paper is confusing (unless the author could provide a reference to support). For example, I am confused why the author keeps the discussion in lines 1063-1068 without providing concrete evidence (figure or table). Does that mean the model is sensitive to the normalization method? If so, what about other normalization methods?

R3.7. Regarding the result section (Subtype differences in the tumor microenvironment by bulk and spatial analysis), using deconvolution for bulk-RNAseq data could distinguish the dominant cell types in TME, but the cell types in deconvolution results still show a low granularity. For example, [MCP-counter] cell types include CD8 T cells, B cells, overall T cells, etc. This may be another reason to perform DSP spatial proteomics experiments (71 proteins) and analyses to show the real cells' colocalization and higher granularity of cell composition. However, the spatial analysis is not eye-catching (the take-home message is not clear to me), and where do the other proteins go (Figure 4d shows 13 proteins)?

The reviewer asks for striking differences in the TMEs of the subtypes. In results from DSP data for subtypes S1, S2 and S4 – PCA in Fig. 4c, and expression of immune-regulatory proteins in Fig. 4d – we find large differences between TMEs of non-responders. We suggest that these spatial findings further support our hypothesis that spatial differences in immune regulation between subtypes may explain the response to pembrolizumab. At the same time, we agree that spatial analysis in immunotherapy is a relatively new field, and we hope to gain further insight with more samples and further analysis in the future analysis with RNA and protein data. Given the limited number of samples assessed by DSP, we have toned down the statements for this part of the work (lines 292-4), and acknowledge the constrained samples in a limitations section in the Discussion (lines 440-51).

To respond to the reviewer's request to indicate 'where the other proteins go', we did the following analysis. **Method.** We read in SNR-normalized DSP data for 71 proteins from tumor microenvironment (TME) areas-of-interest (AOIs, Fig. 4b), and for three negative controls (rabbit IgG, and mouse IgG1 and IgG2a). Because proteins whose abundances are comparable to or below the levels of negative controls are less reliable as indicators of biology, we assessed Spearman correlations between all proteins whose median levels were above the median level for the most-abundant negative control, rabbit (Rb) IgG. For the 17 TME AOIs for one NR sample

from each of subtypes S1, S2, and S4, and the 21 TME AOIs for one CR sample from each of these three subtypes, we used Spearman correlations in R's `corrplot()` function to assess relationships between 35 NR and 42 CR SNR-normalized protein levels. We then annotated the correlation heatmaps using functional 'probe group' information (NanoString, Seattle WA).

We removed approximately 35 to 40% of the 71 TME proteins in NR and CR samples that were expressed at levels that were comparable to or below the negative control Rb IgG (Figs. R3.7a,b). For NR samples, this filtering removed Treg marker proteins CD25 and FOXP3. For both CR and NR samples, filtering removed cancer/testis (CT) antigen markers MART1 and NY-ESO-1, hormone receptor markers ER-alpha and PR, and the neutrophil marker CD66b.

For TME AOIs from one sample each of subtypes S1, S2, and S4 (Fig. 4b), we assessed correlations for the remaining proteins. Samples in these three subtypes had clustered together in both n=82 and n=113 runs (Fig. 5a), suggesting that the biologies of these subtypes were more distinct than those of S5 and S3. We separated CR samples from NR samples, as we did for PCA and the levels of 13 TME proteins in these subtypes (Figs. 4c,d). The correlation heatmaps differed between CR and NR samples, so were generally consistent with Figs. 4c,d).

The figures (pairwise correlation and corresponding cell types) R3.7e-f are less informative, I cannot obtain further information as the author described ("The correlation heatmaps differed between CR and NR samples, so were generally consistent with Figs. 4c,d"). I am not convinced by "generally consistent" without quantitative justification. Therefore, the author should clarify spatial proteomics for validation in the abstract and introduction section.

R3.8. For the last result section (FGFR3 and KDM5B are potential LumE/S1 subtype targets), it used bulk-seq to identify the potential regulator and then uses scRNA-seq & ATAC-seq to validate KDM5B's role. The author could try to use SCENIC (python package) or IRIS3 (webserver) to further investigate the regulon network at the single-cell level with an advanced computational framework and validate the findings.

We appreciate Reviewer #3's recommendation to use SCENIC for regulon analysis of our single-cell RNA-Seq data set. RTN was developed using bulk RNA-Seq data. In such data, it analyzes the expression of target genes that are regulated by transcription factors in a mixture of multiple cell types. Regulon analysis in single-cell RNA-Seq data has the potential to inform on regulatory relationships for specific cell types.

To evaluate the KDM5B regulon activity in scRNA-Seq data, we first used pySCENIC to identify clusters of similar cell types, producing a UMAP clustering (Fig. 6c) (Methods, starting at line 1069). We then reclustered only the epithelial cells (Fig. 6c, and found that luminal marker genes were relatively highly expressed in six of the nine epithelial cell clusters (ii). Consistent with RTN results from bulk RNA-Seq data (Fig. R3.8, Fig. 1b, and Supplementary Fig. 10b). KDM5B regulon activity was activated (i.e. strongly positive) in the subset of luminal epithelial cells in which KDM5B expression was high (iii, iv), and repressed (i.e. strongly negative) in luminal epithelial cells in which KDM5B expression was low (iii,v).

I appreciate the additional effort for analyzing scRNA-seq regulon. However, the description of scRNA-seq analysis was oversimplified. (1) what is the Roman number in the parathesis from the author's response? (2) The evidence that "luminal marker genes were relatively highly expressed in six of the nine epithelial cell clusters" is not solid. The author should calculate the correlation coefficient to make this conclusion. (3) I am curious about how the five major tumor types correspond to nice sub-clusters from epithelial cells in terms of KDM5B regulon activity?

Reviewer #4 (Remarks to the Author):

R4.1 (and R1.5) The subtypes were not reproducible across ABACUS and PURE01 in terms of response rate – which is the most clinically relevant variable. Specifically the S2 and S3 subtypes have high response rates in PURE01 but not in ABACUS. (Figure R4.5). This is a fatal flaw in this manuscript, as all analyses link biology to subtype and infer a correlation with response based on the subtype to response relationship in the PURE01 set. A relationship that is not recapitulated in ABACUS. The requested re-analysis of the results to define the variability of biologic markers across the spectrum of response was not performed. In R1.5 the authors state “This clinical response may be separate than the biologic activity of the subtypes” which is antithetical to the rest of the paper claiming a biologic link to subtypes and hence response.

R1.8 The reviewer is requesting a cross comparison of the mutation analysis across the 2 trials. The authors rebuttal describes comparison of S1 vs not S1 within each trial set separately.

R4.8 Defer to the editor and statistical reviewer as to whether claims are appropriately substantiated statistically and whether a simple parenthetical statistical test result would improve readability over accessing the statistical analysis within the supplement.

R4.9 The DSP analysis and cell line work is not relevant nor of high enough caliber to warrant being included in this publication. I suggested these sections be omitted keeping the focus on the primary sample based analyses. Toning down the conclusions and describing the translational relevance of KDM5B does not adequately address this. This manuscript is too long and too far reaching as is. Paring it down is required and these tangential analyses not related to the samples are a good place to cut.

R1.4 OS data has been published for ABACUS. It is not clear why it could not be provided for this publication.

RESPONSE TO REVIEWERS' COMMENTS

Due to the number of reviews, we have attempted to identify our response to each comment with an (AU)

Reviewer #1

1. As described by authors and others, the immune infiltration is one of the important biomarkers to predict the response to immunotherapy, please include an additional column in Fig 4a to indicate which one is responder or non-responders and arrange the samples based on responses in each subtype.

(AU) The Reviewer asks for a comparison of response to immune infiltration for Fig 4a. We have used ESTIMATE's ImmuneScore as an estimate of overall immune infiltration. We do not observe a significant correlation between response and immune score **Fig 1 (below)**. The below figure shows the immune score for each subtype, comparing complete responder (CR) to non-responder (NR). The results show that immune infiltration may be a predictive biomarker of response for tumours of subtype S4, but is unlikely to be a predictive biomarker for the other subtypes (such as S1, S2, S3 and S5). We have added this result to the manuscript as **Supplementary Fig 7d**.

Figure 1. Distributions of ESTIMATE's ImmuneScore for CR and NR samples in each PURE01 subtype.

2. Authors investigated the association between mutations in DDR and the immunotherapy response in Fig 3. Since the mutations in ERCC2 are associated with the response to immunotherapy, please also include mutation status of ERCC2 in this analysis.

(AU) Unfortunately, ERCC2 is not part of the targeted panel at Foundation Medicine and was not evaluated in the tumors (<https://www.foundationmedicine.com/test/foundationone-cdx>).

Reviewer #3 (Remarks to the Author):

The authors solved most of my concerns, including data accessibility, evaluation, and concept clarification. However, I still have **six comments** on the cancer type classifier, spatial proteomics, and scRNA-seq regulon analysis. In the attached document, the **black** sentences are previous comments. The **blue** sentences are the responses to these comments. The **red** sentences are the remaining comments.

[previous comments] R3.4. In the Method section (single-sample expression subtype classifier), first, please introduce the rationale why choose GLMnet as a classifier basis. Second, could the author provide more results regarding the model training regarding parameter tuning because there are lots of parameters (or hyperparameters) to determine the final model? A table showing parameters' combination and results (i.e., show precision, recall, or accuracy) will be helpful. Third, the authors used two normalization methods as input to train the classifier based on GLMnet. What is the rationale here to compare the two normalization methods? These two normalization methods push me to think about the previous GSEA analysis, whether it has two results?

[responses to previous comments] GLMnet Elastic nets are generalized linear models with regularization. While they are normally used in regression contexts, they can be easily adapted to classification tasks using binomial family models, and the R package caret includes this possibility by default. Indeed, we included them because previous publications in our field had used this type of classifier (Seiler et al, 2017; de Jong 2019). We also tested a random forest (RF) model and a simpler nearest-centroid (NSC) model, all in 3-fold repeated cross-validation runs. Consistently, the best-tuned elastic net models outperformed the random forest models.

The elastic net classifier's cross-validation accuracy was 92%. Given that we lack the ground-truth labels for ABACUS subtypes, we are restricted to assessing accuracy with cross-validation metrics.

De Jong JJ, Liu Y, Robertson AG, Seiler R, Groeneveld CS, et al. Long non-coding RNAs identify a subset of luminal muscle-invasive bladder cancer patients with favorable prognosis. *Genome Med.* 2019 Oct 17;11(1):60. Doi: 10.1186/s13073-019-0669-z. PMID: 31619281.

Seiler R, Ashab HAD, Erho N, van Rhijn BWG, Winters B, et al. Impact of Molecular Subtypes in Muscle-invasive Bladder Cancer on Predicting Response and Survival after Neoadjuvant Chemotherapy. *Eur Urol.* 2017 Oct;72(4):544-554. Doi: 10.1016/j.eururo.2017.03.030. PMID: 28390739.

[3.1. remaining comments] I appropriate the additional efforts regarding model selection. I also suggest the author should disclose these details in the manuscript (e.g., before line 180 or Methods section).

(AU) We thank the reviewer for this suggestion. We have added the 'GLMnet elastic net' text (above) to the start of the Methods' section for 'Single-sample expression subtype classifier'.

[responses to previous comments] Parameter tuning In a new Supplementary Fig. 3, panels (a-c) show parameter tuning metrics, and, for the final best models, panel (d) shows accuracy and Cohen's kappa coefficient. From these results we chose to use a tuned elastic net model. The regularizations in the elastic net model have properties that may explain its good performance for this task: it can select features through ridge penalization, and it will tend to group highly correlated predictors.

[3.2. remaining comments] The authors should provide p-values in [Fig 3]d to demonstrate GLMnet is better than the other two models.

- (AU) **Supplementary Fig. 3d**'s legend is: "Accuracy and Cohen's kappa for the best GLMnet, NSC, and RF models, as a function of the number of classifier features (i.e. genes). Vertical bars indicate standard deviations of metrics calculated in 3-fold repeated cross validation." For the classifier performance issue addressed, the legend states what the vertical bars indicate, and panel 'd' clearly shows that a GLMnet classifier had better performance than either a nearest-centroid (NSC) or random forest (RF) classifier. In the first paragraph of the Methods text for 'Single-sample expression subtype classifier', which we have added in response to the reviewer's suggestion, we now refer to this panel: "Consistently, the best-tuned elastic net models outperformed both alternative models (**Supplementary Figure 3d**)."
 - In our view, this statement is adequately supported by panel **d**, without p values in either the text or figure panel.

[responses to previous comments] Two normalization methods Fig. 1a's heatmap was generated using variance-stabilizing (VST) normalization, and we have modified the Methods text to reflect this (lines 809-10). In contrast, for classifier training and testing, we compared the normalization that we used in all of the analyses with variance-stabilizing (VST) normalization, as suggested by Bioconductor workflows. For subtype classification we found no benefit in using VST normalization; instead, we found that VST led to a small drop in accuracy in all models (accuracy, best GLMnet model: 89.7% with VST vs 92% with log-FPKM). Our intent was to find robust classifier models, and we tested alternate normalizations to gauge whether the choice of normalization would greatly change accuracy. Ultimately, for consistency, we used the FPKM normalization for classification.

[3. remaining comments] I agree to keep the consistency of using one normalization across all analyses. But it is still confusing why to use VST for heatmap and FPKM for the training model from lines 809-810. For the small drop issue regarding using VST, if the author expected to draw a conclusion that VST led to a small drop compared with FPKM, a statistical significance test is necessary. Otherwise, the entire paper should not mention VST.

(AU) We apologize for the confusion over two RNA-Seq normalization methods. We have changed **Fig. 1a**'s heatmap to use FPKM RNA-Seq data, and, as suggested by the reviewer, we have eliminated VST from the manuscript, except for noting the text below in the Methods section under "Single-patient classifier" that VST-normalization did not improve classifier performance:

For comparison, we trained 100-gene and 500-gene GLMnet and random forest classifiers with VST-transformed [50] PURE01 RNA-Seq data in the same manner as described previously and used these classifiers to predict on VST-transformed ABACUS RNA-Seq data. We found that the classifier did not benefit from our using VST normalization [Cross-validation accuracy: log₂(FPKM) = 92.1% (95% CI: 83.2 - 100%), VST = 89.7% (78.8 - 100%); Cross-validation kappa: log₂(FPKM) = 86.7% (95% CI: 71.8 - 100%), VST = 89.7% (78.2 - 100%)]. Given this, we recommend using log₂(FPKMs) with the GLMnet classifier, rather than VST-transformed RNA-Seq data.

[responses to previous comments] GSEA and normalization We used VST-normalized gene expression data in only one analysis: differentially expressed genes for the heatmap in Fig. 1a. As described above, for classifier training and testing we evaluated VST-normalized expression data, but did not use it to generate the results that we report. We ran GSEA analysis only for FPKM-normalized, and not for VST normalized gene expression; consequently, we have only one set of (FPKM-based) GSEA results. As we stated above, ultimately, we did not use the VST-based classifier for the ABACUS validation cohort.

[4. remaining comments] Using two preprocessing steps for the same data in one paper is confusing (unless the author could provide a reference to support). For example, I am confused why the author keeps the discussion in lines 1063-1068 without providing concrete evidence (figure or table). Does that mean the model is sensitive to the normalization method? If so, what about other normalization methods?

(AU) 3.4 The color palette in a clustering heatmap reflects normalized gene expression levels that have been scaled so that expression levels translate into colors that are informative. To see how gene expression varies across a heatmap, each gene must be scaled in order to put gene variation into the same color space. Without scaling, a gene with low expression across samples might not be clearly visible, even if its expression is informative. While clustering heatmaps show general patterns, many persons are relatively unfamiliar with issues involved in normalization of gene expression data (e.g. Evans 2018) and scaling for heatmaps.

- Evans C, Hardin J, Stoebel DM. **Selecting between-sample RNA-Seq normalization methods from the perspective of their assumptions**. Brief Bioinform. 2018;19(5):776-792. PMID: 28334202.

In the original manuscript, we described how we calculated gene expression as FPKMs (see the last paragraph of the Methods section on 'RNA sequencing'). We mentioned VST only in the Methods section for 'Consensus expression subtypes', for **Fig. 1a**'s clustering heatmap. In the current revised manuscript, at the start of the Methods section on the 'Single-sample expression subtype classifier', we note that we compared classifier performance for log₂-transformed FPKMs and for VST-transformed RNA-Seq data, and we state clearly that "...we recommend using log₂(FPKMs) with the GLMnet classifier, rather than VST-transformed RNA-Seq data." In our view, we have clarified how RNA-Seq normalization was handled in this work.

Assessing 'other normalization methods' is beyond the scope of work for the current manuscript.

Reviewer: "For example, I am confused why the author keeps the discussion in lines 1063-1068 without providing concrete evidence (figure or table)."

(AU) For comparison, we trained 100-gene and 500-gene GLMnet and random forest classifiers with VST-transformed [50] PURE01 RNA-Seq data in the same manner as described previously and used these classifiers to predict in VST-transformed ABACUS RNA-Seq data. We found that the VST classifiers predicted far more S3 samples than classifiers trained and tested with log₂ FPKM data and appeared to misclassify S1 and S2 samples as class S3. Given this, we recommend using log₂(FPKMs) with the GLMnet classifier, rather than VST-transformed data.

In our revised manuscript, we responded to R3.4 by adding **Supplementary Fig 3**, which reports on classifier training and performance. However, we supplied no figure or table to support the following

statement: “VST classifiers predicted far more S3 samples than classifiers trained and tested with log₂ FPKM data and appeared to misclassify S1 and S2 samples as class S3”.

Because we lack ground truth labels for the expression subtypes, it is unclear what statistical test to apply. We suggest using confidence intervals of the metrics calculated in cross-validation (**Table 1**). Given these results, we have simplified the manuscript statement to: “We found no benefit in using VST normalization in training and testing the single-sample classifier” in the “Single patient classifier” section of the Methods. However, given the cross-validation accuracy, we have left the final sentence in that paragraph unchanged: “Given this, we recommend using log₂(FPKMs) with the GLMnet classifier, rather than VST-transformed RNA-Seq data.”

Table 1. Cross-validation accuracy and kappa, with 95% confidence intervals, for the 100-gene classifier, for log₂(FPKM) and VST-normalized RNA-Seq data.

	Log ₂ (FPKM)	VST
Accuracy	92.1% (83.2 - 100%, 95% CI)	89.7% (78.8 - 100%, 95% CI)
Kappa	86.7% (71.8 - 100%, 95% CI)	89.7% (78.2 - 100%, 95% CI)

[previous comments] R3.7. Regarding the result section (Subtype differences in the tumor microenvironment by bulk and spatial analysis), using deconvolution for bulk-RNAseq data could distinguish the dominant cell types in TME, but the cell types in deconvolution results still show a low granularity. For example, [MCP-counter] cell types include CD8 T cells, B cells, overall T cells, etc. This may be another reason to perform DSP spatial proteomics experiments (71 proteins) and analyses to show the real cells' colocalization and higher granularity of cell composition. However, the spatial analysis is not eye-catching (the take-home message is not clear to me), and where do the other proteins go (Figure 4d shows 13 proteins)?

[responses to previous comments] The reviewer asks for striking differences in the TMEs of the subtypes. In results from DSP data for subtypes S1, S2 and S4 – PCA in Fig. 4c, and expression of immune-regulatory proteins in Fig. 4d – we find large differences between TMEs of non-responders. We suggest that these spatial findings further support our hypothesis that spatial differences in immune regulation between subtypes may explain the response to pembrolizumab. At the same time, we agree that spatial analysis in immunotherapy is a relatively new field, and we hope to gain further insight with more samples and further analysis in the future analysis with RNA and protein data. Given the limited number of samples assessed by DSP, we have toned down the statements for this part of the work (lines 292-4), and acknowledge the constrained samples in a limitations section in the Discussion (lines 440-51).

To respond to the reviewer's request to indicate ‘where the other proteins go’, we did the following analysis. Method. We read in SNR-normalized DSP data for 71 proteins from tumor microenvironment (TME) areas-of-interest (AOIs, Fig. 4b), and for three negative controls (rabbit IgG, and mouse IgG1 and IgG2a). Because proteins whose abundances are comparable to or below the levels of negative controls are less reliable as indicators of biology, we assessed Spearman correlations between all proteins whose median levels were above the median level for the most-abundant negative control, rabbit (Rb) IgG. For the 17 TME AOIs for one NR sample from each of subtypes S1, S2, and S4, and the 21 TME AOIs for one CR sample from each of these three subtypes, we used Spearman correlations in R's corplot() function to assess relationships between 35 NR and 42 CR SNR-normalized protein levels. We then annotated the correlation heatmaps using functional ‘probe group’ information (NanoString, Seattle WA).

We removed approximately 35 to 40% of the 71 TME proteins in NR and CR samples that were expressed at levels that were comparable to or below the negative control Rb IgG (Figs. R3.7a,b). For NR samples, this filtering removed Treg marker proteins CD25 and FOXP3. For both CR and NR samples, filtering removed cancer/testis (CT) antigen markers MART1 and NY-ESO-1, hormone receptor markers ER-alpha and PR, and the neutrophil marker CD66b.

For TME AOIs from one sample each of subtypes S1, S2, and S4 (Fig. 4b), we assessed correlations for the remaining proteins. Samples in these three subtypes had clustered together in both n=82 and n=113 runs (Fig. 5a), suggesting that the biologies of these subtypes were more distinct than those of S5 and S3. We separated CR samples from NR samples, as we did for PCA and the levels of 13 TME proteins in these subtypes (Figs.

4c,d). The correlation heatmaps differed between CR and NR samples, so were generally consistent with Figs. 4c,d).

[5. remaining comments] The figures (pairwise correlation and corresponding cell types) R3.7e-f are less informative, I cannot obtain further information as the author described ("The correlation heatmaps differed between CR and NR samples, so were generally consistent with Figs. 4c,d"). I am not convinced by "generally consistent" without quantitative justification. Therefore, the author should clarify spatial proteomics for validation in the abstract and introduction section.

(AU) However, the spatial analysis ... take-home message is not clear to me...

The take-home message is: In **Fig. 4**, for TME AOIs from S1, S2, and S4, both the PCA analysis of TME AOIs (**Fig. 4c**) and the analysis of immune cell markers and immune regulatory proteins (**Fig. 4d**) were divergent in nonresponders (NR) being larger than of responders (CR).

- **Figure R3.7** was a response to the last part of Reviewer #3's original review question: "... where do the other proteins go (Figure 4d shows 13 proteins)?"

While the reviewer points to **Fig. 4c,d**, **Fig. R3.7** is unrelated to **Fig. 4c,d**. **Fig. R3.7** uses SNR-normalized protein levels from NR and CR areas-of-interest (AOIs), so would be from PURE01 subtypes S1, S2, and S4. In contrast, **Fig. 4c** shows ring plots for N0/N1 in S6 and S7, while **Fig. 4d** shows violin plots of Tumor Area for S6 and S7.

'Generally consistent' was in our initial response to reviews. It largely refers to **Fig. 4c**, which shows that, for NR samples, a PCA analysis of TME AOIs separates subtypes S1, S2, and S4. We agree with the reviewer that the response statement would have been more precise had it been limited to **Fig. 4c** only, rather than referring to **Figs. 4c,d**:

"The correlation heatmaps differed between CR and NR samples, so were generally consistent with **Fig. 4c**."

Figure 4c's axes quantify the percent of variation explained by principal components PC1 and PC2, for CR and NR AOIs. For NR AOIs in **Fig. 4c**, S1, S2, and S4 are clearly separate. In the Discussion's 'limitations' paragraph we acknowledge that we generated DSP data for relatively few tumor samples.

We have now added **Figure R3.7** to the manuscript as **Supplementary Figure 8**.

- Reviewer: "the author should clarify spatial proteomics for validation in the abstract and introduction section".

AU) Unfortunately, we are constrained by the abstract's 150-word limit from describing further the goals of spatial proteomics. We have included these results to further inspect and validate the bulk RNA-seq findings at the tumor/TME level. We included the rationale for using the GeoMX DSP platform in the first paragraph of the.

[previous comments] R3.8. For the last result section (FGFR3 and KDM5B are potential LumE/S1 subtype targets), it used bulk-seq to identify the potential regulator and then uses scRNA-seq & ATAC-seq to validate KDM5B's role. The author could try to use SCENIC (python package) or IRIS3 (webserver) to further investigate the regulon network at the single-cell level with an advanced computational framework and validate the findings.

[responses to previous comments] We appreciate Reviewer #3's recommendation to use SCENIC for regulon analysis of our single-cell RNA-Seq data set. RTN was developed using bulk RNA-Seq data. In such data, it analyzes the expression of target genes that are regulated by transcription factors in a mixture of multiple cell types. Regulon analysis in single-cell RNA-Seq data has the potential to inform on regulatory relationships for specific cell types.

To evaluate the KDM5B regulon activity in scRNA-Seq data, we first used pySCENIC to identify clusters of similar cell types, producing a UMAP clustering (**Fig. 6c**) (Methods, starting at line 1069). We then reclustered only the epithelial cells (**Fig. 6c**, and found that luminal marker genes were relatively highly expressed in six of

the nine epithelial cell clusters (ii). Consistent with RTN results from bulk RNA-Seq data (Fig. R3.8, Fig. 1b, and Supplementary Fig. 10b). KDM5B regulon activity was activated (i.e. strongly positive) in the subset of luminal epithelial cells in which KDM5B expression was high (iii, iv), and repressed (i.e. strongly negative) in luminal epithelial cells in which KDM5B expression was low (iii,v).

[3.6. remaining comments] I appreciate the additional effort for analyzing scRNA-seq regulon. However, the description of scRNA-seq analysis was oversimplified. (1) what is the Roman number in the parathesis from the author's response? (2) The evidence that "luminal marker genes were relatively highly expressed in six of the nine epithelial cell clusters" is not solid. The author should calculate the correlation coefficient to make this conclusion. (3) I am curious about how the five major tumor types correspond to nice sub-clusters from epithelial cells in terms of KDM5B regulon activity?

(3.6.1) what is the Roman number in the parathesis from the author's response?

(AU) In the revised manuscript that the reviewer is commenting on, the legend of **Fig. 6c** explains the lowercase Roman numbers in that panel.

(3.6.2) The evidence that "luminal marker genes were relatively highly expressed in six of the nine epithelial cell clusters" is not solid. The author should calculate the correlation coefficient to make this conclusion.

(AU) In Methods for scRNA-seq, the manuscript says: "We generated a luminal gene expression score using scanpy.tl.score_genes and 16 luminal markers: CYP2J2, ERBB2, ERBB3, FGFR3, FOXA1, GATA3, GPX2, KRT18, KRT19, KRT20, KRT7, KRT8, PPARG, XBP1, UPK1A, and UPK2, which are associated with the luminal subtype in muscle invasive bladder cancer [14]." Reference [14] is the 2017 TCGA BLCA publication.

Fig. 6c(ii) reports a 'Luminal signature score' as colors assigned to single cells, and provides a scalebar for interpreting these colours. **Supplemental Fig. 10d** reports "Expression of the top five marker genes in the nine sub-clusters identified within the epithelial cell cluster." We suggest that reporting quantitative results for epithelial cell clusters is unlikely to be more informative than the colours currently shown in **Fig. 6c(ii)**.

- It is unclear how the requested "correlation coefficients" would be informative.

(3.6.3) I am curious about how the five major tumor types correspond to nice [nine?] sub-clusters from epithelial cells in terms of KDM5B regulon activity?

(AU) Epithelial cell sub-clusters are shown on the right side of **Fig. 6c**. However, while **Supplementary Fig. 11b** indicates that "KDM5B regulon activity" could suggest which epithelial cell sub-clusters corresponded to a PURE01 subtype, additional data (e.g. expression of subtype- and sub-cluster-specific marker genes) would be required to increase the confidence with which an epithelial sub-cluster was assigned to a PURE01 subtype. Given current data, we can highlight two factors: the 'luminal' character, and KDM5B regulon activity. In **Fig. 1e**, most samples in PURE01 subtypes 1 and 2 were largely classified as being some type of 'luminal'. In scRNA-seq data for nine epithelial cell sub-clusters, sub-clusters 0, 1, 2, 5, 6 and 8 had relatively high luminal signature scores, so likely corresponded to "major tumor" subtypes 1 and 2. Then, **Supp. Fig. 11b** shows that bulk subtype 1 had the highest median KDM5B regulon activity, but also contained some samples that had relatively low regulon activity. Epithelial sub-clusters 0, 5, 6 and 8 had high KDM5B(+) regulon AUC values, and so were the sub-clusters most likely to correspond to PURE01 subtype S1.

Reviewer #4 (Remarks to the Author) (with R1 comments on R4 statements)

R4.1 (and R1.5) The subtypes were not reproducible across ABACUS and PURE01 in terms of response rate – which is the most clinically relevant variable. Specifically the S2 and S3 subtypes have high response rates in PURE01 but not in ABACUS. (Figure R4.5). This is a fatal flaw in this manuscript, as all analyses link biology to subtype and infer a correlation with response based on the subtype to response relationship in the PURE01 set. A relationship that is not recapitulated in ABACUS. The requested re-analysis of the results to define the variability of biologic markers across the spectrum of

response was not performed. In R1.5 the authors state “This clinical response may be separate than the biologic activity of the subtypes” which is antithetical to the rest of the paper claiming a biologic link to subtypes and hence response.

To validate the new subtypes identified in PURE01, we compared them to the only MIBC cohort with RNA-Seq results available. Yet, the trials are not a perfect match and have fundamental differences (Table 2).

Table 2. Comparison of PURE01 and ABACUS trials.

	# doses	Time therapy (weeks)	Drug	Target	Pre-treatment Stage (cT3+%)	PDL1+ rate	Path Response (complete)	Path Response (downstaging)	2-year DFS
PURE01	3	9	pembrolizumab	Anti-PD1	58%	70%	42%	54%	75%
ABACUS	2	6	atezolizumab	Anti-PDL1	26%	41%	31%	38%	68%

In PURE01, patients received three doses of pembrolizumab (anti-PD1, 9 weeks of therapy) and were cisplatin-eligible, while in ABACUS they received only two doses of atezolizumab (anti-PDL1, 6 weeks of therapy) and the patients were cisplatin-ineligible. *Before treatment, the cohorts were different*, with PURE01 reporting 58% cT3+ or greater clinical stage while only 26% of ABACUS was cT3+. PURE01 was enriched with PDL1 positive tumors (70%) compared to ABACUS (41%). The complete response rate to PURE01 was 42% compared to 31% in ABACUS. Pathologic downstaging to pT1 or less was found in 54% of PURE01 and 38% in ABACUS. These pathologic responses are consistent with cancer recurrence; the two-year disease-free survival was 75% in PURE01 and only 68% in ABACUS. Even the biomarker analysis between the cohorts differed, with response to pembrolizumab associated with TMB and PDL1 in PURE01, and no association described in ABACUS. Thus, at baseline, while the trials are similar in structure, multiple pre-treatment and treatment factors differed between trials, which could affect the clinical outcomes.

In our analysis, we identify the same gene expression patterns between subtypes in each cohort, reflecting the reproducibility of the subtyping method. We now have obtained clinical outcomes of ABACUS and show similar outcomes for two of five subtypes (Figure 2c), with S4 having fast recurrences and 25% of S1 tumors recurring over 24 months. Yet, in ABACUS, the recurrence rate of S2, S3, and S5 have decreased pathologic response to atezolizumab (Figure 2e,f) and worse clinical response, approaching that of S1. We hypothesize that worse outcomes of these specific subtypes (S2, S3, and S5) explain the differences in outcomes of ABACUS in terms of clinical and pathologic response and the differences in response of these specific subtypes may be secondary to the drug (anti-PD1 vs anti-PDL1), duration of treatment, PDL1 status, or the number of doses (3 vs 2).

To further validate our findings, **we now include a third cohort, the adjuvant IMvigor010 cohort of 640 patients with MIBC (NCT02450331)**. In this Phase III MIBC trial, participants were randomized to atezolizumab or placebo after radical cystectomy. When we applied the classifier to cystectomy specimens of the cohort, we found that the “responsive” subtypes (S2, S3 and S5) treated with 12 months of atezolizumab had an improvement in outcomes compared to “unresponsive” subtypes (S1 and S4) ($p=0.04$) (see below). In the observation cohort we identified no difference in recurrence between subtypes. Therefore, two important findings from the inclusion of this cohort is that 1) the subtyping can be applied after cystectomy and may identify patients that can benefit from adjuvant atezolizumab and 2) this large cohort further validates our subtyping method.

Fig. 2. Disease-free survival of patients on atezolizumab (left) and on observation (right).

Treatment arms are grouped by responsive (S2, S3, and S5) vs non-responsive (S1, S4). The groups show significantly different survival in subtypes treated with atezolizumab (log-rank $p(\text{atezo}) = 0.046$ vs $p(\text{obs}) = 0.47$).

We would also expect further clarification on why not all 114 patients from

(Rev1 comment on Rev4) This is the same comment that I had in my R1.5. Authors claim that they found the agreements on the biological properties of each subtype in both PURE01 and ABACUS datasets although they were unable to reproduce the clinical utility of the novel subtypes in independent dataset. As the authors addressed the limitations of the study, I thought that this study could serve as a basis for subtype identification that could predict response to ICI. However, I still think if they can't validate the clinical utility in an independent cohort, that might undermine the significance of the study because authors claim that they identified “Expression-Based Subtypes Define Pathologic Response to Neoadjuvant Immune-Checkpoint Inhibitors in Muscle-Invasive Bladder Cancer”.

(AU) We have now validated our findings in IMvigor010 and included these in Fig. 2g. and Supplementary Fig. 2h.

R1.8 The reviewer is requesting a **cross comparison of the mutation analysis across the 2 trials**. The authors rebuttal describes comparison of S1 vs not S1 within each trial set separately.

(AU) In the second revision, we included the oncoprints from ABACUS (Supp. Fig. 6), and provided a comparison of the most significant mutations for S1, as samples in S1 had the *unique* high frequency of genetic alterations to *FGFR3* and *CCND1*, which have been shown to be associated with poor response to immune checkpoint immunotherapy (Table 3). A complete comparison for the 354 genes in the FM panel across five subtypes in two cohorts is a large undertaking and challenging to perform with our relatively small subtypes. A brief empirical comparison of the two most-highly mutated genes per subtype shows a relatively close comparison with the most mutated genes shared between cohorts.

Table 3. For PURE01 and ABACUS cohorts, per-subtype mutation frequencies of the two most frequently mutated genes in each subtype.

S1	PURE	ABACUS		S4	PURE	ABACUS
KDM6A	50%	32%		TP53	50%	67%
TP53	46%	50%		RB1	38%	33%
S2	PURE	ABACUS		S5	PURE	ABACUS
TP53	73%	78%		TP53	50%	100%
RB1	33%	39%		RB1	29%	33%
S3	PURE	ABACUS				
TP53	68%	69%				
RB1	26%	27%				

(Rev1 comments of Rev4) That's true. Authors described that they limited genes from most frequently altered in S1 by somatic mutations or CNV due to the significance of S1. Two of these genes (FGFR3 and CCND1) were significantly altered. The manuscript will be improved if they can perform the same analysis that they did in figure 3.

(AU) The oncprints from **Figure 3** are included in **Supplementary Fig. 6**.

R4.8 Defer to the editor and statistical reviewer as to whether claims are appropriately substantiated statistically and whether a simple parenthetic statistical test result would improve readability over accessing the statistical analysis within the supplement.

(AU) We thank the reviewer for the comment. Whenever possible, we have attempted to use statistics to support our hypotheses in the manuscript.

R4.9 The DSP analysis and cell line work is not relevant nor of high enough caliber to warrant being included in this publication. I suggested these sections be omitted keeping the focus on the primary sample based analyses. Toning down the conclusions and describing the translational relevance of KDM5B does not adequately address this. This manuscript is too long and too far reaching as is. Paring it down is required and these tangential analyses not related to the samples are a good place to cut.

(AU) Describing a new system of subtyping is theoretical unless it has a direct application. We selected S1 to dig deeper into a subtype to illustrate a potential application into direct patient care because it is a large portion of tumors in both PURE01 and ABACUS and has a poor response to immunotherapy.

(Rev1 comment on Rev4) I agree with this comment. Because the compositions of immune and stroma cells in TME have been associated with the response to CPI, DSP analysis is useful for evaluating the response-related features of TME at the spatial level. On page 9, authors described that in PURE01 data, the subtypes with the highest pathological response rates (S2 and S3) had high level of immune cell populations, consisted with immune-inflamed tumors. However, in ABACUS data, S2 and S3 did not show the high pathological response rates.

(AU) Please see the comments comparing the cohorts and trials in our response to the Associate Editor, and subsequent IMVigor010 data.

The authors also described that S3 is 'immune-inflamed' whereas S5 may be 'immune-excluded' due to the high-level expression of CAFs. However, CAFs expression level in S3 is almost as high as S5 compared to the rest of the subtype. Therefore, DSP analysis may not relevant to this publication. In addition, samples for DSP analysis were selected from 3 luminal subtypes (S1, S2 and S5). Why did the authors select samples only from luminal tumors?

(AU) We limited the selection of samples due to the cost of the GeoMX profiling. We compared responders and non-responders and wanted to perform some subtype comparisons. S1, S2, and S4 had the most similarity by bulk-RNA-Seq and would allow us to explore compartmental differences between tumor and TME of responders and non-responders.

R1.4 OS data has been published for ABACUS. It is not clear why it could not be provided for this publication.

(AU) During the revision, this data was published and we have included it in the updated manuscript (**Fig. 2c**). The explanation for differences between PURE01 and ABACUS has been added to the manuscript and is described in the response to the Associate Editor.

REVIEWER COMMENTS

Reviewer #1 (Remarks to the Author):

The author has addressed most of my concerns. However, there are still some concerns that need to be addressed before publication.

1. In Supplementary Table 3, provide the p-values for MDA subtype. Also provide detailed comparison methods (CR vs. NR or CR/PR vs. NR). Further, demonstrate that the new subtype is superior by providing the KM curve for each subtype.
2. Figure 2g should be re-analyzed using K=5 and response data to demonstrate reproducibility of subtypes and treatment responses.
3. Supplementary Figure 1 demonstrates the immune signature itself (IFNG, AUC 0.85 / IFNA, AUC 0.94, etc) to predict the response. Apply the immune signatures in Supplementary Figure 1 to the ABACUS and IMvigor010 data to assess the association between immune signatures and responses. Then, discuss what the advantage of new subtypes is compared to the immune signatures. Additionally, Supplementary Figure 7 (immune signatures and substrate signatures) should also be generated based on responses.

Reviewer #3 (Remarks to the Author):

The author address most of the previous comments. The remaining minor comment is regarding the description of Fig. 6c. For the correlation analysis (comments 3.6.2), I meant the interpretation of Fig. 6c regarding KDM5B expression and activated/repressed KDM5B regulon. Correlation can quantify the association between KDM5B expression and KDM5B regulon activity.

RESPONSE TO REVIEWERS' COMMENTS

Reviewer #1

The author has addressed most of my concerns. However, there are still some concerns that need to be addressed before publication.

R1.1a. In Supplementary Table 3, provide the p-values for MDA subtype.

Response: We have added the result for MDA-predicted subtypes to **Supp. Table 3** (shown below). For clarity, we have now sorted the table by subtype-system name. The five PURE01 consensus subtypes were more strongly associated with CR, PR, and NR pathological response than the predicted subtypes.

Supplementary Table 3: Statistical association of PURE01 pathological response (CR/PR/NR) with pre-treatment or predicted subtypes. P.adj values are from Chi-squared tests with Bonferroni (x8) corrections. As we describe in the manuscript's Methods > 'Consensus expression subtypes' section, we transformed ten Lund subtypes into five simpler subtypes by grouping 'Ba/Sq' and 'Ba/Sq-Inf' as 'Ba/Sq'; 'GU' and 'GU-Inf' as 'GU'; and 'UroA-Prog,' 'UroB,' and 'UroC' as 'Uro.'

Classifier	Classes	p	p.adj
Baylor	2	0.19	1
CIT	7	0.86	1
consensusMIBC	6	0.25	1
Lund (10 subtypes)	10	0.70	1
Lund (5 subtypes)	5	0.67	1
MDA	3	0.12	0.93
PURE01, n=82	5	0.0095	0.076
TCGA	5	0.25	1
UNC	2	0.11	0.87

R1.1b. Also provide detailed comparison methods (CR vs. NR or CR/PR vs. NR).

Response: In Methods, we have added a 'Pathological response' section. This clarifies that the PURE01 clinical data supplied pathologic responses as CR, PR, and NR. In addition, we assigned CR, PR, NR, and Unknown response labels for ABACUS pre-treatment samples from 'pathological complete response (PCR) and 'major pathological response' (MPR). The following Methods section, 'Gene set enrichment analysis (GSEA),' notes that we generated signal-to-noise (S2N)-ranked coding gene lists for CERNO tests for PURE01 and ABACUS data, using either subtypes (one subtype vs. all others) or CR vs. NR pathological response. S2N was the difference of means divided by the sum of standard deviations for these two-group calculations.

R1.1c. Further, demonstrate that the new subtype is superior by providing the KM curve for each subtype.

Response: The KM curves for the PURE01 subtypes are shown in **Fig. 1c** and below in **Fig. R1.1c**, which also reports 24-mo censored relapse for BLCAsubtyping-predicted Baylor, CIT, Lund, MDA, TCGA, and UNC subtypes. **Fig. 1e** describes the relationship of Lund, TCGA, consMIBC, and MDA predicted subtypes to PURE01 consensus subtypes. While the KM plot for CIT subtypes has a very small p-value (6.2×10^{-6}), predicted subtypes MC5 and MC6 each contain only one sample; removing MC5 and MC6 resulted in a multiple-testing-corrected log-rank $p = 0.13$.

Figure R1.1c. 24-mo censored Kaplan-Meier (KM) curves for PURE01 n=82 relapse, for predicted Baylor, CIT, cMIBC, Lund, MDA, TCGA, and UNC subtypes, and for PURE01 consensus subtypes. Log-rank p values are uncorrected for multiple hypothesis testing. In the legend for each KM plot, numbers in curved parentheses (e.g., ‘41/4’ for Baylor basal) are the number of samples in the group, then the number of events, i.e., PURE01 relapses.

R1.2. Figure 2g plot should be re-analyzed using K=5 and response data to demonstrate reproducibility of subtypes and treatment responses.

Response: **Fig. 2g** shows KM curves for the MIBC subset of the IMvigor010 trial (atezo treatment arm), comparing predicted S1+S4 to the other predicted subtypes. We grouped these two subtypes because they had the most consistently poor clinical outcomes in PURE01. Here (**Fig. R1.2**), we show the K=5 KM plot for disease-free survival (DFS), for the IMvigor010 trial’s MIBC samples, in that trial’s atezolizumab treatment arm (log-rank $p = 0.0060$).

Figure R1.2. Kaplan-Meier plot for disease-free survival of the atezolizumab arm of the MIBC subset of the IMvigor010 cohort (n=341).

R1.3. Supplementary Figure 1 demonstrates the immune signature (IFNG, AUC 0.85 / IFNA, AUC 0.94, etc.) to predict the response. Apply the [Hallmark] immune signatures in Supplementary Figure 1 to the [whole-cohort, CR vs. NR] ABACUS and IMvigor010 data to assess the association between immune signatures and responses. Then, discuss what the advantage of new subtypes is compared to the immune signatures.

Response: **Fig. R1.3** below shows CERNO-test results for IFNG response (blue) and IFNA response (red) in PURE01, ABACUS, and IMvigor010 MIBC cohorts for which CR and NR data were available. The tables below the graphs show that AUCs for the IFNG and IFNA responses ranged between 0.72-0.80.

Figure R1.3. Tmod CERNO test results for MSigDB v7.2 Hallmark gene sets for IFNG and IFNA responses, for (left to right) the PURE01 n=82 pre-treatment cohort, the ABACUS n=84 pre-treatment cohort, and the IMvigor010 atezolizumab arm. See Methods below. Coding genes in each cohort were ranked by the signal-to-noise ratio (S2N).

The Reviewer astutely identifies a consistent association of the IFNG and IFNA response Hallmarks in responders to immunotherapy in all three cohorts. This result provides intra-trial validation and validates these cohorts compared to other CPI-treated solid tumors. We agree with the Reviewer that IFN Hallmarks are a potential biomarker of CPI response that, with further work, could help identify which patients could benefit from CPI. For example, patients with higher IFN signatures may consider CPI in the neoadjuvant or adjuvant settings. **Fig. 1f** in the manuscript describes the Hallmarks with enhanced or repressed gene expression. IFNG/A are increased in two responsive subtypes (S3 and S2) and repressed in non-responsive subtypes S1 and S4. Yet, S5, which has a 50% non-response (**Fig. 1b**), has high levels of IFNA/G (**Fig. 1f compared to S4**). For risk stratification, patients could be grouped into S1+S4 vs. S2+S3 and S5, as we have done for IMvigor010 in **Fig. 2g**. Our subtyping method and the developed classifier (**Fig. 2a**) are steps towards tumor characterization for precision treatment of MIBC. Upstream of the expression subtypes, we identify active and repressed regulons (**Supp. Fig. 11a**) and recurrent mutations (**Fig. 3c**) in each subtype; both help explain some of the PURE01 cohort's biologic heterogeneity, and both could potentially be targeted with precision approaches. Downstream, we have used regulons (**Fig. 6a**) and the Connectivity Map (CMap, **Fig. 1h**) to identify possible drug targets for CPI-resistant tumors. For example, **Fig. 6** highlights how regulons and mutations can be leveraged to target S1 tumors. S4 tumors, like S1, have repressed IFN gene sets and have different CMap targets (**Fig. 1h**), and overcoming CPI resistance would involve different therapies. Thus, the subtypes we proposed have basic and translational implications. We have added this potential advantage to the discussion section.

Methods for Fig R1.3

For the PURE01 cohort, the manuscript's Methods describe 'RNA sequencing,' 'Pathological response,' and 'Gene set enrichment analysis (GSEA).'

We used PCR and MPR clinical data for the ABACUS pre-treatment cohort to assign CR/PR/NR/Unknown pathological response labels to its samples (see Methods, Pathological response).

For the IMvigor010 cohort, we downloaded fastq files and metadata from EGAD00001007575. RNA-Seq reads were aligned to the GRCh38 reference genome with STAR v2.5.2. Read counts were generated with HTseq count using Ensembl v103 gene annotations. For MIBC samples in the atezolizumab trial arm (the 'treatment' group), n=127 samples with relapseID "relapse" were assigned as non-responders (NR). In contrast, n=144 samples with relapseID = "nonrelapse" were assigned as complete responders (CR).

For all three cohorts we sorted protein-coding genes by two-group (CR vs. NR) signal-to-noise ratio (S2N, difference-of-means/sum-of-standard-deviations), which we calculated from gene-level FPKMs. We then input the S2N-ranked gene lists into tmod v0.50.11, with which we ran CERNO tests using MSigDB v7.2 Hallmark gene sets, using qval=0.05 as a filter. We reported tmodEvidenceplots for IFNA response and IFNG response gene sets.

R3.3 Additionally, Supplementary Figure 7 (immune signatures and substrate signatures) should also be generated based on responses.

Response: **Supp. Fig. 7** showed boxplots for ESTIMATE and MCP-counter deconvolution results by PURE01 n=82 subtypes. As requested, we have added similar boxplots for ESTIMATE and MCP-counter deconvolution results by PURE01 response (CR/PR/NR). After correcting for multiple testing, only cytotoxic lymphocytes (CTLs) and CD8 T cells were statistically associated with pathological response ($p_{adj} < 0.05$). Because we have limited DSP data, we have removed the DSP protein results from this supplemental figure and now report DSP results in **Fig. 4c,d**, and **Supp. Fig. 8**.

Figure R3.3. The updated Supp. Fig. 7. P values are from Kruskal-Wallis tests with Bonferroni corrections. Panels (d) and (e) show only corrected p values < 0.15.

Reviewer #3

The author address most of the previous comments. The remaining minor comment is regarding the description of Fig. 6c. For the correlation analysis (comments 3.6.2), I meant the interpretation of Fig. 6c regarding KDM5B expression and activated/repressed KDM5B regulon. Correlation can quantify the association between KDM5B expression and KDM5B regulon activity.

Figure R3.1. Correlation of KDM5B expression and KDM5B regulon activity. **a)** Bulk RNA-Seq data for the PURE01 n=82 pre-treatment cohort. Pearson's correlation: $cor = 0.72$, (uncorrected) $p = 1.8 \times 10^{-14}$, alt. hyp.: true correlation $\neq 0$. Note that this correlation was calculated for all 82 samples, not just those in which the KDM5B regulon is 'activated' (red dots). **b)** For sc-RNA-Seq data in Fig. 6c's epithelial cell cluster, KDM5B(+) regulon AUC values as a function of normalized KDM5B expression. Pearson $cor = 0.56$, (uncorrected) $p = 8.8 \times 10^{-296}$.

Response: From bulk RNA-Seq data for PURE01 (Fig. R3.1a), we found a positive correlation between KDM5B expression (RNA-Seq FPKMs) and KDM5B regulon activity, particularly for samples with activated KDM5B regulons. For scRNA-Seq data for three MIBC tumors (Fig. R3.1b), we found a positive correlation between normalized KDM5B expression and a KDM5B regulon activity metric: the KDM5B(+) AUC value.

Methods for Fig. R3.1

For bulk RNA-Seq data in a), see the 'Regulon analysis' section in Methods. Regulon activities of 232 transcription factors, including KDM5B, are given in Supp. Table 16; the corresponding regulon activity status values (+1=activated, 0=undefined, -1=repressed) are given in Supp. Table 17.

For scRNA-Seq data in b), see the 'Single-cell RNA sequencing of MIBC tumors' section in Methods. KDM5B expression was normalized as described in that section ("Post filtering, remaining features (i.e., genes) across 15,922 unique cells were scaled and centered to define the relative expression of features across cells.") KDM5B(+) AUC values were taken as a metric of KDM5B(+) regulon activity.

REVIEWERS' COMMENTS

Reviewer #1 (Remarks to the Author):

All concerns have been addressed by authors.

Reviewer #3 (Remarks to the Author):

The author addressed my comments.